# Olivia: Harmonizing Time Series Foundation Models
# with Power Spectral Density

**Jingru Fei** [1]  **Kun Yi** [2][3]  **Alex Xing Wang** [4]  **Qingsong Wen** [5]  **Xiangxiang Zhu** [6]  **Wei Fan** [7]

## Abstract

Time series foundation models rely on large-scale pretraining over diverse datasets across domains, yet their heterogeneity in temporal patterns could hinder the effectiveness of training and learning transferable time series representations. Inspired a fundamental concept, normalized power spectral density (PSD) in signal processing, we assume harmonizing datasets via PSDs in the spectral domain could reduce mismatches and enhance pretraining. We then go beyond the direct intractable minimization optimization and innovatively reformulate it as a principled *harmonization* approach. Specifically, we propose *Harmonizer*, a module that reshapes spectral structures and implicitly harmonizing PSDs across datasets, which theoretically corresponds to a shared reparameterization of second-order temporal correlations. Our theoretical analysis further reveals token interactions with Harmonizer can be efficiently mediated by a compact set of resonators, motivating a *HarmonicAttention* design that performs self-attention in a low-dimensional interaction space. Then, we propose *Olivia*, a novel time series foundation model built upon these harmonization mechanisms. Extensive experiments on several large-scale benchmarks (TSLib, GIFT-Eval, and GluonTS), demonstrate Olivia consistently achieves state-of-the-art performance under zero-shot, few-shot, and full-shot forecasting scenarios. Our code is at https://github.com/TSTS13/Olivia.

## 1. Introduction

Time series has been a fundamental modality in many real-world applications, including energy management, traffic prediction, climate modeling, finance, and healthcare (Qiu et al., 2024; Yi et al., 2025; Qiu et al., 2025). Modern end-to-end deep learning models for time series forecasting, which could be built upon recurrent networks, convolution structures, Transformer architectures, etc (Luo & Wang, 2024; Liu et al., 2023; Fei et al., 2025), have achieved strong performance on individual datasets, but they are typically trained in task- or dataset-specific settings, limiting their ability to generalize across domains. Inspired by large-scale pretraining in natural language processing (Wang et al., 2023) and computer vision (Awais et al., 2025), recent work has been exploring time series foundation models (Liu et al., 2024; Ansari et al., 2024), which aim to learn an unified, general-purpose model from diverse time series data and transfer to downstream forecasting tasks effectively.

Generally, time series foundation models are pretrained on large collections of datasets spanning multiple application domains (Wang et al., 2025b). The diversity from different data is a significant characteristic in large-scale pretraining, which aims for exposing the model to a wide range of temporal behaviors (Woo et al., 2024). As illustrated in the left part of Figure 1(a), time series originating from different domains exhibit markedly different temporal patterns, reflecting variations in periodic structures and long-term temporal dependencies that arise from domain-specific data generation processes, such as distinct physical dynamics, operational cycles, environmental influences, and measurement settings (Liang et al., 2024). While such diversity is a prerequisite for learning broadly applicable temporal knowledge, it also introduces substantial challenges to pretraining. From an optimization perspective, jointly training on datasets with disparate temporal characteristics often leads to slower convergence and suboptimal optimization behavior. From a representation learning perspective, the model might need to simultaneously accommodate incompatible temporal structures, which could make it difficult to form unified and transferable representations that could generalize well across datasets. Then, these challenges naturally motivate us to ask an important research question: *how can time series foundation models learn more effectively on diverse pretraining datasets?*

To answer this question, the core idea is to make time series foundation models characterize and address such diversity

---

[1]Beijing Institute of Technology [2]North China Institute of Computing Technology [3]State Information Center [4]Victoria University of Wellington [5]Squirrel Ai Learning [6]Northwest Polytechnical University [7]University of Auckland. Correspondence to: Kun Yi <kunyi.cn@gmail.com>, Wei Fan <wei.fan@auckland.ac.nz>.

*Proceedings of the $43^{rd}$ International Conference on Machine Learning*, Seoul, South Korea. PMLR 306, 2026. Copyright 2026 by the author(s).

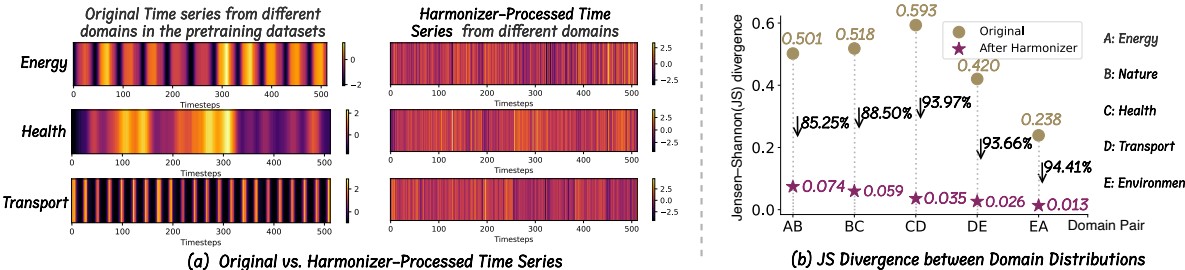

*Figure 1.* **(a):** Original and Harmonizer-processed time series from different domains. Raw time series from multiple domains (left) show distinct temporal patterns. After processing (right), patterns show more consistent across domains. **(b):** Jensen–Shannon (JS) divergence between dataset-level normalized power spectral density distributions across different domain pairs. JS divergence is bounded in $[0, \ln 2]$ (approximately $[0, 0.693]$), with larger values indicating greater spectral discrepancy. More details can be found in the Appendix C.

across data and domains in a principled manner. Inspired by a fundamental concept in the domain of signal processing (Proakis, 2007; Hayes, 1996), we have found the normalized power spectral density (PSD) could serve as an informative dataset-level descriptor by capturing the frequency-wise distribution of temporal variations that could reflect the underlying second-order temporal structure and provide a stable and discriminative summary across datasets (Chatfield & Xing, 2019; Fulcher & Jones, 2017), which directly suggests a potential principled remedy: rather than treating dataset diversity as an unstructured domain gap, we assume that harmonizing datasets via PSDs in the spectral domain, could reduce mismatches between their PSDs and thus improve the efficacy of large-scale time series pretraining.

Alone this line, a following straightforward solution is to measure PSDs and minimize their discrepancies among datasets. As an instance, the distributional discrepancies between PSDs of datasets can be quantified using standard measures such as the Jensen–Shannon (JS) divergence (Lin, 2002; Endres & Schindelin, 2003). Experimentally, as shown in Figure 1(b), different domain pairs exhibit substantial JS divergence in their PSDs quantitatively indicating pronounced spectral discrepancies, and minimizing such divergences seems to be a reasonable objective for promoting PSD consistency. However, directly pursuing this objective is non-trivial in large-scale time series pretraining. Since this objective is defined over aggregated, dataset-level spectral statistics, whereas time series foundation models are usually trained via instance (sequence)-level optimization, this mismatch makes direct optimization hard to implement.

To address above challenges, we go beyond direct divergence minimization for PSDs and innovatively reformulate it as a structural and principled *harmonization* approach. We demonstrate that cross-domain spectral discrepancies can be effectively addressed by projecting time series onto a shared representation space, based on which, we propose an according module termed *Harmonizer*. Specifically, Harmonizer introduces learnable orthogonal temporal transformations that reorganize temporal dynamics, reshaping spectral structures and implicitly harmonizing PSDs across datasets. From a theoretical perspective, it corresponds to a shared

reparameterization of second-order temporal correlations, which induces a common structural form across datasets. Moreover, Harmonizer further reshapes temporal relationships in attention mechanisms in Transformer-based time series foundation models (He et al., 2025). Our theoretical analysis reveals that token interactions can be efficiently mediated by a compact set of resonators, motivating a *HarmonicAttention* design that performs self-attention (Vaswani et al., 2017) in a low-dimensional interaction space instead of dense all-to-all interactions. Derived from harmonized temporal representations, these resonators act as a condensed intermediary for modeling temporal relationships.

Building upon these insights, we introduce a novel time series foundation model, *Olivia*, for forecasting. Olivia starts with incorporating *Harmonizer*, which comprises two bidirectional submodules to perform temporal reorganization before encoding and representation recovery after decoding. For the core architecture, Olivia follows a Transformer-style encoder–decoder design, but departs from standard formulations by employing HarmonicAttention as its primary interaction mechanism which results in a *HarmonicFormer* architecture that models temporal relationships through low-dimensional harmonic interactions rather than dense all-to-all attention. In summary, our contributions are:

- We introduce a principled *harmonization* approach for addressing diversity across pretraining time series datasets by implicitly harmonizing PSDs across data through the reorganization of second-order temporal correlations.

- We propose *HarmonicAttention*, a novel attention mechanism enabling low-dimensional token interactions mediated by a compact set of resonators, and further introduce *HarmonicFormer* for efficient foundational modeling.

- We propose *Olivia*, a novel time series foundation model built upon the harmonization and conduct extensive experiments on two large-scale benchmarks (TSLib and GIFT-Eval) and extra 6 datasets provided by GluonTS, which demonstrate Olivia achieves superior performance under zero-shot, few-shot, and full-shot forecasting scenarios, consistently outperforming state-of-the-art methods.

## 2. Related Work

**Time Series Foundation Models.** Time series foundation models have recently emerged as a promising paradigm through large-scale pretraining on multi-source time series data (Liang et al., 2024; Qiu et al., 2024; Awais et al., 2025). Most existing time series foundation models can be categorized into encoder-only, decoder-only, or encoder–decoder architectures. Among encoder-only models, MOMENT (Goswami et al., 2024) utilizes T5-style masked modeling for self-supervised representation learning, while MOIRAI (Woo et al., 2024) employs masked Transformers for universal probabilistic forecasting. Decoder-only models such as Sundial (Liu et al., 2025) and Time-MoE (Shi et al., 2024) utilize autoregressive decoders. Time-MoE scales deterministic forecasting via sparse Mixture-of-Experts, whereas Sundial integrates decoders within a generative flow-matching framework. Encoder–decoder models constitute another architectural paradigm for time series forecasting. Chronos (Ansari et al., 2024) treats forecasting as discrete token prediction through quantization. Recently, SEMPO (He et al., 2025) introduced energy-aware spectral modeling and prompt-based adaptation in this framework.

**Domain Generalization in Foundation Models.** Foundation models strive for universal generalization by mitigating cross-domain discrepancies (Liang et al., 2024). MOIRAI (Woo et al., 2024) uses frequency-aware patching and mixture-based heads to absorb input-output variations, while ROSE (Wang et al., 2025b) employs spectral masking and adaptive registers to isolate domain-specific features. Time-MoE (Shi et al., 2024) utilizes sparse routing for implicit pattern separation, and SEMPO (He et al., 2025) leverages a mixture-of-prompts to capture cross-domain commonalities. However, these frameworks primarily achieve domain generalization via architectural modularization or capacity-based specialization, rather than explicitly addressing the fundamental discrepancies in temporal distributions. In this paper, we introduce Olivia that moves beyond structural modularization by enforcing a PSD-consistent latent space. Through our proposed Harmonizer, Olivia explicitly aligns disparate spectral signatures into a unified aligned subspace, ensuring that the model learns universal temporal primitives that are invariant to domain-specific shifts.

## 3. Preliminaries

**Power Spectrum Density.** Power Spectral Density (PSD) is a mathematical measure used to characterize how the energy of a time series signal is distributed across frequency components (Oppenheim, 1999). Specifically, given a dataset $\mathcal{D}$ consisting of time series samples $X \in \mathbb{R}^T$, the dataset-level normalized PSD $\mathcal{P}_\mathcal{D}(\omega)$ is defined as:

$$\mathcal{P}_\mathcal{D}(\omega) \triangleq \frac{1}{|\mathcal{D}|} \sum_{X \in \mathcal{D}} S_X(\omega_k) / \sum_k \left( \frac{1}{|\mathcal{D}|} \sum_{X \in \mathcal{D}} S_X(\omega_k) \right) \quad (1)$$

Here, $S_X(\omega_k)$ denotes the PSD of an individual sequence $X$, estimated via the periodogram at discrete frequencies $\omega_k = \frac{2\pi k}{T}$: $S_X(\omega_k) \triangleq \frac{1}{T} \left| \sum_{t=0}^{T-1} X_t\, e^{-j\omega_k t} \right|^2$.

As implied by the preceding definitions, the power spectral density depends solely on the magnitude of Fourier coefficients and is therefore invariant to global temporal shifts (i.e., phase shifts in the frequency domain), while exhibiting relative robustness to minor local temporal misalignments (Percival & Walden, 1993; Brillinger, 2001). This property makes PSD a particularly suitable representation for comparing signals collected under heterogeneous conditions. Moreover, many sources of domain shift in time series, such as variations in sampling rates, sensor characteristics, and dominant periodicities, are naturally reflected as changes in the allocation of spectral energy across frequency bands (Fulcher & Jones, 2017; Chatfield & Xing, 2019).

## 4. Methodology

As mentioned above, the power spectral density (PSD) possesses several desirable properties that make it a principled basis for characterizing and comparing heterogeneous time series datasets. Leveraging these properties, we propose *Olivia*, a time series foundation model that learns domain-agnostic temporal representations via a PSD-guided modeling strategy, thereby promoting consistent representation learning across diverse data sources.

### 4.1. Overview

Following the architectural paradigm of recent time series foundation models (Ansari et al., 2024; He et al., 2025; Wang et al., 2025b), *Olivia* adopts an encoder–decoder framework, as illustrated in Figure 2. The architecuture comprises two core components. (1) The *Harmonizer* is a PSD-guided transformation module that implicitly harmonizes spectral characteristics across heterogeneous datasets by aligning temporal structures prior to encoding and restoring the decoder outputs to the original domain-specific temporal space. (2) The *HarmonicFormer* is a Transformer-derived backbone that operates on the PSD-harmonized representations. It extends the standard Transformer architecture with the proposed *HarmonicAttention*, which replaces dense token-wise dependencies with structured interactions in a low-dimensional subspace, thereby enabling efficient and scalable modeling of global temporal dependencies.

### 4.2. Harmonizer: Bidirectional Power Spectral Density Reparameterization

Discrepancies between heterogeneous time series datasets often manifest as shifts in their spectral energy distributions. These differences in data distribution can be rigorously quantified in the frequency domain by comparing PSDs using various distributional discrepancy measures, such as the

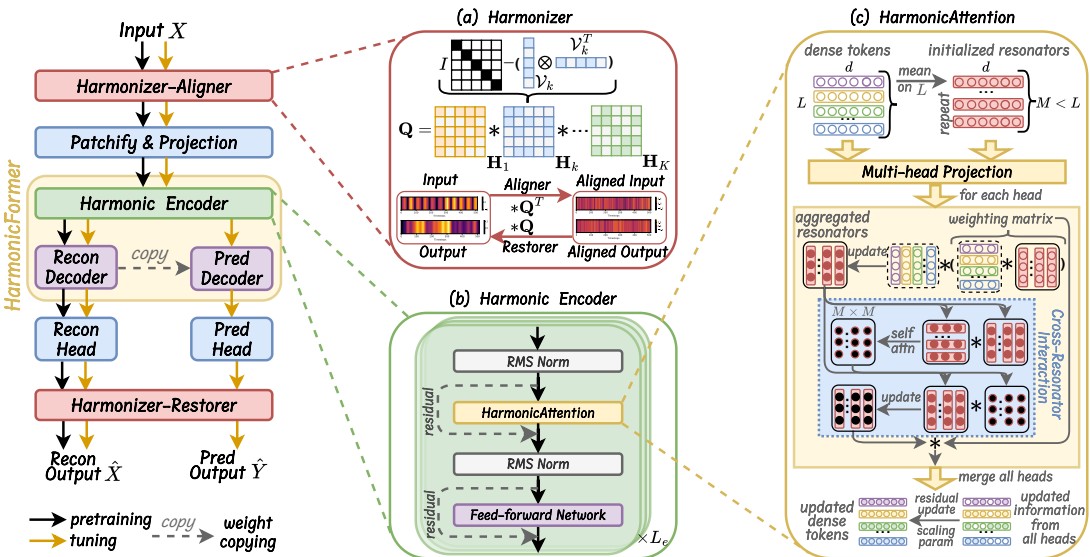

*Figure 2.* The overall architecture of Olivia. *Olivia* adopts an encoder–decoder architecture centered on (i) the Harmonizer, which consists of an Aligner and a Restorer to perform bidirectional temporal reorganization across heterogeneous datasets, and (ii) the HarmonicFormer, a scalable backbone that utilizes HarmonicAttention for efficient temporal dependency modeling within a compressed subspace.

Kullback–Leibler (KL) divergence (Kullback & Leibler, 1951), the Jensen–Shannon (JS) divergence (Cover, 1999; MacKay, 2003), or the Wasserstein distance (Villani, 2021). Among these metrics, we instantiate this distributional comparison using the JS divergence as a representative choice, leveraging its symmetric and bounded properties to ensure stable optimization. A seemingly straightforward approach to cross-domain pretraining is to enforce a spectral consistency constraint, encouraging the foundation model to map diverse inputs into a shared canonical spectral space. This could be formalized as minimizing discrepancies of PSD as:

**Definition 1** (**Power Spectral Density Harmonization**). Let $\mathcal{T}_{\text{pretrain}}$ be a collection of pretraining datasets. For any pair $\mathcal{D}_i, \mathcal{D}_j \in \mathcal{T}_{\text{pretrain}}$, let $f_\theta$ be a model that reparameterize these datasets to a latent space with corresponding spectral energy distributions $\mathcal{P}_{\mathcal{D}_i}^{(\theta)}(\omega)$ and $\mathcal{P}_{\mathcal{D}_j}^{(\theta)}(\omega)$. The harmonization objective seeks to find the optimal parameters $\theta^*$ as:

$$\theta^* = \arg\min_\theta \sum_{\mathcal{D}_i, \mathcal{D}_j \in \mathcal{T}_{\text{pretrain}}} \text{JSD}\left(\mathcal{P}_{\mathcal{D}_i}^{(\theta)}(\omega) \,\Big\|\, \mathcal{P}_{\mathcal{D}_j}^{(\theta)}(\omega)\right) \quad (2)$$

where $\text{JSD}(\cdot\|\cdot)$ denotes the Jensen-Shannon Divergence, measuring the functional similarity between spectral energy distributions across the pretraining collection.

However, directly optimizing this objective is impractical in large-scale pretraining, as training is performed via mini-batch stochastic optimization and the dataset-level objective can only be approximated from individual batches, leading to highly noisy and often misaligned gradient estimates, unstable training dynamics, and poor convergence (Bottou et al., 2018; Qian & Klabjan, 2020). To overcome these limitations, we pivot from direct divergence minimization to a structural alignment approach that operates on second-

order temporal statistics underlying power spectral densities. In particular, we consider a structural reparameterization in which cross-domain spectral discrepancies can be mitigated by projecting time series data onto a shared latent subspace.

**Proposition 1.** *Let $\mathcal{U} \subset \mathbb{R}^T$ be an $r$-dimensional subspace that is invariant under the second-order moment matrices of all datasets $\mathcal{D}$ in the pretraining collection $\mathcal{T}_{\text{pretrain}}$, that is, $\boldsymbol{\Sigma}_{\mathcal{D}}^X \mathcal{U} \subseteq \mathcal{U}$ where $\boldsymbol{\Sigma}_{\mathcal{D}}^X \triangleq \mathbb{E}[(X_{\mathcal{D}})^\top X_{\mathcal{D}}]$. Then, there exists a shared orthogonal matrix $\mathbf{Q} \in \mathbb{R}^{T \times T}$ whose first $r$ columns span $\mathcal{U}$, such that the transformed signal $\mathcal{X}_{\mathcal{D}} \triangleq X_{\mathcal{D}} \mathbf{Q}^\top$ yields a block-diagonal moment matrix:*

$$\boldsymbol{\Sigma}_{\mathcal{D}}^{\mathcal{X}} = \mathbf{Q} \boldsymbol{\Sigma}_{\mathcal{D}}^X \mathbf{Q}^\top = \text{diag}\left(\boldsymbol{\Lambda}_{\mathcal{D}}, \boldsymbol{\Phi}_{\mathcal{D}}\right), \quad (3)$$

*where $\boldsymbol{\Lambda}_{\mathcal{D}} \in \mathbb{R}^{r \times r}$ characterizes the shared subspace dynamics and $\boldsymbol{\Phi}_{\mathcal{D}} \in \mathbb{R}^{(T-r) \times (T-r)}$ captures dataset-specific variations in the orthogonal complement $\mathcal{U}^\perp$.*

Proposition 1 establishes that an appropriate orthogonal transformation $\mathbf{Q}$ can decouple shared second-order temporal correlations from dataset-specific variations. By projecting time series data into this common coordinate system, the resulting representations expose a shared latent subspace that remains invariant across the entire collection of datasets. A complete derivation is provided in Appendix A.1.

Motivated by this result, we introduce the *Harmonizer*, a structural module designed to implement this shared reparameterization. By reorganizing second-order temporal correlations, the Harmonizer implicitly enforces PSD consistency across diverse datasets without the instability of direct divergence optimization. To maintain the integrity of the signal, this reparameterization is designed to be fully invertible, inducing a bidirectional mapping between the original

and transformed temporal domains. This is operationalized through two complementary submodules: the *Aligner*, which projects time series data into the shared subspace, and the *Restorer*, which maps them back to the original domain.

### 4.2.1. ALIGNER

Given a raw univariate time series $X \in \mathbb{R}^{1 \times T}$, the Aligner performs a structural reparameterization by projecting the time series into a reorganized coordinate system as:

$$\mathcal{X} = X\mathbf{Q}^{\top}, \quad \text{s.t. } \mathbf{Q}^{\top}\mathbf{Q} = \mathbf{I}, \qquad (4)$$

where $\mathbf{Q} \in \mathbb{R}^{T \times T}$ is a learnable orthogonal matrix shared across the entire pretraining collection. This transformation preserves the $\ell_2$ norm and is strictly invertible, ensuring that temporal information is reorganized without loss of variance or structural integrity (Strang, 2012; Horn & Johnson, 2012).

The primary challenge lies in optimizing $\mathbf{Q}$ while maintaining strict orthogonality. Rather than relying on soft constraints or iterative orthogonalization (e.g., Stiefel manifold optimization), we adopt an explicit parameterization via Householder reflections (Golub & Van Loan, 2013). Specifically, $\mathbf{Q}$ is decomposed into a sequence of $K$ reflections:

$$\mathbf{Q} = \prod_{k=1}^{K} \mathbf{H}_k, \quad \mathbf{H}_k = \mathbf{I} - 2\mathcal{V}_k\mathcal{V}_k^{\top}, \qquad (5)$$

where each $\mathcal{V}_k \in \mathbb{R}^T$ is a learnable unit vector. Because each $\mathbf{H}_k$ is an elementary orthogonal matrix, their product is orthogonal by construction (Horn & Johnson, 2012). This approach allows $\mathbf{Q}$ to be optimized via standard gradient-based methods while guaranteeing that it remains on the orthogonal group. The hyperparameter $K$ regulates the transformation's degrees of freedom, enabling the model to capture complex shared temporal structures even across datasets with highly divergent power spectral profiles.

### 4.2.2. RESTORER

After deep temporal modeling within the HarmonicFormer backbone, the resulting processed representation $\mathcal{Y} \in \mathbb{R}^{1 \times T}$ resides within the canonical aligned subspace. To facilitate domain-specific inference, the Restorer executes an inverse isometric mapping to project these latent representations back onto the original domain as follows:

$$Y = \mathcal{Y}\mathbf{Q}, \qquad (6)$$

where $\mathbf{Q} \in \mathbb{R}^{T \times T}$ is the same universal orthogonal operator utilized by the Aligner. This bidirectional reparameterization ensures that the overall pipeline preserves the original temporal structure at the output, while fully leveraging the spectral harmonization achieved within the shared subspace.

### 4.3. HarmonicFormer: Efficient Attention in the PSD-Harmonized Subspace

According to Proposition 1, projecting time series data into a common coordinate system reveals that a substantial portion of the spectral energy is concentrated within a shared $r$-dimensional subspace. This structural reorganization naturally motivates the design of efficient attention that operates on these harmonized and compressed representations.

### 4.3.1. HARMONICATTENTION

Let $Z = \{z_\ell\}_{\ell=1}^L$, with $z_\ell \in \mathbb{R}^d$, denote token representations derived from the output $\mathcal{X}$ of the Harmonizer's Aligner. Here, $L$ and $d$ represent the number of patches and the embedding dimensions, respectively. These tokens are generated through standard patchification and linear embedding (Liu et al., 2025; He et al., 2025).

**Proposition 2.** *Let $\mathcal{X} \in \mathbb{R}^{1 \times T}$ denote an aligned temporal signal with finite second moments, and let $\mathbf{\Sigma}_{\mathcal{X}} \triangleq \mathbb{E}[\mathcal{X}^{\top}\mathcal{X}] \in \mathbb{R}^{T \times T}$ denote its second-order moment matrix. Assume that $\mathbf{\Sigma}_{\mathcal{X}}$ admits a block-diagonal decomposition $\mathbf{\Sigma}_{\mathcal{X}} = \text{diag}(\mathbf{\Lambda}, \mathbf{\Phi}), \mathbf{\Lambda} \in \mathbb{R}^{r \times r}, \mathbf{\Phi} \in \mathbb{R}^{(T-r) \times (T-r)}$, with $r \ll T$. Let token representations be generated by linear temporal operators $z_\ell = M_\ell \mathcal{X}^{\top}, M_\ell \in \mathbb{R}^{d \times T}, \ell = 1, \ldots, L$, and stack them as $Z = [z_1^{\top}; \ldots; z_L^{\top}] \in \mathbb{R}^{L \times d}$. Define the token Gram matrix $\mathbf{C}_Z \triangleq \mathbb{E}[ZZ^{\top}] \in \mathbb{R}^{L \times L}, [\mathbf{C}_Z]_{\ell,\ell'} = \mathbb{E}[z_\ell^{\top} z_{\ell'}]$. Partition each temporal operator $M_\ell$ conformably with the block structure of $\mathbf{\Sigma}_{\mathcal{X}}$ as $M_\ell = [A_\ell \ B_\ell], A_\ell \in \mathbb{R}^{d \times r}, B_\ell \in \mathbb{R}^{d \times (T-r)}$. Then, for all $\ell, \ell'$, the token Gram entries admit the decomposition $[\mathbf{C}_Z]_{\ell,\ell'} = \text{tr}(A_\ell \mathbf{\Lambda} A_{\ell'}^{\top}) + \text{tr}(B_\ell \mathbf{\Phi} B_{\ell'}^{\top})$. Moreover, the truncated Gram matrix defined by $[\mathbf{C}_Z^{(r)}]_{\ell,\ell'} \triangleq \text{tr}(A_\ell \mathbf{\Lambda} A_{\ell'}^{\top})$ is positive semidefinite and satisfies $\text{rank}(\mathbf{C}_Z^{(r)}) \leq dr$. In addition, the approximation error is bounded as*

$$\left| [\mathbf{C}_Z]_{\ell,\ell'} - [\mathbf{C}_Z^{(r)}]_{\ell,\ell'} \right| \leq \|\mathbf{\Phi}\|_2 \|B_\ell\|_F \|B_{\ell'}\|_F. \qquad (7)$$

The proof of Proposition 2, detailed in Appendix A.2, establishes that PSD-aligned representations allow token correlations to be decomposed into a dominant low-rank principal term and a bounded residual. This concentration of temporal energy provides a rigorous justification for approximating dense dependencies with a compact set of spectral modes. We operationalize this theoretical result through HarmonicAttention, a mechanism that mediates global interactions via $M$ latent resonators across three functional stages.

Specifically, following the standard attention convention (Vaswani et al., 2017), HarmonicAttention projects $Z \in \mathbb{R}^{L \times d}$ into head-specific subspaces. For each attention head, let $W_h \in \mathbb{R}^{d \times P}$ be a learnable projection matrix, and define the projected tokens as $\tilde{Z}^{(h)} = ZW_h \in \mathbb{R}^{L \times P}$.

**Token-to-Resonator Aggregation.** We first synthesize a set of $M$ latent resonators ($M < L$) that aggregate global information from the tokens $Z$. Specifically, we initialize a global resonator as $\mathcal{G} = \frac{1}{L}\mathbf{1}^{\top}Z \in \mathbb{R}^{1 \times d}$, i.e., the mean token representation. Then we project it to the head-specific subspace as $\tilde{r}^{(h)} = \mathcal{G}W_h \in \mathbb{R}^{1 \times P}$. Using the projected

global resonator as a shared query template, we compute a token-to-resonator aggregation matrix $A^{(h)}$ as $A^{(h)} = \text{Softmax}_{\text{token}}\left(\tilde{Z}^{(h)}(\mathbf{1}_M \otimes \tilde{r}^{(h)})^\top / \sqrt{P}\right) \in \mathbb{R}^{L \times M}$, where $\text{Softmax}_{\text{token}}$ normalizes each column of $A^{(h)}$ over the $L$ tokens. Then the aggregated resonator representations are:

$$R^{(h)} = (A^{(h)})^\top \tilde{Z}^{(h)} \in \mathbb{R}^{M \times P}. \tag{8}$$

**Cross-Resonator Interaction.** Unlike standard self-attention (Vaswani et al., 2017), whose computational cost scales quadratically with the number of tokens $L$, we perform interaction within the compact $M \times M$ resonator space. Specifically, for each head, we treat the aggregated resonator representations $R^{(h)}$ as resonator-wise queries, keys, and values. The resonator interaction is formulated as:

$$\text{ResAct}(R^{(h)}) = \text{Softmax}_{\text{res}}\left(R^{(h)}(R^{(h)})^\top / \sqrt{P}\right) R^{(h)}, \tag{9}$$

where $\text{Softmax}_{\text{res}}$ normalizes across the $M$ resonators.

**Global Resonator Projection.** The interacted resonator information is redistributed back to the head-specific token $\text{Head}^{(h)} \in \mathbb{R}^{L \times P}$ using the same weights $A^{(h)}$ employed during token-to-resonator aggregation, ensuring the final output consistent with the spectral alignment:

$$\text{Head}^{(h)} = A^{(h)} \text{ResAct}(R^{(h)}). \tag{10}$$

Then it is mapped from the head-specific subspace to the original token space via the learnable output projection $W_h^\top$. The final output is obtained by aggregating all heads with a residual connection, scaled by a learnable parameter $\gamma$:

$$\text{HarmonicAttn}(Z) = Z + \gamma \sum_{h=1}^{H} \text{Head}^{(h)} W_h^\top, \tag{11}$$

where $H$ denotes the number of head-specific subspaces in HarmonicAttention (abbreviated as HarmonicAttn).

**Complexity and Scalability.** By mediating interactions through a resonator bottleneck, HarmonicAttention reduces complexity from $\mathcal{O}(L^2 P)$ to $\mathcal{O}(LMP + M^2 P)$. For long sequences where $M \ll L$, this architecture provides linear scalability while maintaining the capacity to model global dependencies through the harmonized spectral subspace.

### 4.3.2. HARMONICFORMER

Building upon HarmonicAttention, we develop the *HarmonicFormer* architecture as the central modeling backbone of *Olivia*. HarmonicFormer consists of a stacked encoder–decoder configuration where HarmonicAttention serves as the primary mechanism for interaction. This design enables efficient global dependency modeling directly within the PSD-aligned temporal subspace.

By operating on representations spectrally aligned by the Harmonizer, HarmonicFormer bridges structured low-dimensional spectral interactions with the representative depth of Transformers. The result is a highly scalable and expressive backbone optimized for large-scale time-series pretraining across heterogeneous datasets.

### 4.4. Model Training

To accommodate varying downstream tasks, we define distinct output formulations and optimization objectives for the pretraining and tuning phases. A comprehensive description of these stages, including specific loss functions and architectural headers, is provided in Appendix B.

## 5. Experiments

### 5.1. Experimental Setup

**Baselines.** To ensure a fair and thorough evaluation, we compare our method with a diverse set of baselines, including (i) pretrained time series foundation models: SEMPO (He et al., 2025), Time-MoE (Shi et al., 2024), Timer (Liu et al., 2024), Moirai (Woo et al., 2024), Chronos (Ansari et al., 2024), TimesFM (Das et al., 2024), Moment (Goswami et al., 2024), Sundial (Liu et al., 2025), ROSE (Wang et al., 2025b), TTM (Ekambaram et al., 2024); (ii) LLM-based models: Time-LLM (Jin et al., 2023), GPT4TS (Zhou et al., 2023), S²IP-LLM (Pan et al., 2024); and (iii) task-specific forecasting models: iTransformer (Liu et al., 2023), DLinear (Zeng et al., 2023), PatchTST (Nie, 2022), TimesNet (Wu et al., 2022), Stationary (Liu et al., 2022b), FEDformer (Zhou et al., 2022).

**Datasets.** For pretraining, we select a subset from the large-scale publicly available time series collection UTSD (Liu et al., 2024), which contains approximately 67 million time points and covers multiple domains, including Transport, Environment, Health, Energy, and Nature. Following Timer (Liu et al., 2024) and SEMPO (He et al., 2025), we split the pretraining data into training and validation sets with a 9:1 ratio. For model evaluation, we conduct experiments on the Time-Series-Library (TSLib) benchmark (Wu et al., 2022), the GIFT-Eval benchmark (Aksu et al., 2024), and datasets provided by GluonTS (Alexandrov et al., 2020).

Additional details of the experimental setup including the implementation details are provided in Appendix E.

### 5.2. Main Results

As detailed in Tables 1 and 2, Olivia consistently demonstrates strong zero-shot generalization across multiple benchmarks, with particularly pronounced improvements over representative foundation model baselines. On the TSLib benchmark (see Table 1), Olivia achieves clear gains over the lightweight model SEMPO on large-scale datasets

*Table 1.* Zero-shot results on the TSLib benchmark (Wu et al., 2022), where the lookback window length is fixed to $T = 512$ following prior works (Goswami et al., 2024; He et al., 2025), and the reported results are averaged over multiple prediction horizons $\tau \in \{96, 192, 336, 720\}$; the best and second-best performances are highlighted in **red** and **blue**, respectively. A lower MSE/MAE indicates a better result. '-' denotes dataset used in pre-training and excluded in the evaluation. 'S': Small, 'B': Base, 'L': Large.

| Models | Olivia Ours | | SEMPO [NeurIPS'25] | | Time-MoE$_B$ [ICLR'25] | | Time-MoE$_L$ [ICLR'25] | | Timer [ICML'24] | | Moirai$_S$ [ICML'24] | | Moirai$_B$ [ICML'24] | | Moirai$_L$ [ICML'24] | | Chronos$_S$ [TMLR'24] | | Chronos$_B$ [TMLR'24] | | Chronos$_L$ [TMLR'24] | | TimesFM [ICML'24] | |
|---|---|---|---|---|---|---|---|---|---|---|---|---|---|---|---|---|---|---|---|---|---|---|---|---|
| Metrics | MSE | MAE | MSE | MAE | MSE | MAE | MSE | MAE | MSE | MAE | MSE | MAE | MSE | MAE | MSE | MAE | MSE | MAE | MSE | MAE | MSE | MAE | MSE | MAE |
| ETTh1 | 0.399 | 0.421 | 0.410 | 0.430 | 0.445 | 0.449 | 0.435 | 0.449 | 0.451 | 0.463 | 0.448 | 0.432 | 0.433 | 0.431 | 0.466 | 0.443 | 0.551 | 0.463 | 0.524 | 0.439 | 0.541 | 0.443 | 0.489 | 0.444 |
| ETTh2 | 0.339 | 0.388 | 0.341 | 0.391 | 0.566 | 0.479 | 0.477 | 0.452 | 0.366 | 0.408 | 0.355 | 0.401 | 0.360 | 0.399 | 0.382 | 0.397 | 0.394 | 0.409 | 0.392 | 0.401 | 0.385 | 0.400 | 0.396 | 0.405 |
| ETTm2 | 0.291 | 0.344 | 0.289 | 0.343 | 0.538 | 0.463 | 0.509 | 0.452 | 0.298 | 0.346 | 0.323 | 0.351 | 0.339 | 0.356 | 0.334 | 0.352 | 0.320 | 0.355 | 0.308 | 0.344 | 0.315 | 0.350 | 0.320 | 0.353 |
| Weather | 0.247 | 0.287 | 0.248 | 0.287 | 0.279 | 0.309 | 0.318 | 0.334 | 0.292 | 0.312 | 0.267 | 0.306 | 0.312 | 0.295 | 0.477 | 0.289 | 0.298 | 0.302 | 0.283 | 0.295 | 0.292 | 0.297 | - | - |
| Electricity | 0.188 | 0.285 | 0.196 | 0.295 | - | - | - | - | 0.297 | 0.375 | 0.243 | 0.329 | 0.207 | 0.296 | 0.224 | 0.309 | 0.246 | 0.312 | 0.336 | 0.329 | 0.326 | 0.328 | - | - |
| Traffic | 0.458 | 0.330 | 0.466 | 0.344 | - | - | - | - | 0.613 | 0.407 | - | - | - | - | - | - | 0.614 | 0.420 | 0.603 | 0.413 | 0.600 | 0.411 | - | - |
| 1$^{st}$ Count | 10 | | 3 | | 0 | | 0 | | 0 | | 0 | | 0 | | 0 | | 0 | | 0 | | 0 | | 0 | |

*Table 2.* Zero-shot performance evaluation on datasets provided by GluonTS (Alexandrov et al., 2020). A lower NRMSE/SMAPE indicates a better result. The best results are in **red** and the second best are **blue**. 'B': Base, 'L': Large.

| Datasets | Metrics | Olivia | SEMPO | Time-MoE$_B$ | Time-MoE$_L$ |
|---|---|---|---|---|---|
| solar_nips | NRMSE | 1.443 | 1.633 | 2.127 | 2.093 |
| | SMAPE | 1.447 | 1.460 | 1.930 | 1.821 |
| elecdemand | NRMSE | 0.066 | 0.097 | 0.250 | 0.284 |
| | SMAPE | 0.052 | 0.075 | 0.235 | 0.281 |
| fred_md | NRMSE | 2.575 | 2.495 | 4.330 | 4.330 |
| | SMAPE | 0.296 | 0.350 | 1.727 | 1.711 |
| wiki2000_nips | NRMSE | 4.760 | 4.886 | 4.937 | 4.937 |
| | SMAPE | 0.392 | 0.470 | 1.999 | 1.998 |
| m4_hourly | NRMSE | 0.179 | 0.250 | 5.819 | 5.819 |
| | SMAPE | 0.136 | 0.171 | 1.862 | 1.767 |
| oikolab_weather | NRMSE | 0.023 | 0.018 | 2.816 | 2.816 |
| | SMAPE | 0.340 | 0.360 | 1.117 | 1.294 |

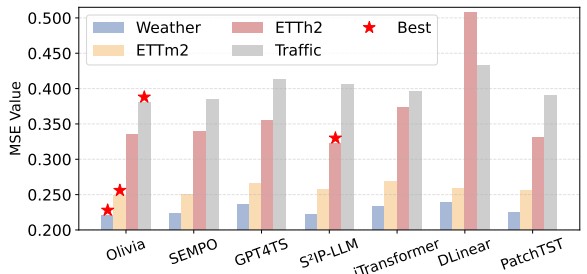

*Figure 3.* Full-shot results on the ETTh2, ETTm2, Weather and Traffic datasets. There ported results are averaged across all prediction lengths. Full details in Appendix G.2.

such as Electricity and Traffic, and substantially outperforms the large-scale Time-MoE family, reducing MSE by an average of 26.3% and MAE by 14.6% on the reported datasets. Similar trends are observed on datasets provided by GluonTS (see Table 2). Compared with SEMPO, Olivia achieves an NRMSE reduction of 11.6% on solar_nips and more pronounced improvements on elecdemand, with NRMSE and SMAPE reductions of 32.0% and 30.7%, respectively. When compared with the Time-MoE family, Olivia achieves an average reduction of over 86% in both NRMSE and SMAPE on the elecdemand and m4_hourly datasets. More results are provided in the Appendix G.1.

Beyond the zero-shot setting, Olivia also demonstrates strong performance under the full-shot setting on ETTh2,

ETTm2, Weather, and Traffic (see Figure 3), achieving lower average MSE and MAE than both pretrained foundation models and task-specific baselines in most cases. Overall, these results indicate that Olivia delivers robust performance across both zero-shot and full-shot settings, consistently outperforming lightweight and large-scale foundation models on challenging real-world time series datasets.

### 5.3. Model Analysis

#### 5.3.1. STUDIES OF HARMONIZER

We evaluate the effectiveness of the proposed Harmonizer by training two model variants from scratch, with and without the Harmonizer, and assessing their zero-shot forecasting performance. As shown in Figure 5, the model with the Harmonizer consistently achieves lower MSE across all prediction horizons on both ETTh2 and ETTm2, demonstrating robust gains under long-horizon zero-shot forecasting. We additionally integrate Harmonizer into the existing foundation model SEMPO, as shown in Figure 6, where the consistent MSE reductions highlight its effectiveness beyond the original architecture.

To investigate the mechanism behind these improvements, we analyze the impact of the Harmonizer in Figure 4. Although the raw diverse pretraining datasets exhibit highly heterogeneous and domain-specific patterns, the Aligner reorganizes these dependencies into consistent structured representations across datasets. Moreover, these aligned structures persist in unseen datasets (the right part in Figure 4), indicating that the Harmonizer captures transferable temporal regularities. By enforcing this spectral consistency, the model promotes domain-invariant representations that mitigate the accumulation of correlation mismatches.

#### 5.3.2. STUDIES OF HARMONICATTENTION

We further study the effectiveness of the proposed HarmonicAttention by replacing it with alternative attention mechanisms, including Full Attention (Vaswani et al., 2017), Linear Attention (Katharopoulos et al., 2020), and Nyström Attention (Xiong et al., 2021), while keeping all other components unchanged. As shown in Table 3, replacing HarmonicAttention with Full Attention or its approximations

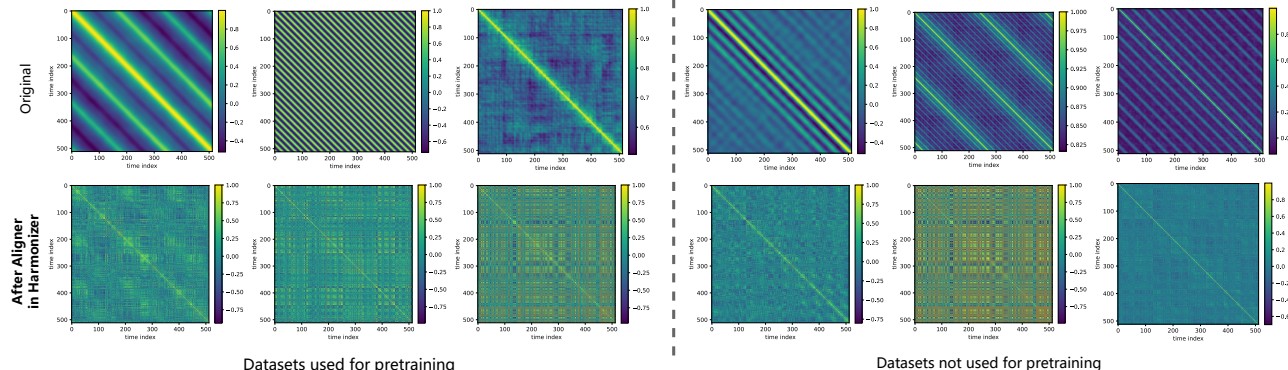

Datasets used for pretraining | Datasets not used for pretraining

*Figure 4.* Visualization of the second-order temporal correlation comparison between original data and those processed by Aligner. This figure illustrates the temporal correlation of raw time series from diverse pretraining datasets spanning multiple domains (**Left**), together with the corresponding correlation patterns after processing with the Aligner in the Harmonizer. Comparable structures are further observed on datasets not used during pretraining (**Right**), suggesting that the learned alignment generalizes beyond the training data.

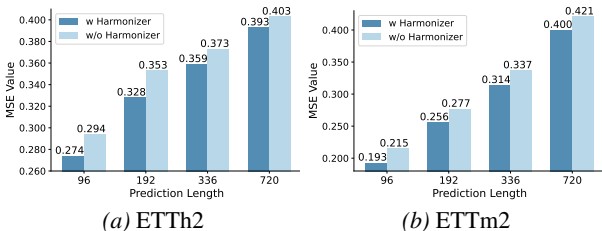

*(a) ETTh2*      *(b) ETTm2*

*Figure 5.* Ablations of Harmonizer under the zero-shot scenario.

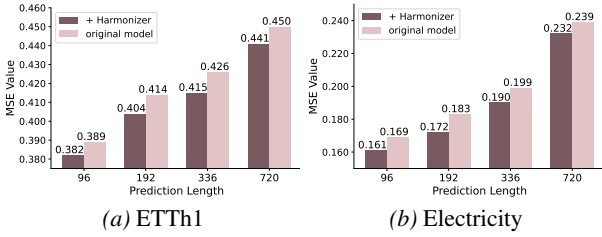

*(a) ETTh1*      *(b) Electricity*

*Figure 6.* Effectiveness of the proposed Harmonizer on SEMPO. Both the original SEMPO and the Harmonizer-enhanced SEMPO are trained from scratch and evaluated under the zero-shot setting.

consistently leads to degraded performance across ETTh1, Electricity, and Weather. This suggests that the observed gains are not merely due to attention expressiveness, but rather stem from the structured low-dimensional interaction pattern induced by HarmonicAttention, which is better matched to the PSD-consistent temporal representations produced by the Harmonizer. By operating on compact harmonic tokens instead of dense token-wise interactions, HarmonicAttention more effectively captures global temporal dependencies while suppressing spurious correlations, leading to improved robustness across diverse datasets.

### 5.3.3. EFFICIENCY ANALYSIS

Table 4 evaluates the efficiency of Olivia on the ETTh1 dataset in a zero-shot setting. It highlights that Olivia surpasses other foundation models, regardless of model scale, while maintaining the most compact parameter footprint (5.1M). While our approach exhibits slightly higher latency than the lightweight SEMPO, our effective results outper-

*Table 3.* Ablations on HarmonicAttention, where we replace HarmonicAttention with other attention mechanisms. The reported results are averaged across all prediction lengths.

| Datasets | ETTh1 | | Electricity | | Weather | |
|---|---|---|---|---|---|---|
| Metrics | MSE | MAE | MSE | MAE | MSE | MAE |
| **HarmonicAttention** | **0.399** | **0.421** | **0.188** | **0.285** | **0.247** | **0.287** |
| Full Attention | 0.472 | 0.470 | 0.204 | 0.300 | 0.268 | 0.305 |
| Linear Attention | 0.412 | 0.428 | 0.194 | 0.292 | 0.257 | 0.257 |
| Nyström Attention | 0.488 | 0.482 | 0.216 | 0.308 | 0.266 | 0.304 |

*Table 4.* Efficiency comparison on ETTh1 dataset. The reported MSE and inference time values are averaged over all prediction horizons. More results are provided in the Appendix G.6.

| Model | MSE | Inference Time (s) | Model Size (M) |
|---|---|---|---|
| Olivia | **0.399** | 43.051 | **5.1** |
| SEMPO | 0.410 | **8.162** | 6.5 |
| Time-MoE$_B$ | 0.445 | 4050.209 | 113 |
| Timer | 0.451 | 103.822 | 67.4 |
| Moirai$_B$ | 0.433 | 60.455 | 91 |
| Sundial | 0.464 | 67.901 | 128 |

form its performance. Nevertheless, Olivia achieves lower inference time than the Moirai family and remains orders of magnitude faster than Time-MoE. These results demonstrate that Olivia strikes a favorable balance between architectural rigor and practical speed, delivering state-of-the-art accuracy with significantly fewer resources.

## 6. Conclusion and Limitation

We propose Olivia, a time series foundation model that addresses cross-domain heterogeneity through PSD-consistent temporal representations. By combining the Harmonizer for reorganizing second-order temporal dependencies with HarmonicAttention for efficient global modeling, Olivia captures transferable temporal structures across diverse domains. Extensive experiments demonstrate that Olivia achieves strong performance across multiple forecasting benchmarks, particularly on challenging real-world datasets, while maintaining a favorable efficiency–accuracy trade-off.

These results underscore the value of structurally harmonizing temporal dynamics for building scalable and generalizable time series foundation models.

Nevertheless, a limitation of Olivia is the additional inference overhead compared with lightweight architectures. This overhead mainly arises from the Harmonizer, where the orthogonal transformation matrix $\mathbf{Q}$ is constructed through sequential Householder reflections. Developing more efficient orthogonal parameterization strategies remains an important direction for future work.

## Acknowledgements

This work was partially supported by the National Natural Science Foundation of China under Grant No. 62402531, the Shaanxi Fundamental Science Research Project for Mathematics and Physics under Grant No. 25JSQ053, and the China Postdoctoral Science Foundation under Grant No. 2024M753010.

## Impact Statement

This work aims to advance time series foundation models by improving representation learning and generalization across heterogeneous datasets. The proposed methods are general-purpose and are not specifically developed for high-risk or sensitive applications.

As with many advances in machine learning, the techniques presented in this paper may be incorporated into downstream systems across a wide range of domains. We do not identify any direct or immediate negative societal impacts arising from this work. Considerations related to responsible use and deployment depend on the specific application context and are the responsibility of practitioners.

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

# A. Theoretical Supplement

## A.1. Proof of Proposition 1

*Proposition* 1. Let $\mathcal{U} \subset \mathbb{R}^T$ be an $r$-dimensional subspace that is invariant under the second-order moment matrices of all datasets $\mathcal{D}$ in the pretraining collection $\mathcal{T}_{\text{pretrain}}$, that is, $\boldsymbol{\Sigma}_{\mathcal{D}}^X \mathcal{U} \subseteq \mathcal{U}$ where $\boldsymbol{\Sigma}_{\mathcal{D}}^X \triangleq \mathbb{E}[(X_{\mathcal{D}})^\top X_{\mathcal{D}}]$. Then, there exists a shared orthogonal matrix $\mathbf{Q} \in \mathbb{R}^{T \times T}$ whose first $r$ columns span $\mathcal{U}$, such that the transformed signal $\mathcal{X}_{\mathcal{D}} \triangleq X_{\mathcal{D}} \mathbf{Q}^\top$ yields a block-diagonal moment matrix:

$$\boldsymbol{\Sigma}_{\mathcal{D}}^{\mathcal{X}} \triangleq \mathbb{E}\left[(\mathcal{X}_{\mathcal{D}})^\top \mathcal{X}_{\mathcal{D}}\right] = \mathbf{Q} \boldsymbol{\Sigma}_{\mathcal{D}}^X \mathbf{Q}^\top = \text{diag}\left(\boldsymbol{\Lambda}_{\mathcal{D}}, \boldsymbol{\Phi}_{\mathcal{D}}\right), \tag{12}$$

where $\boldsymbol{\Lambda}_{\mathcal{D}} \in \mathbb{R}^{r \times r}$ characterizes the shared subspace dynamics and $\boldsymbol{\Phi}_{\mathcal{D}} \in \mathbb{R}^{(T-r) \times (T-r)}$ captures dataset-specific variations in the orthogonal complement $\mathcal{U}^\perp$.

*Proof.* We first note that for each dataset $\mathcal{D}$, the matrix $\boldsymbol{\Sigma}_{\mathcal{D}}^X = \mathbb{E}\left[(X_{\mathcal{D}})^\top X_{\mathcal{D}}\right] \in \mathbb{R}^{T \times T}$ is symmetric and positive semidefinite. Symmetry follows from $\left((X_{\mathcal{D}})^\top X_{\mathcal{D}}\right)^\top = (X_{\mathcal{D}})^\top X_{\mathcal{D}}$, and for any $v \in \mathbb{R}^T$, $v^\top \boldsymbol{\Sigma}_{\mathcal{D}}^X v = \mathbb{E}\left[v^\top (X_{\mathcal{D}})^\top X_{\mathcal{D}} v\right] = \mathbb{E}\left[\|X_{\mathcal{D}} v\|_2^2\right] \geq 0$.

**Step 1: Invariance of the orthogonal complement** Let $\mathcal{U}^\perp$ denote the orthogonal complement of $\mathcal{U}$ in $\mathbb{R}^T$. We claim that for every $\mathcal{D} \in \mathcal{T}_{\text{pretrain}}$,

$$\boldsymbol{\Sigma}_{\mathcal{D}}^X \mathcal{U}^\perp \subseteq \mathcal{U}^\perp. \tag{13}$$

To prove this, take any $y \in \mathcal{U}^\perp$ and any $x \in \mathcal{U}$. Using symmetry of $\boldsymbol{\Sigma}_{\mathcal{D}}^X$, we have

$$\left\langle \boldsymbol{\Sigma}_{\mathcal{D}}^X y, x \right\rangle = y^\top \boldsymbol{\Sigma}_{\mathcal{D}}^X x. \tag{14}$$

Since $\mathcal{U}$ is invariant, $\Sigma_{\mathcal{D}}^X x \in \mathcal{U}$. Because $y \in \mathcal{U}^\perp$, it holds that $y^\top u = 0$ for all $u \in \mathcal{U}$, hence $y^\top \boldsymbol{\Sigma}_{\mathcal{D}}^X x = 0$. Therefore $\left\langle \boldsymbol{\Sigma}_{\mathcal{D}}^X y, x \right\rangle = 0$ for all $x \in \mathcal{U}$, which implies $\Sigma_{\mathcal{D}}^X y \in \mathcal{U}^\perp$. This proves $\boldsymbol{\Sigma}_{\mathcal{D}}^X \mathcal{U}^\perp \subseteq \mathcal{U}^\perp$.

Consequently, for every $\mathcal{D}$, the space $\mathbb{R}^T$ decomposes as an orthogonal direct sum

$$\mathbb{R}^T = \mathcal{U} \oplus \mathcal{U}^\perp, \tag{15}$$

and both $\mathcal{U}$ and $\mathcal{U}^\perp$ are invariant under $\boldsymbol{\Sigma}_{\mathcal{D}}^X$.

**Step 2: Constructing a shared orthogonal basis** Choose an orthonormal basis $\{q_1, \ldots, q_r\}$ of $\mathcal{U}$, and an orthonormal basis $\{q_{r+1}, \ldots, q_T\}$ of $\mathcal{U}^\perp$. Define

$$\mathbf{Q} \triangleq \begin{bmatrix} q_1 & \cdots & q_r & q_{r+1} & \cdots & q_T \end{bmatrix} \in \mathbb{R}^{T \times T}. \tag{16}$$

Then $\mathbf{Q}^\top \mathbf{Q} = \mathbf{I}$, i.e., $\mathbf{Q}$ is orthogonal, and by construction its first $r$ columns span $\mathcal{U}$. Importantly, $\mathbf{Q}$ is determined solely by the shared subspace $\mathcal{U}$ (and $\mathcal{U}^\perp$), hence it is shared across all datasets $\mathcal{D} \in \mathcal{T}_{\text{pretrain}}$.

**Step 3: Block-diagonalization** Partition $\mathbf{Q}$ as $\mathbf{Q} = [\mathbf{Q}_{\mathcal{U}} \mathbf{Q}_\perp]$, where $\mathbf{Q}_{\mathcal{U}} \in \mathbb{R}^{T \times r}$ spans $\mathcal{U}$ and $\mathbf{Q}_\perp \in \mathbb{R}^{T \times (T-r)}$ spans $\mathcal{U}^\perp$. Consider

$$\mathbf{Q} \boldsymbol{\Sigma}_{\mathcal{D}}^X \mathbf{Q}^\top = \begin{pmatrix} \mathbf{Q}_{\mathcal{U}}^\top \boldsymbol{\Sigma}_{\mathcal{D}}^X \mathbf{Q}_{\mathcal{U}} & \mathbf{Q}_{\mathcal{U}}^\top \boldsymbol{\Sigma}_{\mathcal{D}}^X \mathbf{Q}_\perp \\ \mathbf{Q}_\perp^\top \boldsymbol{\Sigma}_{\mathcal{D}}^X \mathbf{Q}_{\mathcal{U}} & \mathbf{Q}_\perp^\top \boldsymbol{\Sigma}_{\mathcal{D}}^X \mathbf{Q}_\perp \end{pmatrix}. \tag{17}$$

We show the off-diagonal blocks are zero. Since $\boldsymbol{\Sigma}_{\mathcal{D}}^X \mathcal{U} \subseteq \mathcal{U}$, we have $\boldsymbol{\Sigma}_{\mathcal{D}}^X \mathbf{Q}_{\mathcal{U}} \subseteq \mathcal{U}$. Each column of $\mathbf{Q}_\perp$ lies in $\mathcal{U}^\perp$, hence

$$\mathbf{Q}_\perp^\top \boldsymbol{\Sigma}_{\mathcal{D}}^X \mathbf{Q}_{\mathcal{U}} = \mathbf{0}. \tag{18}$$

By symmetry of $\boldsymbol{\Sigma}_{\mathcal{D}}^X$, the other off-diagonal block is the transpose:

$$\mathbf{Q}_{\mathcal{U}}^\top \boldsymbol{\Sigma}_{\mathcal{D}}^X \mathbf{Q}_\perp = \left(\mathbf{Q}_\perp^\top \boldsymbol{\Sigma}_{\mathcal{D}}^X \mathbf{Q}_{\mathcal{U}}\right)^\top = \mathbf{0}. \tag{19}$$

Therefore,

$$\mathbf{Q}\boldsymbol{\Sigma}_{\mathcal{D}}^X \mathbf{Q}^\top = \begin{pmatrix} \boldsymbol{\Lambda}_{\mathcal{D}} & \mathbf{0} \\ \mathbf{0} & \boldsymbol{\Phi}_{\mathcal{D}} \end{pmatrix} = \mathrm{diag}\left(\boldsymbol{\Lambda}_{\mathcal{D}}, \boldsymbol{\Phi}_{\mathcal{D}}\right), \tag{20}$$

Finally, define the transformed signal $\mathcal{X}_{\mathcal{D}} \triangleq X_{\mathcal{D}}\mathbf{Q}^\top$. By direct expansion,

$$\boldsymbol{\Sigma}_{\mathcal{D}}^{\mathcal{X}} \triangleq \mathbb{E}\left[(\mathcal{X}_{\mathcal{D}})^\top \mathcal{X}_{\mathcal{D}}\right] = \mathbb{E}\left[(\mathbf{Q}X_{\mathcal{D}}^\top)(X_{\mathcal{D}}\mathbf{Q}^\top)\right] = \mathbf{Q}\mathbb{E}\left[X_{\mathcal{D}}^\top X_{\mathcal{D}}\right]\mathbf{Q}^\top = \mathbf{Q}\boldsymbol{\Sigma}_{\mathcal{D}}^X \mathbf{Q}^\top. \tag{21}$$

Combining with the block-diagonalization above yields

$$\boldsymbol{\Sigma}_{\mathcal{D}}^{\mathcal{X}} = \mathrm{diag}\left(\boldsymbol{\Lambda}_{\mathcal{D}}, \boldsymbol{\Phi}_{\mathcal{D}}\right), \tag{22}$$

which proves the claim. $\qquad\square$

## A.2. Proof of Proposition 2

*Proposition* 2. Let $\mathcal{X} \in \mathbb{R}^{1\times T}$ denote an aligned temporal signal with finite second moments, and let $\boldsymbol{\Sigma}_{\mathcal{X}} \triangleq \mathbb{E}\left[\mathcal{X}^\top \mathcal{X}\right] \in \mathbb{R}^{T\times T}$ denote its second-order moment matrix. Assume that $\boldsymbol{\Sigma}_{\mathcal{X}}$ admits a block-diagonal decomposition $\boldsymbol{\Sigma}_{\mathcal{X}} = \mathrm{diag}(\boldsymbol{\Lambda}, \boldsymbol{\Phi}), \boldsymbol{\Lambda} \in \mathbb{R}^{r\times r}, \boldsymbol{\Phi} \in \mathbb{R}^{(T-r)\times(T-r)}$, with $r \ll T$. Let token representations be generated by linear temporal operators $z_\ell = M_\ell \mathcal{X}^\top, M_\ell \in \mathbb{R}^{d\times T}, \ell = 1,\ldots,L$, and stack them as $Z = [z_1^\top; \ldots; z_L^\top] \in \mathbb{R}^{L\times d}$. Define the token Gram matrix $\mathbf{C}_Z \triangleq \mathbb{E}[ZZ^\top] \in \mathbb{R}^{L\times L}, [\mathbf{C}_Z]_{\ell,\ell'} = \mathbb{E}\left[z_\ell^\top z_{\ell'}\right]$. Partition each temporal operator $M_\ell$ conformably with the block structure of $\boldsymbol{\Sigma}_{\mathcal{X}}$ as $M_\ell = \begin{bmatrix} A_\ell & B_\ell \end{bmatrix}, A_\ell \in \mathbb{R}^{d\times r}, B_\ell \in \mathbb{R}^{d\times(T-r)}$. Then, for all $\ell, \ell'$, the token Gram entries admit the decomposition $[\mathbf{C}_Z]_{\ell,\ell'} = \mathrm{tr}\left(A_\ell \boldsymbol{\Lambda} A_{\ell'}^\top\right) + \mathrm{tr}\left(B_\ell \boldsymbol{\Phi} B_{\ell'}^\top\right)$. Moreover, the truncated Gram matrix defined by $[\mathbf{C}_Z^{(r)}]_{\ell,\ell'} \triangleq \mathrm{tr}\left(A_\ell \boldsymbol{\Lambda} A_{\ell'}^\top\right)$ is positive semidefinite and satisfies

$$\mathrm{rank}(\mathbf{C}_Z^{(r)}) \leq dr.$$

In addition, the approximation error is bounded as

$$\left|[\mathbf{C}_Z]_{\ell,\ell'} - [\mathbf{C}_Z^{(r)}]_{\ell,\ell'}\right| \leq \|\boldsymbol{\Phi}\|_2 \|B_\ell\|_F \|B_{\ell'}\|_F. \tag{23}$$

*Proof.* We first rewrite each Gram entry in a form that explicitly involves the second-order moment matrix $\boldsymbol{\Sigma}_{\mathcal{X}}$. Recall that $z_\ell = M_\ell \mathcal{X}^\top \in \mathbb{R}^d$ and $[\mathbf{C}_Z]_{\ell,\ell'} = \mathbb{E}\left[z_\ell^\top z_{\ell'}\right]$. For any $\ell, \ell'$, we have

$$z_\ell^\top z_{\ell'} = (M_\ell \mathcal{X}^\top)^\top (M_{\ell'} \mathcal{X}^\top) = \mathcal{X}M_\ell^\top M_{\ell'} \mathcal{X}^\top. \tag{24}$$

Using the identity $u^\top K u = \mathrm{tr}(Kuu^\top)$ with $u = \mathcal{X}^\top$, and taking expectation, we obtain

$$\begin{aligned}[\mathbf{C}_Z]_{\ell,\ell'} &= \mathbb{E}\left[\mathcal{X}M_\ell^\top M_{\ell'} \mathcal{X}^\top\right] = \mathrm{tr}\left(M_\ell^\top M_{\ell'}\, \mathbb{E}[\mathcal{X}^\top \mathcal{X}]\right) \\ &= \mathrm{tr}\left(M_\ell^\top M_{\ell'} \boldsymbol{\Sigma}_{\mathcal{X}}\right) = \mathrm{tr}\left(M_\ell \boldsymbol{\Sigma}_{\mathcal{X}} M_{\ell'}^\top\right), \end{aligned} \tag{25}$$

where the last equality follows from cyclicity of trace.

**Step 1: Decomposition of Gram entries.** Assume $\boldsymbol{\Sigma}_{\mathcal{X}} = \mathrm{diag}(\boldsymbol{\Lambda}, \boldsymbol{\Phi})$ with $\boldsymbol{\Lambda} \in \mathbb{R}^{r\times r}$ and $\boldsymbol{\Phi} \in \mathbb{R}^{(T-r)\times(T-r)}$. Partition each $M_\ell \in \mathbb{R}^{d\times T}$ conformably as $M_\ell = [A_\ell\ B_\ell]$ with $A_\ell \in \mathbb{R}^{d\times r}$ and $B_\ell \in \mathbb{R}^{d\times(T-r)}$. Then

$$M_\ell \boldsymbol{\Sigma}_{\mathcal{X}} M_{\ell'}^\top = \begin{bmatrix} A_\ell & B_\ell \end{bmatrix} \begin{bmatrix} \boldsymbol{\Lambda} & 0 \\ 0 & \boldsymbol{\Phi} \end{bmatrix} \begin{bmatrix} A_{\ell'}^\top \\ B_{\ell'}^\top \end{bmatrix} = A_\ell \boldsymbol{\Lambda} A_{\ell'}^\top + B_\ell \boldsymbol{\Phi} B_{\ell'}^\top. \tag{26}$$

Combining (25) and (26) yields, for all $\ell, \ell'$,

$$[\mathbf{C}_Z]_{\ell,\ell'} = \mathrm{tr}\left(A_\ell \boldsymbol{\Lambda} A_{\ell'}^\top\right) + \mathrm{tr}\left(B_\ell \boldsymbol{\Phi} B_{\ell'}^\top\right), \tag{27}$$

which proves the stated decomposition.

**Step 2: Positive semidefiniteness and rank bound of the truncated Gram matrix.** Define the truncated Gram matrix $\mathbf{C}_Z^{(r)} \in \mathbb{R}^{L \times L}$ by

$$[\mathbf{C}_Z^{(r)}]_{\ell,\ell'} \triangleq \mathrm{tr}\big(A_\ell \mathbf{\Lambda} A_{\ell'}^\top\big).$$

Since $\mathbf{\Sigma}_\mathcal{X}$ is a second-order moment matrix, it is symmetric positive semidefinite, and hence its principal block $\mathbf{\Lambda}$ is also symmetric positive semidefinite. Let $\mathbf{\Lambda}^{1/2}$ denote its symmetric positive semidefinite square root. Using cyclicity of trace, we can rewrite each entry as

$$[\mathbf{C}_Z^{(r)}]_{\ell,\ell'} = \mathrm{tr}\big(A_\ell \mathbf{\Lambda} A_{\ell'}^\top\big) = \mathrm{tr}\Big(A_\ell \mathbf{\Lambda}^{1/2}(\mathbf{\Lambda}^{1/2} A_{\ell'}^\top)\Big) = \Big\langle A_\ell \mathbf{\Lambda}^{1/2}, A_{\ell'} \mathbf{\Lambda}^{1/2} \Big\rangle_F, \tag{28}$$

where $\langle U, V \rangle_F \triangleq \mathrm{tr}(UV^\top)$ denotes the Frobenius inner product. Therefore, $\mathbf{C}_Z^{(r)}$ is a Gram matrix of the set $\{A_\ell \mathbf{\Lambda}^{1/2}\}_{\ell=1}^L$ under $\langle \cdot, \cdot \rangle_F$, which implies that $\mathbf{C}_Z^{(r)}$ is positive semidefinite.

Moreover, define vectors $g_\ell \triangleq \mathrm{vec}(A_\ell \mathbf{\Lambda}^{1/2}) \in \mathbb{R}^{dr}$. Then (28) becomes $[\mathbf{C}_Z^{(r)}]_{\ell,\ell'} = g_\ell^\top g_{\ell'}$. Let $G \triangleq [g_1, \ldots, g_L]^\top \in \mathbb{R}^{L \times dr}$. We can write $\mathbf{C}_Z^{(r)} = GG^\top$, hence

$$\mathrm{rank}(\mathbf{C}_Z^{(r)}) \leq \mathrm{rank}(G) \leq dr.$$

**Step 3: Entrywise approximation error bound.** By the decomposition in Step 1, we have

$$[\mathbf{C}_Z]_{\ell,\ell'} - [\mathbf{C}_Z^{(r)}]_{\ell,\ell'} = \mathrm{tr}\big(B_\ell \mathbf{\Phi} B_{\ell'}^\top\big).$$

Since $\mathbf{\Phi}$ is symmetric positive semidefinite, let $\mathbf{\Phi}^{1/2}$ denote its square root. Then

$$\mathrm{tr}\big(B_\ell \mathbf{\Phi} B_{\ell'}^\top\big) = \mathrm{tr}\Big(B_\ell \mathbf{\Phi}^{1/2}(B_{\ell'} \mathbf{\Phi}^{1/2})^\top\Big) = \Big\langle B_\ell \mathbf{\Phi}^{1/2}, B_{\ell'} \mathbf{\Phi}^{1/2} \Big\rangle_F. \tag{29}$$

Applying Cauchy–Schwarz under the Frobenius inner product yields

$$\begin{aligned}\big|\mathrm{tr}\big(B_\ell \mathbf{\Phi} B_{\ell'}^\top\big)\big| &\leq \|B_\ell \mathbf{\Phi}^{1/2}\|_F \|B_{\ell'} \mathbf{\Phi}^{1/2}\|_F \\ &\leq \|\mathbf{\Phi}^{1/2}\|_2^2 \|B_\ell\|_F \|B_{\ell'}\|_F = \|\mathbf{\Phi}\|_2 \|B_\ell\|_F \|B_{\ell'}\|_F,\end{aligned} \tag{30}$$

where we used the submultiplicativity $\|UV\|_F \leq \|U\|_F\|V\|_2$ and $\|\mathbf{\Phi}^{1/2}\|_2^2 = \|\mathbf{\Phi}\|_2$. This proves the claimed entrywise error bound:

$$\big|[\mathbf{C}_Z]_{\ell,\ell'} - [\mathbf{C}_Z^{(r)}]_{\ell,\ell'}\big| \leq \|\mathbf{\Phi}\|_2 \|B_\ell\|_F \|B_{\ell'}\|_F.$$

$\square$

## B. Details of Pretraining, Tuning, and Inference Stages

**Pretraining stage.** As illustrated in Figure 7, Olivia is pretrained under a reconstruction objective, enabling it to learn domain-agnostic temporal representations from large-scale heterogeneous datasets. Given an input time series $X_{1:T}$, the Aligner in Harmonizer first performs a learnable orthogonal temporal reparameterization, projecting the input into a PSD-harmonized temporal space. The aligned signal is then patchified and projected into token representations, which are processed by the Harmonic Encoder. A reconstruction decoder and head are applied to recover the aligned temporal signal, which is subsequently mapped back to the original temporal domain by the Restorer in Harmonizer. All components in this stage, including the Harmonizer, encoder, decoder, and reconstruction head, are jointly optimized, enabling the model to learn a shared temporal organization that reduces cross-domain spectral discrepancies. The pretraining objective is defined as the mean squared reconstruction error:

$$\mathcal{L}_{\text{pretraining}} = \frac{1}{T}\left\|X_{1:T} - \hat{X}_{1:T}\right\|_2^2. \tag{31}$$

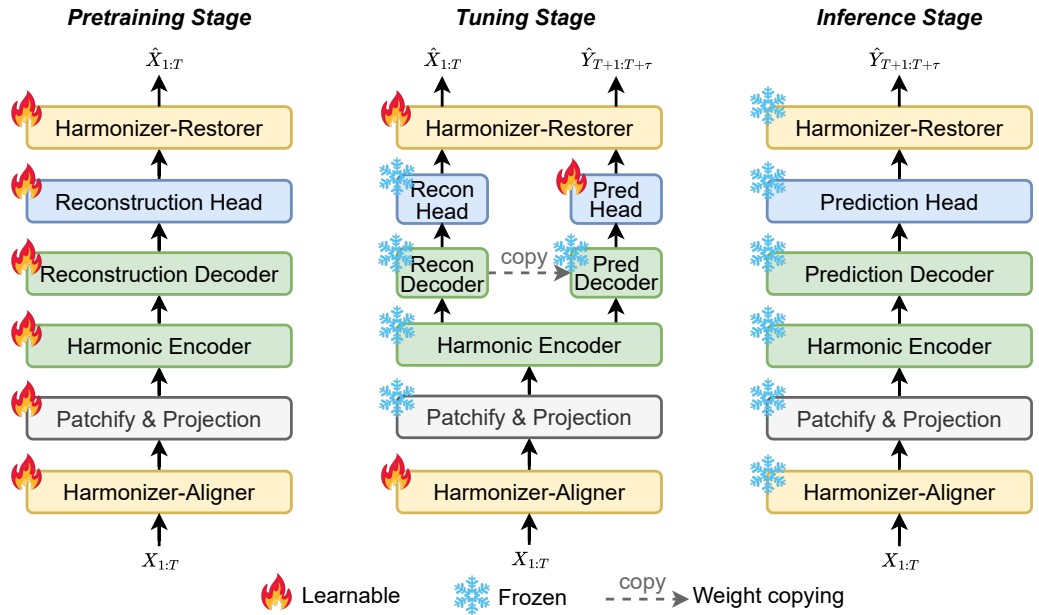

*Figure 7.* Overview of Olivia's training and inference pipeline across the pretraining, tuning, and inference stages.

**Tuning stage.** During the tuning stage, Olivia adapts the pretrained representations to task-oriented forecasting while retaining the PSD-consistent temporal organization learned during pretraining. As illustrated in Figure 7, the Aligner in Harmonizer remains learnable, allowing the temporal reparameterization to be further refined for downstream forecasting tasks. In contrast, the patch embedding layers and the Harmonic Encoder are frozen to preserve the shared temporal representations learned from large-scale pretraining. To enable a smooth transition from reconstruction-based pretraining to forecasting, the reconstruction decoder learned during pretraining is retained, and its weights are copied to the prediction decoder. A prediction head is attached to generate future forecasts, while the reconstruction head is preserved to provide an auxiliary reconstruction signal. During tuning, only the Aligner and Restorer in Harmonizer, together with the prediction head, are updated, whereas the remaining components are frozen to prevent disruption of the learned representation space. To enhance model generalization across different forecasting horizons, the tuning objective jointly optimizes forecasting accuracy and representation stability. Let $\mathcal{T} = \{\tau_1, \tau_2, \ldots, \tau_H\}$ denote a collection of forecasting horizons, where each $\tau_h$ specifies the number of future steps to be predicted at the $h$-th horizon. Given the corresponding ground-truth sequences $Y_{T+1:T+\tau_h}$ for $h = 1, 2, \ldots, H$, the tuning loss is defined as

$$\mathcal{L}_{\text{tuning}} = \sum_{\tau \in \mathcal{T}} \left\| Y_{T+1:T+\tau} - \hat{Y}_{T+1:T+\tau} \right\|_2^2 + \left\| X_{1:T} - \hat{X}_{1:T} \right\|_2^2. \tag{32}$$

**Inference Stage.** During inference, Olivia performs forecasting using the fully trained model without updating any parameters. As illustrated in Figure 7, all components remain frozen during inference, ensuring stable and efficient prediction while fully leveraging the temporal organization learned during pretraining and tuning.

## C. Empirical Analysis of Temporal Patterns and Power Spectral Density Discrepancies Across Domains

### C.1. Temporal Pattern Visualization

To visualize temporal patterns across domains, we randomly sample time series segments of length $T = 512$ from the pretraining datasets and represent each sequence using a one-dimensional heatmap. This visualization enables clear inspection of domain-specific temporal structures, such as periodicity and rhythm. Following the data preprocessing pipeline used during model training (He et al., 2025), we apply a StandardScaler to each sampled sequence before visualization. This step ensures consistency with the inputs seen by the model and avoids misleading differences caused purely by scale or offset variations. The resulting visualizations are shown in Figure 1(a). It is important to note that applying the StandardScaler

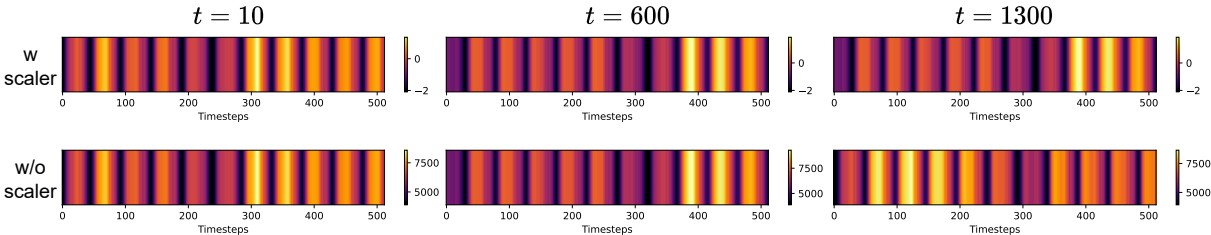

*Figure 8.* Visualization of temporal patterns with and without StandardScaler[1] on sequences sampled from the same dataset in the Energy domain. Time series segments of length $T = 512$ are extracted starting from different time indices $t_i \in \{10, 600, 1300\}$. The top row shows sequences after applying StandardScaler, while the bottom row shows the corresponding raw sequences without scaling. Although StandardScaler rescales the numerical values, the underlying temporal patterns and temporal correlations remain unchanged, indicating that normalization does not alter the intrinsic temporal structure of the time series.

does not alter the underlying temporal patterns or temporal correlations of the sequence, but only rescales the numerical values. As illustrated in Figure 8, temporal structures remain visually consistent before and after scaling, even at different time indices. Therefore, the observed differences in Figure 1(a) reflect genuine domain-specific temporal patterns rather than artifacts introduced by normalization.

### C.2. Empirical Estimation of Dataset-Level Dataset-Level Power Spectral Density

Given a domain dataset $\mathcal{D}$, we empirically estimate its dataset-level power spectral density (PSD) to characterize how spectral energy is distributed across frequencies at the dataset level.

**Window Sampling and Preprocessing.**  From the raw long time series in each domain dataset, we first construct a collection of fixed-length windows. Specifically, we randomly sample $N$ windows $\left\{x^{(n)}\right\}_{n=1}^{N}, x^{(n)} \in \mathbb{R}^T$, where $T = 512$ in all experiments and $N = 2000$ windows are sampled per domain dataset. Each window is treated as a univariate time series. To mitigate instance-level distributional variations within a domain, we apply instance-wise normalization to each window:

$$\tilde{x}^{(n)} = \frac{x^{(n)} - \mu^{(n)}}{\sigma^{(n)} + \varepsilon}, \tag{33}$$

where $\mu^{(n)}$ and $\sigma^{(n)}$ denote the mean and standard deviation of the $n$-th window along the temporal dimension, $\varepsilon$ is a small constant to prevent division by zero. This normalization ensures that each window is approximately zero-mean and unit-variance, while preserving its temporal correlation structure.

**Spectral Estimation via Periodogram.**  For each normalized window $\tilde{x}^{(n)}$, we estimate its power spectral density using the discrete Fourier transform (Brigham, 1988). The periodogram-based PSD (Welch, 2003) of $\tilde{x}^{(n)}$ is computed as:

$$S_{\tilde{x}^{(n)}}(\omega_k) = \frac{1}{T} \left| \sum_{t=0}^{T-1} \tilde{x}_t^{(n)} e^{-j\omega_k t} \right|^2, \tag{34}$$

where $\omega_k = \frac{2\pi k}{T}, k = 0, 1, \ldots, \lfloor T/2 \rfloor$.

**Dataset-Level PSD Aggregation.**  The normalized dataset-level PSD for domain dataset $\mathcal{D}$ is defined as the average of individual periodograms over all sampled windows, normalized to form a probability distribution over frequencies:

$$\mathcal{P}_{\mathcal{D}}(\omega_k) = \frac{\frac{1}{N} \sum_{n=1}^{N} S_{\tilde{x}^{(n)}}(\omega_k)}{\sum_k \frac{1}{N} \sum_{n=1}^{N} S_{\tilde{x}^{(n)}}(\omega_k)}. \tag{35}$$

This normalized dataset-level PSD $\mathcal{P}_{\mathcal{D}}$ forms a probability distribution over frequencies, capturing the relative allocation of spectral energy and serving as the basis for cross-domain spectral discrepancy measurement in Appendix C.3.

### C.3. Jensen–Shannon Divergence for Measuring Cross-Domain PSD Discrepancies

To quantify spectral discrepancies between different domains, we compute the Jensen–Shannon (JS) divergence (Lin, 2002; Endres & Schindelin, 2003) between pairs of dataset-level normalized PSD distributions. Given two domain datasets $\mathcal{D}_i$ and

$\mathcal{D}_j$, let $\mathcal{P}_{\mathcal{D}_i}(\omega_k)$ and $\mathcal{P}_{\mathcal{D}_j}(\omega_k)$ denote their normalized dataset-level PSDs estimated as described in Appendix C.2, where $\omega_k = 2\pi k / T$ and $k = 0, 1, \ldots, \lfloor T/2 \rfloor$. The pairwise JS divergence is defined as

$$\mathrm{JS}\left(\mathcal{P}_{\mathcal{D}_i}, \mathcal{P}_{\mathcal{D}_j}\right) = \frac{1}{2}\mathrm{KL}\left(\mathcal{P}_{\mathcal{D}_i} \| \mathcal{M}_{i,j}\right) + \frac{1}{2}\mathrm{KL}\left(\mathcal{P}_{\mathcal{D}_j} \| \mathcal{M}_{i,j}\right), \tag{36}$$

where $\mathcal{M}_{i,j} = \frac{1}{2}\left(\mathcal{P}_{\mathcal{D}_i} + \mathcal{P}_{\mathcal{D}_j}\right)$ denotes the mixture distribution, and $\mathrm{KL}(\cdot\|\cdot)$ is the Kullback–Leibler (KL) divergence (Kullback & Leibler, 1951). The KL divergence between two discrete distributions $P$ and $Q$, defined over the discrete frequency bins $\{\omega_k\}_{k=0}^{\lfloor T/2 \rfloor}$, is given by

$$\mathrm{KL}(P\|Q) = \sum_{k=0}^{\lfloor T/2 \rfloor} P\left(\omega_k\right) \log \frac{P\left(\omega_k\right)}{Q\left(\omega_k\right)}. \tag{37}$$

Compared to the KL divergence, the JS divergence enjoys several favorable properties in our setting: it is symmetric, always finite, and bounded within $[0, \log 2]$, which makes it more stable and interpretable for measuring discrepancies between empirical PSD distributions across domains. In Figure 1(b) and Table 5, we report this pairwise JS divergence for domain pairs before and after applying the Harmonizer, providing a quantitative measure of cross-domain spectral discrepancy and its reduction through the Aligner in Harmonizer.

## D. Further Discussion

### D.1. Distribution Shift vs Domain Shift

In time series forecasting, distribution shift typically refers to temporal changes in instance-level statistical properties, such as mean and variance, between the training and test splits of the same dataset, which are usually separated by time (Kim et al., 2021; Fan et al., 2023). Such shifts can degrade forecasting performance even when the underlying temporal structure remains unchanged, and are commonly addressed by instance-wise normalization techniques, such as Reversible Instance Normalization (RevIN) (Kim et al., 2021).

However, domain shift arises in a fundamentally different setting, where training and evaluation data originate from different datasets or domains with heterogeneous temporal dynamics. In this case, discrepancies are not limited to first-order statistics, but are often manifested in second-order temporal structures, such as correlation patterns and power spectral density (PSD) distributions. To explicitly disentangle these two types of shift and assess whether instance-level normalization is sufficient to mitigate cross-domain discrepancies, we conduct the analysis reported in Table 5.

Table 5 compares cross-domain Jensen–Shannon (JS) divergence of normalized PSD distributions before and after applying the Harmonizer, with and without RevIN. Without the Harmonizer, substantial JS divergences are observed across all domain pairs, indicating pronounced domain-level heterogeneity in spectral structure. Applying RevIN alone does not consistently reduce these divergences and, in several cases, even increases cross-domain JS divergence, suggesting that alleviating instance-level distribution shift is insufficient to reconcile structural discrepancies across datasets.

In contrast, the Harmonizer consistently and substantially reduces cross-domain JS divergence across all domain pairs, regardless of whether RevIN is applied. This empirical evidence highlights a clear distinction between distribution shift and domain shift: while RevIN effectively addresses within-dataset, instance-level distribution variations, it does not resolve cross-dataset discrepancies rooted in heterogeneous temporal correlation and spectral structures. By explicitly reorganizing temporal dynamics into a PSD-consistent representation space, the Harmonizer directly targets domain shift, achieving robust cross-domain alignment beyond the scope of instance-wise normalization.

## E. Experimental Details

### E.1. Baselines

We evaluate our model against a diverse and representative set of baselines spanning multiple modeling paradigms in time series forecasting. Specifically, the comparison includes three major categories. (i) Pretrained time series foundation models, which aim to learn transferable temporal representations from large-scale corpora and have recently demonstrated strong generalization capabilities across domains and tasks. This category includes SEMPO (He et al., 2025), Time-MoE (Shi

*Table 5.* Cross-domain JS divergence before and after Harmonizer, with and without RevIN.

| Cross-domain | Without RevIN | | With RevIN | |
|---|---|---|---|---|
| | Original | After Harmonizer | Original | After Harmonizer |
| Energy & Nature | 0.501872 | 0.074023 | 0.508711 | 0.081131 |
| Nature & Health | 0.518192 | 0.059645 | 0.533421 | 0.091080 |
| Health & Transport | 0.593384 | 0.035786 | 0.590491 | 0.077249 |
| Transport & Environment | 0.420261 | 0.026662 | 0.290352 | 0.051821 |
| Environment & Energy | 0.238952 | 0.013364 | 0.221595 | 0.026951 |

et al., 2024), Timer (Liu et al., 2024), Moirai (Woo et al., 2024), Chronos (Ansari et al., 2024), TimesFM (Das et al., 2024), Moment (Goswami et al., 2024), Sundial (Liu et al., 2025), ROSE (Wang et al., 2025b), and TTM (Ekambaram et al., 2024). Table 6 provides a comparison of representative time-series foundation models in terms of their architectural designs, model scales, training data sizes, and context lengths. (ii) LLM-based models, including Time-LLM (Jin et al., 2023), GPT4TS (Zhou et al., 2023), and $S^2$IP-LLM (Pan et al., 2024), which leverage large language models as sequence encoders or reasoning backbones for time series forecasting. These methods represent an emerging line of research that explores the transferability of linguistic priors to temporal data and serve as strong baselines for evaluating the effectiveness of foundation-model-based approaches. (iii) Task-specific forecasting models, such as iTransformer (Liu et al., 2023), DLinear (Zeng et al., 2023), PatchTST (Nie, 2022), TimesNet (Wu et al., 2022), Stationary (Liu et al., 2022b), and FEDformer (Zhou et al., 2022), which are explicitly designed and optimized for supervised forecasting tasks. These models capture diverse inductive biases, including linear decomposition, patch-based temporal modeling, frequency-domain attention, and stationarity-aware transformations, and remain strong competitors in controlled evaluation settings.

*Table 6.* Comparison of representative time-series foundation models.

| Model | Architecture | (Min) Model Size | (Max) Model Size | Token Level | Dataset Scale | Context Length | Source |
|---|---|---|---|---|---|---|---|
| Olivia | Enc-Dec | 5.1M | 9.6M | Patch | 67M | 512 / 1024 / 1536 | Ours |
| SEMPO | Enc-Dec | 6.5M | 9.9M | Patch | 83M | 512 / 1024 / 1536 | (He et al., 2025) |
| Sundial | Decoder | 32M | 444M | Patch | 1032B | ≤2880 | (Liu et al., 2025) |
| TTM | Mixer-style | 1M | 5M | Patch | 1B | 512 / 1024 / 1536 | (Ekambaram et al., 2024) |
| Time-MoE | Decoder | 113M | 2.4B | Point | 309B | ≤4096 | (Shi et al., 2024) |
| Timer | Decoder | 67M | 67M | Patch | 28B | ≤1440 | (Liu et al., 2024) |
| Moirai | Encoder | 14M | 311M | Patch | 27B / 231B | ≤5000 | (Woo et al., 2024) |
| Chronos | Enc-Dec | 46M | 710M | Point | 84B | ≤512 | (Ansari et al., 2024) |
| TimesFM | Decoder | 200M | 200M | Patch | 100B | ≤512 | (Das et al., 2024) |
| Moment | Encoder | 40M | 385M | Patch | 1.13B | 512 | (Goswami et al., 2024) |
| ROSE | Enc-Dec | 7.4M | 7.4M | Patch | 0.89B | 512 | (Wang et al., 2025b) |

### E.2. Datasets

**Pretraining Datasets.** Unified Time Series Dataset (UTSD) (Liu et al., 2024) is a curated collection of time series data assembled from a combination of publicly available online repositories and empirical data obtained from real-world machine operations. To ensure data quality and consistency, missing values are systematically handled using linear interpolation. All datasets follow a unified data storage format based on the Parquet/Apache Arrow standard, consistent with the data organization adopted in prior work (Woo et al., 2024). Based on UTSD, we construct a diverse pretraining subset spanning multiple application domains, comprising approximately 67 million time points in total. Following the protocol in (Liu et al., 2024; He et al., 2025), the data are split into training and validation sets with a ratio of 9:1. The key characteristics of the included datasets are reported in Table 7, including domain categories, temporal resolution, data scale, storage size, Augmented Dickey–Fuller (ADF) statistics, forecastability, and data sources.

**Evaluation Datasets.** For zero-, few-, and full-shot forecasting scenarios, we evaluate Olivia on the widely used Time-Series-Library (TSLib) benchmark (Wu et al., 2022), which includes representative datasets such as ETTh1, ETTh2, ETTm2, Weather, Electricity, and Traffic. These datasets span diverse application domains and exhibit varying temporal dynamics, seasonal patterns, and prediction difficulties, providing a standardized testbed for assessing forecasting performance under different data availability settings. We also include GIFT-Eval (Aksu et al., 2024) for evaluation, a large-scale

*Table 7.* List of pretraining datasets. All datasets are selected from the UTSD data repositories (Liu et al., 2024). Dataset attributes follow the original UTSD specifications.

| Dataset | Domain | Resolution | Time Points | File Size | ADF. | Forecast. | Source |
|---|---|---|---|---|---|---|---|
| PEMS04 | Transport | 5 Min | 15.65M | 60M | -15.192 | 0.494 | (Liu et al., 2022a) |
| PEMS08 | Transport | 5 Min | 9.11M | 35M | -14.918 | 0.551 | (Liu et al., 2022a) |
| Pedestrian Counts | Transport | Hourly | 3.13M | 12M | -23.462 | 0.297 | (Godahewa et al., 2021) |
| Beijing PM2.5 Quality | Environment | Hourly | 3.66M | 14M | -31.415 | 0.404 | (Tan et al., 2021) |
| PIGCVP | Health | – | 0.62M | 3M | -4.855 | 0.577 | (Dau et al., 2019) |
| Australian Electricity Demand | Energy | 30 Min | 1.16M | 5M | -27.554 | 0.730 | (Godahewa et al., 2021) |
| Wind | Energy | 4 Sec | 7.40M | 29M | -29.174 | 0.811 | (Godahewa et al., 2021) |
| KDD Cup 2018 | Nature | Hourly | 2.94M | 12M | -10.107 | 0.362 | (Godahewa et al., 2021) |
| Temperature Rain | Nature | Daily | 23.25M | 93M | -10.952 | 0.133 | (Godahewa et al., 2021) |
| Saugeen River Flow | Nature | Daily | 0.02M | 1M | -19.305 | 0.300 | (Godahewa et al., 2021) |
| Sunspot | Nature | Daily | 0.07M | 1M | -7.866 | 0.287 | (Godahewa et al., 2021) |

and comprehensive benchmark for assessing general time series forecasting models, particularly in zero-shot settings. It comprises 23 datasets with over 144,000 time series and 177 million data points, covering seven application domains and 10 temporal frequencies, and supports multivariate inputs with forecasting horizons ranging from short- to long-term. By unifying diverse datasets under a consistent evaluation protocol, GIFT-Eval facilitates systematic assessment of model generalization and promotes the development of foundation models for time series forecasting. Following Sundial, we also conduct zero-shot performance evaluation on the datasets provided by GluonTS (Alexandrov et al., 2020).

### E.3. Implementation Details

All experiments are conducted on a workstation equipped with two NVIDIA RTX 5090 GPUs, each with 32GB of memory. Model pretraining takes approximately 5 hours, while the subsequent tuning stage requires around 2.5 hours. Unless otherwise specified, we adopt a consistent default configuration across all experiments, using a batch size of 2048, 10 epochs for pretraining, and 2 epochs for tuning. The model is instantiated with 6 transformer layers and 16 attention heads, a patch length of 64, and an embedding dimension of 256, while the number of learnable Householder reflections is set to 256, balancing model capacity and computational efficiency. For optimization, we use the AdamW optimizer with a learning rate of $1 \times 10^{-3}$ for pretraining and $8 \times 10^{-4}$ for tuning, respectively.

## F. Visualization Analysis

### F.1. Visualizations of Temporal Correlation

Figure 9 provides a comparative visualization of temporal correlation structures across multiple domains before and after processing with the Aligner in the Harmonizer. In the first row, the raw correlation matrices exhibit strong domain-specific characteristics, such as pronounced periodic stripes, smooth diagonal dominance, or irregular textures, reflecting substantial differences in spectral composition and temporal dynamics across domains. After applying the Harmonizer, the correlation patterns in the second row become more comparable in their overall structure, indicating that the Aligner reorganizes second-order temporal dependencies into a shared correlation space. Importantly, however, the resulting matrices do not collapse into identical or trivial patterns. Although their global appearances are more unified, noticeable differences remain among domains, manifested in variations of local textures, diagonal decay behaviors, and fine-grained correlation structures. This observation suggests that the Harmonizer does not eliminate domain-specific information, but instead transforms how such information is expressed.

Functionally, the Harmonizer maps domain-specific correlation structures into a common coordinate system, in which transferable temporal regularities are expressed more consistently while domain-relevant characteristics are preserved. By reducing superficial discrepancies induced by factors such as sampling rates, dominant periodicities, or energy distributions, while preserving intrinsic temporal dynamics, the Aligner achieves a balance between cross-domain comparability and domain-level fidelity. The coexistence of increased structural consistency and preserved inter-domain differences in the second row thus provides qualitative evidence that the Harmonizer promotes transferable representations without collapsing heterogeneous temporal patterns into a degenerate template.

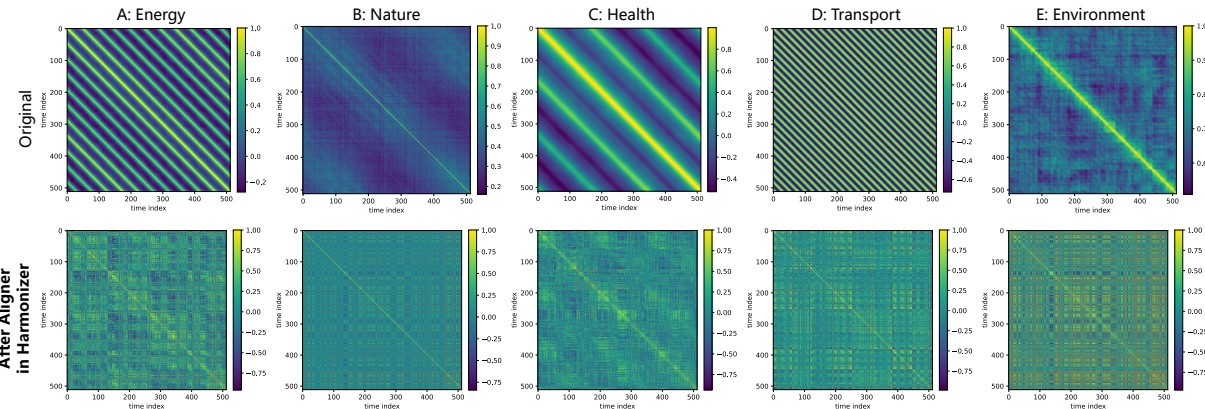

*Figure 9.* Visualization of temporal correlation. This figure illustrates the temporal correlation of raw time series from different domains in the pretraining datasets, as well as the corresponding correlation patterns after processing with the Aligner in the Harmonizer.

## F.2. Weight Visualization of Learnable Orthogonal Matrix **Q**

Figure 10 visualizes the learned orthogonal matrix **Q**, where the structured patterns can be observed beyond a trivial identity-like form. While the dominant diagonal reflects the orthogonality constraint, the presence of non-negligible off-diagonal components indicates that **Q** performs non-trivial temporal reorganization rather than simple time-wise preservation. To further examine how such structure is constructed, Figure 11 presents visualizations of the individual learnable Householder reflections $\{\mathbf{H}_k\}$ at different reflection indices $k$. The progressive variation in their patterns suggests that successive reflections contribute complementary transformations, which are explicitly composed to form the overall orthogonal matrix **Q**.

From an optimization perspective, parameterizing **Q** as a product of multiple Householder reflections provides a gradual and stable mechanism for learning orthogonal transformations. Instead of enforcing orthogonality through hard constraints or penalty terms in the optimization objective, this formulation incrementally builds **Q** via a sequence of simple, well-conditioned reflections. Such a design avoids common optimization bottlenecks associated with constrained orthogonal learning, while still allowing sufficient flexibility to capture structured temporal reorganization. The observed patterns across different $\mathbf{H}_k$ further suggest that the orthogonal transformation is learned in a progressive and distributed manner, rather than being dominated by a single abrupt rotation.

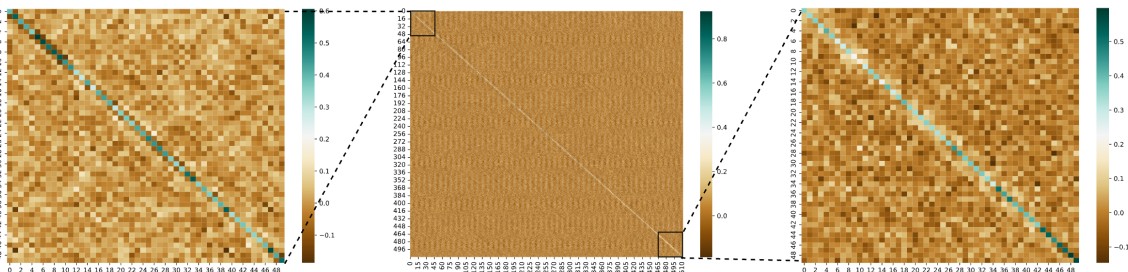

*Figure 10.* Visualization of the learnable orthogonal matrix **Q**

## F.3. Visualizations of Predictions

Figure 12 and Figure 13 provide qualitative comparisons of zero-shot forecasting behaviors on the ETTm2 and Weather datasets under the zero-shot setting. The predictions produced by Olivia exhibit smoother trajectories and more stable oscillatory patterns over extended horizons, whereas several baseline methods display noticeable deviations such as phase offsets or attenuated fluctuations. These visual differences suggest that Olivia maintains more reliable temporal evolution when extrapolating beyond the observed context.

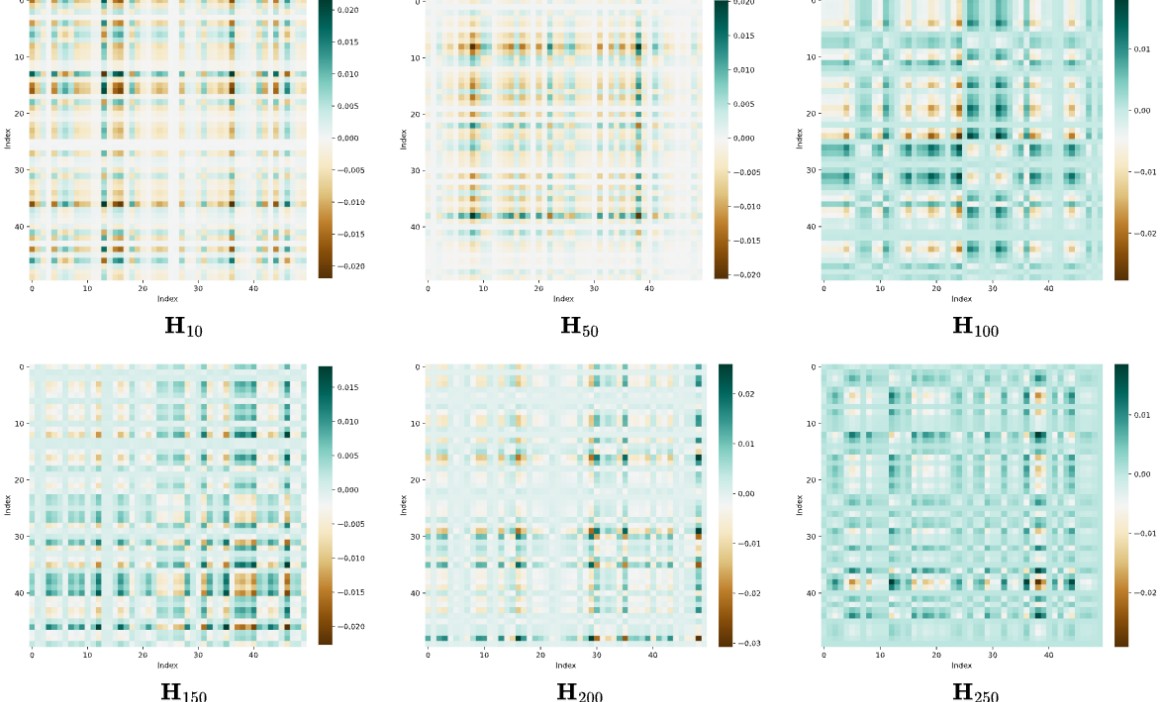

$\mathbf{H}_{10}$ $\quad$ $\mathbf{H}_{50}$ $\quad$ $\mathbf{H}_{100}$

$\mathbf{H}_{150}$ $\quad$ $\mathbf{H}_{200}$ $\quad$ $\mathbf{H}_{250}$

*Figure 11.* Visualization of learnable Householder Reflections $\{\mathbf{H}_k\}$ at different reflection indices $k$. The structured patterns across reflections illustrate how successive learnable Householder transformations contribute to the construction of the overall orthogonal matrix $\mathbf{Q}$. To better reveal the learned transformation patterns, each reflection is shown relative to the identity matrix.

## G. Supplementary Results

### G.1. Zero-shot Forecasting

Table 8 reports zero-shot results on the GIFT-Eval benchmark, showing that Olivia consistently outperforms SEMPO and the Moirai variants on most datasets. In particular, Olivia exhibits the most pronounced improvements over SEMPO on bizitobs_service and bizitobs_application in terms of NRMSE, achieving relative reductions of over 23%, while also delivering consistent gains on M_DENSE (SMAPE) and SZ_TAXI (SMAPE).

Table 9 reports the detailed experimental results corresponding to those summarized in Table 1 of the main text, further confirming that Olivia consistently achieves superior forecasting accuracy across a wide range of datasets and prediction horizons, while maintaining stable performance advantages over both lightweight and large-scale foundation model baselines. Table 10 presents a comprehensive zero-shot comparison with representative time series foundation models on the TSLib benchmark. Overall, the results indicate that Olivia delivers more stable and robust zero-shot forecasting across diverse temporal domains and horizons, suggesting a stronger ability to generalize under heterogeneous temporal distributions.

Table 11 reports a zero-shot comparison with representative lightweight time series foundation models across multiple datasets and prediction horizons. Despite using significantly fewer parameters, Olivia achieves the highest number of first-place results (28), outperforming TTM (18) and SEMPO (4), indicating consistently strong zero-shot generalization. Notably, Olivia shows clear advantages on large-scale datasets such as Electricity and Traffic, where it attains the best average MSE and MAE across all horizons. These results suggest that PSD-guided temporal harmonization enables an effective balance between model efficiency and generalization capability.

### G.2. Few-shot and Full-shot Forecasting

**Few-shot Forecasting.** Table 12 reports the few-shot forecasting results on the TSLib benchmark with only 10% training data. Overall, Olivia consistently achieves the best or second-best performance across most datasets and prediction horizons, demonstrating strong data efficiency under limited supervision. When compared with pretrained time-series foundation models such as SEMPO and TTM, Olivia shows clear advantages in both MSE and MAE on the majority of

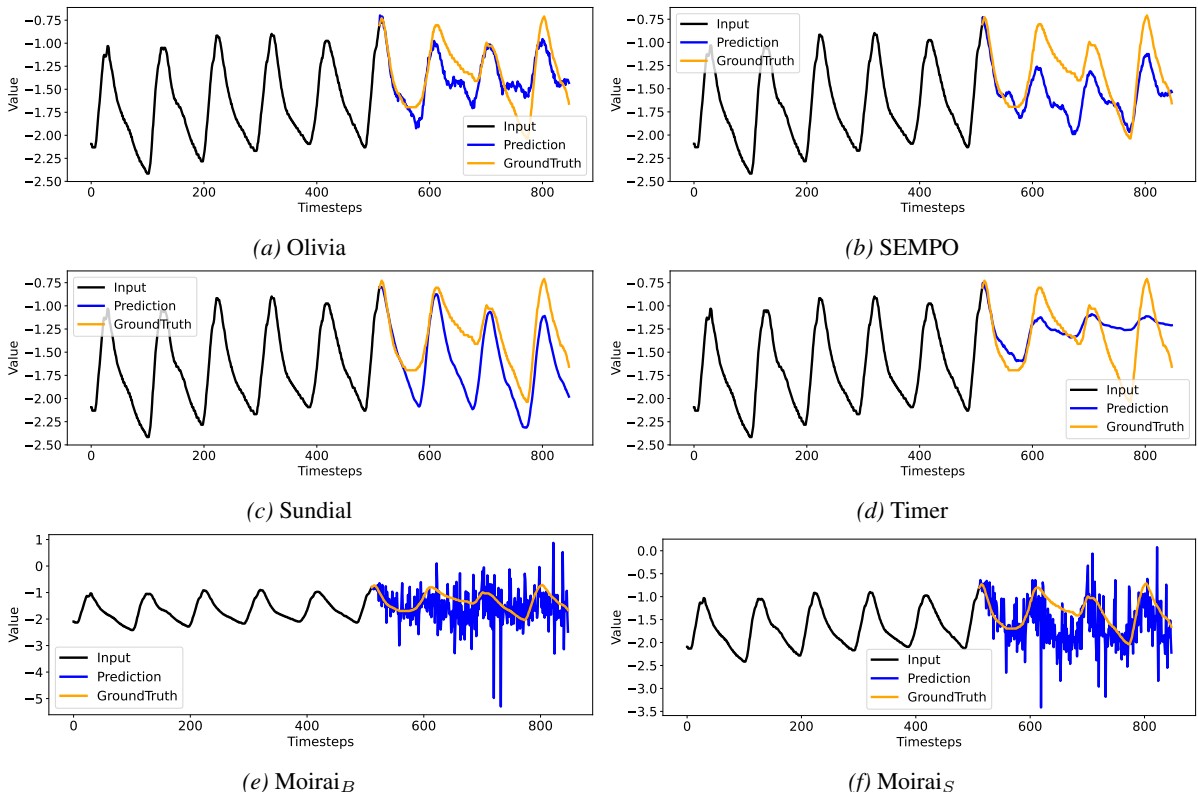

*Figure 12.* Visualization of prediction results on the ETTm2 dataset under the zero-shot setting, with an input length of 512 and a prediction horizon of 336.

datasets. This indicates that, beyond large-scale pretraining alone, explicitly aligning cross-domain temporal structures is crucial for effective few-shot adaptation. Relative to LLM-based models, including Time-LLM, GPT4TS, and S$^2$IP-LLM, Olivia delivers substantially more stable and competitive performance across all benchmarks. While LLM-based approaches benefit from strong language priors, their forecasting accuracy appears less reliable under limited labeled data, whereas Olivia maintains robust performance without relying on external language supervision. Finally, compared with task-specific forecasting models such as iTransformer, DLinear, PatchTST, TimesNet, Stationary, and FEDformer, Olivia consistently outperforms these methods across most datasets. This suggests that Olivia effectively bridges the gap between foundation-level generalization and task-specific inductive biases, achieving superior few-shot forecasting performance without sacrificing cross-dataset robustness.

**Full-shot Forecasting.** Table 13 reports the full-shot forecasting results on ETTh2, ETTm2, Weather, and Traffic datasets, corresponding to the aggregated visualization in Figure 3. Overall, Olivia consistently achieves strong performance across all four datasets and prediction horizons, yielding the highly competitive average MSE and MAE in most cases. Compared with pretrained time series foundation models (e.g., SEMPO) and LLM-based approaches (e.g., GPT4TS and S$^2$IP-LLM), Olivia demonstrates more stable accuracy across diverse datasets and horizons. In contrast, task-specific forecasting models such as iTransformer, DLinear, and PatchTST generally deliver inferior performance compared with Olivia across most datasets and prediction horizons. These results indicate that Olivia's PSD-consistent temporal representations generalize effectively under the full-shot setting, providing robust performance across diverse datasets.

### G.3. Comparison of Different Model variants

Table 14 compares different model variants under the zero-shot setting by varying input context length for both Olivia and SEMPO. Across all four datasets, Olivia$_B$ ($T = 512$, 5.1M parameters) consistently achieves the best or highly competitive average performance among all Olivia variants, despite being the smallest configuration. Increasing the context length from Olivia$_B$ to Olivia$_E$ ($T = 1024$, 6.3M parameters) and Olivia$_A$ ($T = 1536$, 9.6M parameters) does not lead to systematic improvements, and in several cases results in slightly degraded accuracy, indicating diminishing returns from extended

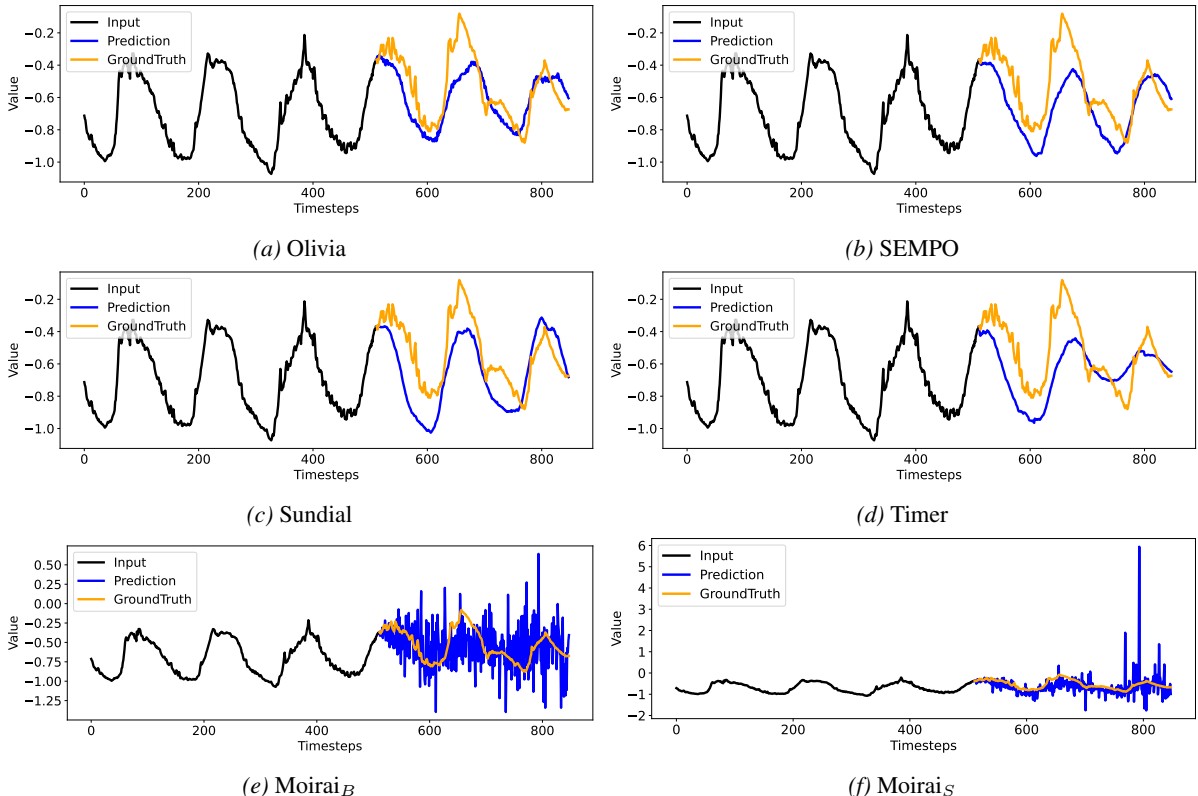

*(a)* Olivia

*(b)* SEMPO

*(c)* Sundial

*(d)* Timer

*(e)* Moirai$_B$

*(f)* Moirai$_S$

*Figure 13.* Visualization of prediction results on the Weather dataset under the zero-shot setting, with an input length of 512 and a prediction horizon of 336.

temporal context under the same pretraining setup.

A similar pattern is observed for SEMPO, where SEMPO$_B$ ($T = 512$, 6.5M parameters) often matches or outperforms its larger counterparts SEMPO$_E$ ($T = 1024$, 7.3M parameters) and SEMPO$_A$ ($T = 1536$, 9.9M parameters). Notably, despite using fewer parameters, Olivia variants achieve forecasting performance comparable to, and in many cases better than, SEMPO variants across multiple datasets and prediction horizons. These results indicate that increasing model size or input context length alone does not necessarily improve zero-shot generalization. Meanwhile, Olivia consistently attains comparable performance with fewer parameters, demonstrating favorable parameter efficiency under this evaluation setting.

### G.4. Scaling Analysis

Figures 14(a)-(b) show a consistent non-monotonic trend on both the Weather and Traffic datasets as the scale of pretraining data increases. Forecasting error initially decreases when moving from a small to a moderate data scale, indicating that additional data helps the model capture more representative temporal patterns. However, beyond this range, further data expansion no longer yields improvements and instead leads to gradual performance degradation. To this end, we discuss several plausible factors that may underlie this observed behavior.

One natural point of reference is the notion of scaling laws (Kaplan et al., 2020; Hestness et al., 2017), which describe empirical relationships between model performance and data or model scale, and have been widely validated in domains such as natural language processing (Chen et al., 2025; Isik et al., 2024) and computer vision (Wang et al., 2025a; Zhai et al., 2022). In these areas, larger datasets primarily improve coverage of semantic or visual variability. Compared with language and image data, time series data are generated by underlying dynamical systems and exhibit stronger temporal dependencies, domain-specific time scales, and structured spectral properties (Wang et al., 2024). Consequently, increasing pretraining data scale not only adds samples but also introduces heterogeneous temporal statistics across datasets, making effective scaling more dependent on how such structures are coordinated. In addition, model capacity may also play a role. Olivia is a lightweight foundation model with 5.1M parameters, which offers clear efficiency benefits but may limit its ability to fully absorb increasingly large and heterogeneous pretraining corpora. Taken together, these observations suggest

*Table 8.* Zero-shot performance evaluation on the GIFT-Eval benchmark (Aksu et al., 2024). A lower NRMSE/SMAPE indicates a better result. The best results are in **red** and the second best are **blue**. 'S': Small, 'B': Base, 'L': Large. '/' denotes degenerate results with abnormally large errors under the zero-shot setting, which are omitted for clarity.

| Datasets | Metrics | **Olivia** | SEMPO | Moirai$_S$ | Moirai$_B$ | Moirai$_L$ |
|---|---|---|---|---|---|---|
| M_DENSE | NRMSE | **0.356** | **0.363** | 0.679 | 0.491 | 0.528 |
| | SMAPE | **0.259** | **0.297** | 0.356 | 0.299 | 0.298 |
| LOOP_SEATTLE | NRMSE | **0.194** | **0.191** | / | / | / |
| | SMAPE | **0.157** | **0.155** | 0.203 | 0.188 | 0.184 |
| SZ_TAXI | NRMSE | **0.348** | **0.369** | / | / | / |
| | SMAPE | **0.394** | **0.422** | 0.624 | 0.644 | 0.644 |
| bizitobs_application | NRMSE | **0.175** | **0.229** | 0.274 | 0.277 | 0.307 |
| | SMAPE | 0.120 | 0.160 | 0.128 | **0.114** | **0.118** |
| bizitobs_l2c | NRMSE | **0.879** | **0.928** | / | 2.896 | 10.855 |
| | SMAPE | **0.969** | **1.078** | 1.164 | 1.180 | 1.222 |
| bizitobs_service | NRMSE | **0.220** | **0.290** | 0.630 | / | 0.411 |
| | SMAPE | **0.201** | 0.270 | 0.245 | **0.231** | 0.234 |
| jena_weather | NRMSE | **0.220** | **0.241** | 0.675 | 0.791 | 0.972 |
| | SMAPE | **0.661** | **0.699** | 0.802 | 0.771 | 0.778 |

that scaling in time series pretraining is influenced by multiple interacting factors, including data composition, temporal structure consistency, and model capacity. Effective scaling therefore requires a careful balance among these factors, rather than relying on data expansion alone.

Figures 14(c)–(d) examine the effect of model depth on zero-shot forecasting performance for the Electricity and Traffic datasets. As the number of layers increases from 2 to 4, the forecasting error decreases noticeably on both datasets, suggesting that moderate depth helps the model capture more expressive temporal representations. However, further increasing the depth beyond this point does not lead to continued improvements, and performance instead exhibits mild degradation. This behavior indicates that the benefits of increasing model depth are subject to diminishing returns in the

*Table 9.* Full zero-shot results on the TSLib benchmark, with the best results are in **red** and the second best are **blue**. '-' denotes dataset used in pre-training and excluded in the evaluation. 'S': Small, 'B': Base, 'L': Large. Avg means the average results from all four prediction lengths.

| Models | | **Olivia Ours** | | SEMPO [NeurIPS'25] | | Time-MoE$_B$ [ICLR'25] | | Time-MoE$_L$ [ICLR'25] | | Timer [ICML'24] | | Moirai$_S$ [ICML'24] | | Moirai$_B$ [ICML'24] | | Moirai$_L$ [ICML'24] | | Chronos$_S$ [TMLR'24] | | Chronos$_B$ [TMLR'24] | | Chronos$_L$ [TMLR'24] | | TimesFM [ICML'24] | |
|---|---|---|---|---|---|---|---|---|---|---|---|---|---|---|---|---|---|---|---|---|---|---|---|---|---|
| Metrics | | MSE | MAE | MSE | MAE | MSE | MAE | MSE | MAE | MSE | MAE | MSE | MAE | MSE | MAE | MSE | MAE | MSE | MAE | MSE | MAE | MSE | MAE | MSE | MAE |
| ETTh1 | 96 | 0.370 | 0.397 | 0.384 | 0.408 | **0.358** | **0.382** | **0.350** | **0.382** | 0.414 | 0.439 | 0.407 | 0.405 | 0.394 | 0.399 | 0.410 | 0.403 | 0.476 | 0.411 | 0.452 | **0.396** | 0.453 | 0.401 | 0.432 | 0.405 |
| | 192 | **0.396** | **0.415** | 0.409 | 0.426 | 0.404 | **0.417** | **0.402** | 0.423 | 0.440 | 0.455 | 0.443 | 0.425 | 0.431 | 0.430 | 0.459 | 0.434 | 0.547 | 0.452 | 0.513 | 0.429 | 0.525 | 0.428 | 0.492 | 0.438 |
| | 336 | **0.407** | **0.424** | **0.417** | **0.433** | 0.449 | 0.454 | 0.447 | 0.459 | 0.455 | 0.463 | 0.465 | 0.438 | 0.450 | 0.437 | 0.486 | 0.453 | 0.589 | 0.481 | 0.553 | 0.450 | 0.570 | 0.452 | 0.519 | 0.458 |
| | 720 | **0.424** | **0.447** | **0.432** | **0.454** | 0.570 | 0.543 | 0.543 | 0.534 | 0.496 | 0.496 | 0.475 | 0.461 | 0.456 | 0.457 | 0.511 | 0.483 | 0.592 | 0.509 | 0.578 | 0.484 | 0.619 | 0.493 | 0.512 | 0.477 |
| | Avg | **0.399** | **0.421** | **0.410** | **0.430** | 0.445 | 0.449 | 0.435 | 0.449 | 0.451 | 0.463 | 0.448 | 0.432 | 0.433 | 0.431 | 0.466 | 0.443 | 0.551 | 0.463 | 0.524 | 0.439 | 0.541 | 0.443 | 0.489 | 0.444 |
| ETTh2 | 96 | **0.274** | **0.340** | **0.282** | **0.342** | 0.302 | 0.356 | 0.301 | 0.354 | 0.305 | 0.355 | 0.288 | 0.345 | 0.285 | **0.342** | 0.294 | 0.344 | 0.309 | 0.354 | 0.307 | 0.352 | 0.297 | 0.344 | 0.311 | 0.345 |
| | 192 | **0.328** | **0.378** | **0.334** | **0.384** | 0.438 | 0.431 | 0.407 | 0.417 | 0.365 | 0.406 | 0.352 | 0.396 | 0.352 | 0.391 | 0.368 | 0.387 | 0.386 | 0.399 | 0.388 | 0.388 | 0.379 | 0.392 | 0.401 | 0.397 |
| | 336 | **0.359** | **0.402** | **0.355** | **0.403** | 0.617 | 0.509 | 0.514 | 0.476 | 0.378 | 0.413 | 0.375 | 0.420 | 0.384 | 0.418 | 0.403 | 0.411 | 0.428 | 0.426 | 0.432 | 0.422 | 0.414 | 0.412 | 0.436 | 0.430 |
| | 720 | **0.393** | **0.432** | **0.395** | **0.435** | 0.907 | 0.622 | 0.689 | 0.561 | 0.414 | 0.457 | 0.405 | 0.442 | 0.418 | 0.446 | 0.465 | 0.453 | 0.454 | 0.455 | 0.442 | 0.443 | 0.451 | 0.452 | 0.437 | 0.450 |
| | Avg | **0.339** | **0.388** | **0.341** | **0.391** | 0.566 | 0.479 | 0.477 | 0.452 | 0.366 | 0.408 | 0.355 | 0.401 | 0.360 | 0.399 | 0.382 | 0.397 | 0.394 | 0.409 | 0.392 | 0.401 | 0.385 | 0.400 | 0.396 | 0.405 |
| ETTm2 | 96 | **0.193** | 0.286 | 0.199 | 0.289 | 0.197 | 0.287 | 0.197 | 0.285 | 0.203 | 0.285 | 0.228 | 0.296 | 0.227 | 0.290 | 0.223 | 0.288 | 0.209 | 0.287 | 0.204 | **0.280** | 0.206 | **0.283** | **0.189** | 0.257 |
| | 192 | **0.256** | **0.323** | 0.256 | 0.325 | 0.338 | 0.381 | 0.329 | 0.376 | **0.265** | 0.327 | 0.275 | 0.324 | 0.306 | 0.334 | 0.303 | 0.331 | 0.280 | 0.332 | 0.271 | **0.321** | 0.279 | 0.330 | 0.277 | 0.325 |
| | 336 | **0.314** | **0.358** | 0.308 | 0.355 | 0.586 | 0.501 | 0.534 | 0.481 | 0.319 | 0.361 | 0.335 | 0.360 | 0.366 | 0.373 | 0.354 | 0.364 | 0.344 | 0.371 | 0.327 | 0.362 | 0.339 | 0.365 | 0.350 | 0.381 |
| | 720 | **0.400** | **0.409** | 0.392 | 0.404 | 1.034 | 0.683 | 0.978 | 0.668 | 0.405 | 0.410 | 0.453 | 0.425 | 0.456 | 0.429 | 0.456 | 0.423 | 0.448 | 0.430 | 0.432 | 0.415 | 0.437 | 0.420 | 0.464 | 0.448 |
| | Avg | **0.291** | **0.344** | 0.289 | 0.343 | 0.538 | 0.463 | 0.509 | 0.452 | 0.298 | 0.346 | 0.323 | 0.351 | 0.339 | 0.356 | 0.334 | 0.352 | 0.320 | 0.355 | 0.308 | **0.344** | 0.315 | 0.350 | 0.320 | 0.353 |
| Weather | 96 | **0.169** | 0.228 | 0.171 | 0.228 | **0.159** | **0.213** | **0.159** | **0.214** | 0.190 | 0.236 | 0.180 | 0.229 | 0.208 | 0.221 | 0.213 | **0.213** | 0.212 | 0.238 | 0.198 | 0.276 | 0.207 | 0.232 | - | - |
| | 192 | **0.216** | **0.266** | 0.218 | 0.269 | **0.214** | 0.267 | 0.217 | 0.271 | 0.261 | 0.293 | 0.231 | 0.279 | 0.281 | 0.270 | 0.342 | **0.262** | 0.266 | 0.283 | 0.248 | 0.276 | 0.260 | 0.277 | - | - |
| | 336 | **0.266** | **0.303** | 0.267 | 0.304 | 0.292 | 0.326 | 0.312 | 0.343 | 0.332 | 0.340 | 0.289 | 0.330 | 0.340 | 0.313 | 0.527 | 0.312 | 0.321 | 0.321 | 0.301 | 0.314 | 0.310 | 0.313 | - | - |
| | 720 | **0.335** | **0.352** | 0.336 | 0.350 | 0.453 | 0.430 | 0.586 | 0.508 | 0.385 | 0.381 | 0.367 | 0.385 | 0.420 | 0.376 | 0.826 | 0.369 | 0.393 | 0.367 | 0.386 | 0.361 | 0.390 | 0.365 | - | - |
| | Avg | **0.247** | **0.287** | 0.248 | 0.287 | 0.279 | 0.309 | 0.318 | 0.334 | 0.292 | 0.312 | 0.267 | 0.306 | 0.312 | 0.306 | 0.477 | **0.289** | 0.298 | 0.302 | 0.283 | 0.295 | 0.292 | 0.297 | - | - |
| Electricity | 96 | **0.158** | **0.260** | **0.168** | 0.271 | - | - | - | - | 0.210 | 0.312 | 0.212 | 0.304 | 0.169 | 0.269 | 0.193 | 0.275 | 0.199 | **0.267** | 0.196 | 0.273 | 0.198 | 0.278 | - | - |
| | 192 | **0.173** | **0.273** | 0.183 | **0.283** | - | - | - | - | 0.239 | 0.337 | 0.224 | 0.315 | 0.186 | 0.285 | 0.186 | 0.296 | 0.218 | 0.289 | 0.207 | 0.285 | 0.214 | 0.294 | - | - |
| | 336 | **0.190** | **0.288** | 0.198 | 0.297 | - | - | - | - | 0.284 | 0.372 | 0.244 | 0.331 | 0.215 | 0.299 | 0.221 | 0.311 | 0.244 | 0.321 | 0.238 | 0.314 | 0.239 | 0.314 | - | - |
| | 720 | **0.230** | **0.320** | 0.238 | 0.329 | - | - | - | - | 0.456 | 0.479 | 0.291 | 0.365 | 0.257 | 0.332 | 0.296 | 0.355 | 0.324 | 0.371 | 0.310 | 0.355 | 0.316 | 0.362 | - | - |
| | Avg | **0.188** | **0.285** | 0.196 | 0.295 | - | - | - | - | 0.297 | 0.375 | 0.243 | 0.329 | 0.207 | 0.296 | 0.224 | 0.309 | 0.246 | 0.312 | 0.336 | 0.329 | 0.326 | 0.328 | - | - |
| Traffic | 96 | **0.432** | **0.318** | 0.441 | 0.333 | - | - | - | - | 0.526 | 0.368 | - | - | - | - | - | - | 0.562 | 0.378 | 0.558 | 0.375 | 0.541 | 0.364 | - | - |
| | 192 | **0.448** | **0.325** | 0.456 | 0.339 | - | - | - | - | 0.561 | 0.385 | - | - | - | - | - | - | 0.579 | 0.412 | 0.560 | 0.399 | 0.570 | 0.406 | - | - |
| | 336 | **0.460** | **0.331** | 0.467 | 0.344 | - | - | - | - | 0.614 | 0.412 | - | - | - | - | - | - | 0.594 | 0.420 | 0.584 | 0.413 | 0.571 | 0.404 | - | - |
| | 720 | **0.493** | **0.347** | 0.503 | 0.360 | - | - | - | - | 0.749 | 0.464 | - | - | - | - | - | - | 0.723 | 0.472 | 0.711 | 0.464 | 0.719 | 0.469 | - | - |
| | Avg | **0.458** | **0.330** | 0.466 | 0.344 | - | - | - | - | 0.613 | 0.407 | - | - | - | - | - | - | 0.614 | 0.420 | 0.603 | 0.413 | 0.600 | 0.411 | - | - |
| $1^{st}$ Count | | **43** | | **10** | | 4 | | 3 | | 0 | | 0 | | 0 | | 2 | | 1 | | 2 | | 0 | | 1 | |

*Table 10.* Zero-shot comparison with other representative time series foundation models on the TSLib benchmark, with the best results are in **red**. Avg means the average results from all four prediction lengths.

| Datasets | | ETTh1 | | ETTh2 | | ETTm2 | | Weather | | Electricity | | Traffic | | $1^{st}$ Count |
|---|---|---|---|---|---|---|---|---|---|---|---|---|---|---|---|
| Metrics | | MSE | MAE | MSE | MAE | MSE | MAE | MSE | MAE | MSE | MAE | MSE | MAE | |
| Moment | 96 | 0.706 | 0.561 | 0.373 | 0.416 | 0.230 | 0.308 | 0.216 | 0.271 | 0.844 | 0.761 | 1.390 | 0.800 | |
| | 192 | 0.716 | 0.579 | 0.384 | 0.422 | 0.285 | 0.338 | 0.264 | 0.306 | 0.850 | 0.762 | 1.403 | 0.802 | |
| | 336 | 0.705 | 0.583 | 0.386 | 0.426 | 0.339 | 0.369 | 0.313 | 0.336 | 0.862 | 0.766 | 1.415 | 0.804 | 0 |
| | 720 | 0.705 | 0.597 | 0.425 | 0.454 | 0.423 | 0.424 | 0.369 | 0.380 | 0.888 | 0.774 | 1.437 | 0.808 | |
| | Avg | 0.708 | 0.580 | 0.392 | 0.430 | 0.319 | 0.360 | 0.291 | 0.323 | 0.861 | 0.766 | 1.411 | 0.804 | |
| ROSE | 96 | 0.382 | 0.408 | 0.298 | 0.362 | 0.224 | 0.309 | 0.200 | 0.260 | 0.209 | 0.307 | 0.572 | 0.407 | |
| | 192 | 0.400 | 0.420 | 0.336 | 0.385 | 0.266 | 0.333 | 0.239 | 0.288 | 0.219 | 0.315 | 0.575 | 0.406 | |
| | 336 | **0.404** | 0.426 | **0.353** | **0.399** | **0.310** | **0.358** | 0.279 | 0.315 | 0.236 | 0.330 | 0.588 | 0.411 | 10 |
| | 720 | **0.420** | **0.447** | 0.395 | **0.432** | **0.395** | **0.407** | 0.340 | 0.357 | 0.273 | 0.328 | 0.618 | 0.411 | |
| | Avg | 0.401 | 0.425 | 0.346 | 0.394 | 0.299 | 0.352 | 0.265 | 0.305 | 0.234 | 0.320 | 0.618 | 0.422 | |
| Sundial | 96 | 0.420 | 0.423 | 0.336 | 0.369 | 0.215 | **0.281** | 0.202 | **0.228** | **0.135** | **0.230** | 0.509 | **0.303** | |
| | 192 | 0.458 | 0.449 | 0.405 | 0.413 | 0.288 | 0.328 | 0.266 | 0.280 | **0.156** | **0.250** | 0.541 | **0.322** | |
| | 336 | 0.472 | 0.460 | 0.421 | 0.430 | 0.350 | 0.366 | 0.322 | 0.319 | **0.174** | **0.269** | 0.564 | 0.334 | 15 |
| | 720 | 0.507 | 0.488 | 0.448 | 0.456 | 0.442 | 0.421 | 0.392 | 0.368 | **0.217** | **0.305** | 0.611 | 0.362 | |
| | Avg | 0.464 | 0.455 | 0.403 | 0.417 | 0.324 | 0.349 | 0.296 | 0.299 | **0.171** | **0.264** | 0.556 | **0.330** | |
| Olivia | 96 | **0.370** | **0.397** | **0.274** | **0.340** | **0.193** | 0.286 | **0.169** | **0.228** | 0.158 | 0.260 | **0.432** | 0.318 | |
| | 192 | **0.396** | **0.415** | **0.328** | **0.378** | **0.256** | **0.323** | **0.216** | **0.266** | 0.173 | 0.273 | **0.448** | 0.325 | |
| | 336 | 0.407 | **0.424** | 0.359 | 0.402 | 0.314 | **0.358** | **0.266** | **0.303** | 0.190 | 0.288 | **0.460** | **0.331** | 39 |
| | 720 | 0.424 | **0.447** | **0.393** | **0.432** | 0.400 | 0.409 | **0.335** | **0.352** | 0.230 | 0.320 | **0.493** | **0.347** | |
| | Avg | **0.399** | **0.421** | **0.339** | **0.388** | **0.291** | **0.344** | **0.247** | **0.287** | 0.188 | 0.285 | **0.458** | 0.330 | |

considered setting. While deeper architectures offer greater representational capacity, they may also introduce additional optimization difficulty or over-parameterization relative to the scale and structure of the pretraining data. Notably, the best-performing configurations remain consistently below the SEMPO baseline across both datasets, highlighting that Olivia achieves a favorable balance between model capacity and efficiency within a compact architectural regime.

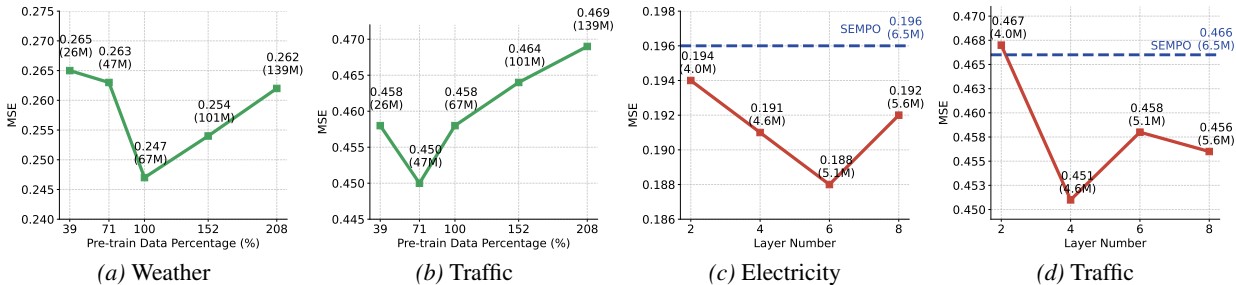

| *(a)* Weather | *(b)* Traffic | *(c)* Electricity | *(d)* Traffic |

*Figure 14.* Scaling analysis under the zero-shot setting, considering both model size and the scale of pretraining datasets. The reported results are averaged across all prediction lengths.

### G.5. Parameter Sensitivity Analysis

**The number of learnable Householder reflections $K$.** Figure 15 presents the sensitivity analysis with respect to the number of learnable Householder reflections $K$ used to construct the orthogonal matrix $\mathbf{Q}$. Across both ETTh1 and ETTh2, the performance exhibits a clear non-monotonic trend as $K$ varies, with moderate values consistently yielding better results than either very small or excessively large configurations. In particular, setting $K = 256$ achieves the lowest MSE on both datasets, indicating a favorable balance between transformation expressiveness and optimization stability. When $K$ is too small, the resulting orthogonal transformation lacks sufficient capacity to reorganize temporal correlations effectively, leading to suboptimal forecasting performance. Conversely, overly large $K$ introduces additional computational overhead and may increase optimization difficulty without providing commensurate gains. These observations suggest that the proposed Householder-based parameterization benefits from a moderate number of reflections, supporting our design choice of using a fixed, intermediate $K$ in all experiments.

*Table 11.* Zero-shot comparison with current lightweight time series foundation models, with the best results are in **red**. Values in ( , ) denote model size and pre-training data size, respectively. Avg means the average results from all four prediction lengths.

| Datasets | | | ETTh2 | | ETTm2 | | Weather | | Electricity | | Traffic | | $1^{st}$ Count |
|---|---|---|---|---|---|---|---|---|---|---|---|---|---|
| Metrics | | | MSE | MAE | MSE | MAE | MSE | MAE | MSE | MAE | MSE | MAE | |
| TTM | (1M, 1B) | 96 | 0.276 | **0.335** | **0.177** | **0.259** | **0.151** | **0.196** | 0.182 | 0.274 | 0.508 | 0.342 | |
| | | 192 | 0.346 | 0.383 | **0.246** | **0.307** | **0.195** | **0.241** | 0.196 | 0.287 | 0.538 | 0.355 | |
| | | 336 | 0.385 | 0.417 | 0.324 | **0.350** | **0.256** | **0.285** | 0.219 | 0.307 | 0.574 | 0.373 | 18 |
| | | 720 | 0.420 | 0.450 | 0.406 | 0.405 | **0.319** | **0.333** | 0.261 | 0.342 | 0.617 | 0.390 | |
| | | Avg | 0.357 | 0.396 | **0.288** | **0.330** | **0.230** | **0.264** | 0.215 | 0.303 | 0.559 | 0.365 | |
| SEMPO | (6.5M, 83M) | 96 | 0.282 | 0.342 | 0.199 | 0.289 | 0.171 | 0.228 | 0.168 | 0.271 | 0.447 | 0.334 | |
| | | 192 | 0.334 | 0.384 | 0.256 | 0.325 | 0.218 | 0.269 | 0.183 | 0.283 | 0.464 | 0.341 | |
| | | 336 | **0.355** | 0.403 | **0.308** | 0.355 | 0.267 | 0.304 | 0.198 | 0.297 | 0.474 | 0.344 | 4 |
| | | 720 | 0.395 | 0.435 | **0.392** | **0.404** | 0.336 | 0.350 | 0.238 | 0.329 | 0.509 | 0.359 | |
| | | Avg | 0.341 | 0.391 | 0.289 | 0.343 | 0.248 | 0.287 | 0.196 | 0.295 | 0.474 | 0.345 | |
| Olivia | (5.1M, 67M) | 96 | **0.274** | 0.340 | 0.193 | 0.286 | 0.169 | 0.228 | **0.158** | **0.260** | **0.432** | **0.318** | |
| | | 192 | **0.328** | **0.378** | 0.256 | 0.323 | 0.216 | 0.266 | **0.173** | **0.273** | **0.448** | **0.325** | |
| | | 336 | 0.359 | **0.402** | 0.314 | 0.358 | 0.266 | 0.303 | **0.190** | **0.288** | **0.460** | **0.331** | 28 |
| | | 720 | **0.393** | **0.432** | 0.400 | 0.409 | 0.335 | 0.352 | **0.230** | **0.320** | **0.493** | **0.347** | |
| | | Avg | **0.339** | **0.388** | 0.291 | 0.344 | 0.247 | 0.287 | **0.188** | **0.285** | **0.458** | **0.330** | |

**The number of resonators** $M$**.** The design of HarmonicAttention is motivated by the low-rank structure of token interactions in the PSD-aligned space (Proposition 2), where most temporal energy is concentrated in a small number of dominant modes. Accordingly, $M$ serves as an efficiency–capacity trade-off parameter rather than a fragile bottleneck. To empirically validate this, we conduct a sensitivity analysis with $M \in \{L/4, L/2, L\}$, where the results are averaged over different forecasting horizons. As shown in Table 15, smaller values (e.g., $M = L/4$) already achieve the best performance across nearly all datasets, while increasing $M$ does not bring further improvements and may even lead to slight degradation. These results suggest that the proposed mechanism can effectively capture the dominant interaction structure without requiring a large number of resonators.

*Table 12.* Full few-shot results on the TSLib benchmark with 10% training data. The best results are in **red** and the second best are **blue**. Avg means the average results from all four prediction lengths.

| Models | | Olivia | | SEMPO | | TTM | | Time-LLM | | GPT4TS | | S²IP-LLM | | iTransformer | | DLinear | | PatchTST | | TimesNet | | Stationary | | FEDformer | |
|---|---|---|---|---|---|---|---|---|---|---|---|---|---|---|---|---|---|---|---|---|---|---|---|---|---|---|---|
| Metrics | | MSE | MAE | MSE | MAE | MSE | MAE | MSE | MAE | MSE | MAE | MSE | MAE | MSE | MAE | MSE | MAE | MSE | MAE | MSE | MAE | MSE | MAE | MSE | MAE |
| ETTh1 | 96 | **0.365** | **0.392** | 0.378 | 0.405 | **0.362** | **0.389** | 0.448 | 0.460 | 0.458 | 0.456 | 0.463 | 0.459 | 0.790 | 0.586 | 0.492 | 0.495 | 0.516 | 0.485 | 0.861 | 0.628 | 0.918 | 0.639 | 0.512 | 0.499 |
| | 192 | **0.392** | **0.408** | 0.407 | 0.424 | **0.386** | 0.408 | 0.484 | 0.483 | 0.570 | 0.516 | 0.482 | 0.487 | 0.837 | 0.609 | 0.565 | 0.538 | 0.598 | 0.524 | 0.797 | 0.593 | 0.915 | 0.629 | 0.624 | 0.555 |
| | 336 | **0.404** | **0.420** | 0.426 | 0.439 | **0.399** | **0.420** | 0.589 | 0.540 | 0.608 | 0.535 | 0.603 | 0.543 | 0.780 | 0.575 | 0.721 | 0.622 | 0.657 | 0.550 | 0.941 | 0.648 | 0.939 | 0.644 | 0.691 | 0.574 |
| | 720 | **0.432** | **0.452** | 0.456 | 0.474 | 0.465 | 0.477 | 0.700 | 0.604 | 0.725 | 0.591 | 0.713 | 0.588 | 1.234 | 0.811 | 0.986 | 0.743 | 0.762 | 0.610 | 0.877 | 0.641 | 0.887 | 0.645 | 0.728 | 0.614 |
| | Avg | **0.398** | **0.418** | 0.417 | 0.435 | **0.403** | **0.423** | 0.556 | 0.522 | 0.590 | 0.525 | 0.565 | 0.524 | 0.910 | 0.860 | 0.691 | 0.600 | 0.633 | 0.542 | 0.869 | 0.628 | 0.915 | 0.639 | 0.639 | 0.561 |
| ETTh2 | 96 | **0.272** | **0.337** | 0.276 | 0.340 | **0.275** | **0.335** | 0.275 | 0.326 | 0.331 | 0.374 | 0.300 | 0.360 | 0.404 | 0.435 | 0.357 | 0.411 | 0.353 | 0.389 | 0.378 | 0.409 | 0.389 | 0.411 | 0.382 | 0.416 |
| | 192 | **0.326** | **0.375** | **0.332** | 0.376 | 0.345 | 0.383 | 0.374 | **0.373** | 0.402 | 0.411 | 0.372 | 0.371 | 0.470 | 0.474 | 0.569 | 0.519 | 0.403 | 0.414 | 0.490 | 0.467 | 0.473 | 0.455 | 0.478 | 0.474 |
| | 336 | **0.357** | **0.399** | 0.354 | **0.399** | 0.384 | 0.416 | 0.406 | 0.429 | 0.406 | 0.433 | 0.389 | **0.413** | 0.489 | 0.485 | 0.671 | 0.572 | 0.426 | 0.441 | 0.537 | 0.494 | 0.507 | 0.480 | 0.504 | 0.501 |
| | 720 | **0.392** | **0.430** | 0.395 | 0.433 | 0.419 | 0.450 | 0.427 | 0.449 | 0.449 | 0.464 | 0.403 | 0.426 | 0.593 | 0.538 | 0.824 | 0.648 | 0.477 | 0.480 | 0.510 | 0.491 | 0.477 | 0.472 | 0.499 | 0.509 |
| | Avg | **0.337** | **0.385** | **0.339** | **0.387** | 0.355 | 0.391 | 0.370 | 0.394 | 0.397 | 0.421 | 0.366 | 0.392 | 0.489 | 0.483 | 0.605 | 0.538 | 0.415 | 0.431 | 0.479 | 0.465 | 0.462 | 0.455 | 0.466 | 0.475 |
| ETTm2 | 96 | **0.163** | **0.251** | 0.166 | 0.256 | 0.176 | 0.260 | 0.177 | 0.261 | 0.188 | 0.269 | **0.140** | **0.242** | 0.245 | 0.322 | 0.213 | 0.303 | 0.191 | 0.274 | 0.212 | 0.285 | 0.229 | 0.308 | 0.291 | 0.399 |
| | 192 | **0.220** | **0.290** | 0.223 | 0.295 | 0.242 | 0.304 | 0.241 | 0.314 | 0.251 | 0.309 | **0.207** | **0.293** | 0.274 | 0.338 | 0.252 | 0.317 | 0.270 | 0.323 | 0.291 | 0.343 | 0.307 | 0.379 |
| | 336 | 0.280 | 0.333 | 0.276 | **0.329** | 0.315 | 0.345 | **0.274** | 0.327 | 0.307 | 0.346 | **0.264** | **0.331** | 0.361 | 0.394 | 0.338 | 0.385 | 0.306 | 0.353 | 0.323 | 0.353 | 0.348 | 0.376 | 0.543 | 0.559 |
| | 720 | **0.361** | **0.385** | 0.357 | 0.381 | 0.394 | 0.399 | 0.417 | 0.390 | 0.426 | 0.417 | 0.381 | 0.387 | 0.467 | 0.442 | 0.436 | 0.440 | 0.433 | 0.427 | 0.474 | 0.449 | 0.461 | 0.438 | 0.712 | 0.614 |
| | Avg | 0.256 | **0.315** | **0.255** | **0.315** | 0.281 | 0.327 | 0.277 | 0.323 | 0.293 | 0.335 | **0.248** | **0.313** | 0.336 | 0.373 | 0.316 | 0.368 | 0.296 | 0.343 | 0.320 | 0.353 | 0.332 | 0.366 | 0.463 | 0.488 |
| Weather | 96 | **0.149** | 0.202 | 0.152 | 0.204 | **0.152** | **0.199** | 0.161 | 0.210 | 0.163 | 0.215 | 0.154 | **0.201** | 0.253 | 0.307 | 0.171 | 0.224 | 0.165 | 0.215 | 0.184 | 0.230 | 0.192 | 0.234 | 0.188 | 0.253 |
| | 192 | **0.194** | **0.240** | 0.196 | 0.243 | **0.193** | 0.245 | 0.204 | 0.248 | 0.210 | 0.254 | 0.195 | **0.241** | 0.292 | 0.328 | 0.215 | 0.263 | 0.210 | 0.257 | 0.245 | 0.283 | 0.269 | 0.295 | 0.250 | 0.304 |
| | 336 | **0.244** | **0.279** | 0.246 | 0.282 | 0.246 | 0.282 | 0.261 | 0.302 | 0.256 | 0.292 | 0.260 | 0.302 | 0.322 | 0.346 | 0.258 | 0.299 | 0.259 | 0.297 | 0.305 | 0.321 | 0.370 | 0.357 | 0.312 | 0.346 |
| | 720 | 0.319 | 0.336 | 0.319 | 0.336 | 0.336 | 0.346 | **0.309** | **0.332** | 0.321 | 0.339 | **0.303** | **0.329** | 0.365 | 0.374 | 0.320 | 0.346 | 0.332 | 0.346 | 0.381 | 0.371 | 0.441 | 0.405 | 0.387 | 0.393 |
| | Avg | 0.227 | **0.264** | 0.228 | 0.266 | 0.232 | 0.268 | 0.234 | 0.273 | 0.238 | 0.275 | **0.228** | 0.268 | 0.308 | 0.338 | 0.241 | 0.283 | 0.242 | 0.279 | 0.279 | 0.301 | 0.318 | 0.323 | 0.284 | 0.324 |
| Electricity | 96 | **0.128** | **0.220** | 0.134 | 0.231 | 0.140 | 0.237 | 0.139 | 0.241 | 0.139 | 0.237 | 0.142 | 0.243 | 0.154 | 0.257 | 0.150 | 0.253 | 0.140 | 0.238 | 0.299 | 0.373 | 0.420 | 0.466 | 0.231 | 0.323 |
| | 192 | **0.143** | **0.239** | 0.151 | 0.247 | 0.163 | 0.259 | **0.151** | 0.248 | 0.156 | 0.252 | 0.163 | 0.260 | 0.171 | 0.272 | 0.164 | 0.264 | 0.160 | 0.255 | 0.305 | 0.379 | 0.411 | 0.459 | 0.261 | 0.356 |
| | 336 | **0.158** | **0.260** | 0.167 | 0.267 | 0.180 | 0.275 | 0.169 | 0.270 | 0.175 | 0.270 | 0.173 | 0.270 | 0.196 | 0.295 | 0.181 | 0.282 | 0.180 | 0.276 | 0.319 | 0.391 | 0.434 | 0.473 | 0.360 | 0.445 |
| | 720 | **0.193** | **0.288** | 0.201 | 0.298 | 0.241 | 0.326 | 0.240 | 0.322 | 0.233 | 0.317 | 0.237 | 0.320 | 0.263 | 0.348 | 0.223 | 0.321 | 0.241 | 0.323 | 0.369 | 0.426 | 0.510 | 0.521 | 0.530 | 0.585 |
| | Avg | **0.156** | **0.252** | 0.163 | 0.261 | 0.181 | 0.274 | 0.175 | 0.270 | 0.176 | 0.269 | 0.178 | 0.273 | 0.196 | 0.293 | 0.180 | 0.280 | 0.180 | 0.273 | 0.323 | 0.392 | 0.444 | 0.480 | 0.346 | 0.427 |
| Traffic | 96 | **0.383** | 0.266 | 0.395 | 0.277 | 0.408 | 0.292 | 0.418 | 0.291 | 0.414 | 0.297 | 0.401 | 0.285 | 0.448 | 0.329 | 0.419 | 0.298 | 0.403 | 0.289 | 0.719 | 0.416 | 1.412 | 0.802 | 0.639 | 0.400 |
| | 192 | **0.402** | 0.274 | 0.407 | 0.282 | 0.417 | 0.295 | 0.414 | 0.296 | 0.426 | 0.301 | 0.410 | 0.293 | 0.487 | 0.360 | 0.434 | 0.305 | 0.415 | 0.296 | 0.748 | 0.428 | 1.419 | 0.806 | 0.637 | 0.416 |
| | 336 | **0.405** | 0.278 | 0.417 | 0.287 | 0.430 | 0.302 | 0.421 | 0.311 | 0.434 | 0.303 | 0.425 | 0.314 | 0.514 | 0.372 | 0.449 | 0.313 | 0.426 | 0.304 | 0.853 | 0.471 | 1.443 | 0.815 | 0.655 | 0.427 |
| | 720 | **0.445** | 0.299 | 0.450 | 0.304 | 0.476 | 0.331 | 0.462 | 0.327 | 0.487 | 0.337 | 0.470 | 0.330 | 0.532 | 0.383 | 0.484 | 0.336 | 0.474 | 0.331 | 1.485 | 0.825 | 1.539 | 0.837 | 0.722 | 0.456 |
| | Avg | **0.409** | 0.279 | 0.417 | 0.287 | 0.432 | 0.305 | 0.429 | 0.306 | 0.440 | 0.310 | 0.426 | 0.305 | 0.495 | 0.361 | 0.447 | 0.313 | 0.430 | 0.305 | 0.951 | 0.535 | 1.453 | 0.815 | 0.663 | 0.425 |
| $1^{st}$ Count | | 39 | | 5 | | 9 | | 1 | | 0 | | 9 | | 0 | | 0 | | 0 | | 0 | | 0 | | 0 | |

### G.6. Additional Results

**Detailed Ablation Results on HarmonicAttention.** Table 16 reports the detailed ablation results of HarmonicAttention across individual prediction horizons, serving as the full version of the averaged results summarized in Table 3 of the main text. The horizon-wise results reveal consistent and systematic trends that are obscured by averaging.

*Table 13.* Full-shot results on the ETTh2, ETTm2, Weather and Traffic datasets. The best results are in **red**. Avg means the average results from all four prediction lengths.

| Models | | **Olivia** | | SEMPO | | GPT4TS | | S²IP-LLM | | iTransformer | | DLinear | | PatchTST | |
|---|---|---|---|---|---|---|---|---|---|---|---|---|---|---|---|
| Metrics | | MSE | MAE | MSE | MAE | MSE | MAE | MSE | MAE | MSE | MAE | MSE | MAE | MSE | MAE |
| ETTh2 | 96 | **0.269** | **0.330** | 0.273 | 0.334 | 0.285 | 0.342 | 0.278 | 0.340 | 0.297 | 0.348 | 0.302 | 0.368 | 0.274 | 0.337 |
| | 192 | 0.324 | **0.372** | 0.333 | 0.376 | 0.354 | 0.389 | **0.246** | 0.385 | 0.371 | 0.403 | 0.404 | 0.433 | 0.341 | 0.382 |
| | 336 | 0.355 | 0.401 | 0.355 | 0.400 | 0.373 | 0.407 | 0.367 | 0.406 | 0.404 | 0.428 | 0.511 | 0.498 | **0.329** | **0.384** |
| | 720 | 0.397 | 0.433 | 0.399 | 0.435 | 0.406 | 0.441 | 0.400 | 0.436 | 0.424 | 0.444 | 0.815 | 0.640 | **0.379** | **0.422** |
| | Avg | 0.336 | 0.384 | 0.340 | 0.386 | 0.355 | 0.395 | **0.323** | 0.392 | 0.374 | 0.406 | 0.508 | 0.485 | 0.331 | **0.381** |
| ETTm2 | 96 | **0.155** | 0.254 | 0.160 | **0.251** | 0.173 | 0.262 | 0.165 | 0.257 | 0.175 | 0.266 | 0.164 | 0.255 | 0.166 | 0.256 |
| | 192 | **0.218** | **0.290** | 0.221 | 0.294 | 0.229 | 0.301 | 0.222 | 0.299 | 0.242 | 0.312 | 0.224 | 0.304 | 0.223 | 0.296 |
| | 336 | **0.270** | 0.330 | 0.273 | **0.328** | 0.286 | 0.341 | 0.277 | 0.330 | 0.282 | 0.340 | 0.277 | 0.339 | 0.274 | 0.329 |
| | 720 | 0.353 | 0.382 | **0.349** | **0.380** | 0.378 | 0.401 | 0.363 | 0.390 | 0.378 | 0.398 | 0.371 | 0.401 | 0.362 | 0.385 |
| | Avg | **0.249** | 0.314 | 0.251 | **0.313** | 0.266 | 0.326 | 0.257 | 0.319 | 0.269 | 0.329 | 0.259 | 0.325 | 0.256 | 0.317 |
| Weather | 96 | **0.143** | **0.189** | **0.143** | 0.193 | 0.162 | 0.212 | 0.145 | 0.195 | 0.159 | 0.208 | 0.170 | 0.230 | 0.149 | 0.198 |
| | 192 | **0.185** | **0.229** | 0.189 | 0.234 | 0.204 | 0.248 | 0.190 | 0.235 | 0.200 | 0.248 | 0.212 | 0.267 | 0.194 | 0.241 |
| | 336 | **0.237** | **0.276** | 0.240 | 0.280 | 0.254 | 0.286 | 0.243 | 0.280 | 0.253 | 0.289 | 0.257 | 0.305 | 0.245 | 0.282 |
| | 720 | 0.320 | 0.335 | 0.325 | 0.338 | 0.326 | 0.337 | **0.312** | **0.326** | 0.321 | 0.338 | 0.318 | 0.356 | 0.314 | 0.334 |
| | Avg | **0.221** | **0.257** | 0.224 | 0.261 | 0.236 | 0.271 | 0.222 | 0.259 | 0.233 | 0.271 | 0.239 | 0.289 | 0.225 | 0.264 |
| Traffic | 96 | **0.350** | 0.250 | 0.355 | **0.246** | 0.388 | 0.282 | 0.379 | 0.274 | 0.363 | 0.265 | 0.411 | 0.294 | 0.360 | 0.249 |
| | 192 | **0.369** | **0.250** | 0.373 | 0.253 | 0.407 | 0.290 | 0.397 | 0.282 | 0.385 | 0.273 | 0.421 | 0.298 | 0.379 | 0.256 |
| | 336 | **0.383** | **0.256** | 0.384 | 0.260 | 0.412 | 0.294 | 0.407 | 0.289 | 0.396 | 0.277 | 0.431 | 0.304 | 0.392 | 0.264 |
| | 720 | **0.420** | **0.280** | 0.427 | 0.286 | 0.450 | 0.312 | 0.440 | 0.301 | 0.445 | 0.312 | 0.468 | 0.325 | 0.432 | 0.286 |
| | Avg | **0.381** | **0.260** | 0.385 | 0.261 | 0.414 | 0.294 | 0.406 | 0.286 | 0.397 | 0.282 | 0.433 | 0.305 | 0.391 | 0.264 |

*Table 14.* Comparison of different model variants under the zero-shot setting. The best results are highlighted in **red**. Values in ( , ) denote model size and pretraining data scale, respectively. 'B': Base, 'E': Enhanced, 'A': Advanced. Avg means the average results from all four prediction lengths.

| Models | | Olivia$_B$ (5.1M, 67M) | | Olivia$_E$ (6.3M, 67M) | | Olivia$_A$ (9.6M, 67M) | | SEMPO$_B$ (6.5M, 83M) | | SEMPO$_E$ (7.3M, 83M) | | SEMPO$_A$ (9.9M, 83M) | |
|---|---|---|---|---|---|---|---|---|---|---|---|---|---|
| Metrics | | MSE | MAE | MSE | MAE | MSE | MAE | MSE | MAE | MSE | MAE | MSE | MAE |
| ETTh1 | 96 | **0.370** | **0.397** | 0.375 | 0.408 | 0.381 | 0.414 | 0.384 | 0.408 | 0.392 | 0.419 | 0.392 | 0.423 |
| | 192 | **0.396** | **0.415** | 0.402 | 0.427 | 0.410 | 0.434 | 0.409 | 0.426 | 0.415 | 0.433 | 0.422 | 0.443 |
| | 336 | **0.407** | **0.424** | 0.415 | 0.437 | 0.426 | 0.445 | 0.417 | 0.433 | 0.423 | 0.440 | 0.435 | 0.452 |
| | 720 | **0.424** | **0.447** | 0.442 | 0.465 | 0.448 | 0.469 | 0.432 | 0.454 | 0.440 | 0.465 | 0.447 | 0.468 |
| | Avg | **0.399** | **0.421** | 0.409 | 0.434 | 0.416 | 0.441 | 0.410 | 0.430 | 0.417 | 0.439 | 0.424 | 0.446 |
| ETTh2 | 96 | **0.274** | **0.340** | 0.299 | 0.353 | 0.310 | 0.363 | 0.282 | 0.342 | 0.309 | 0.365 | 0.308 | 0.362 |
| | 192 | **0.328** | **0.378** | 0.352 | 0.390 | 0.370 | 0.403 | 0.334 | 0.384 | 0.362 | 0.400 | 0.370 | 0.405 |
| | 336 | 0.359 | **0.402** | 0.381 | 0.413 | 0.387 | 0.418 | **0.355** | 0.403 | 0.378 | 0.415 | 0.388 | 0.420 |
| | 720 | **0.393** | **0.432** | 0.402 | **0.432** | 0.412 | 0.436 | 0.395 | 0.435 | 0.399 | 0.417 | 0.409 | 0.440 |
| | Avg | **0.339** | **0.388** | 0.359 | 0.397 | 0.370 | 0.405 | 0.341 | 0.391 | 0.362 | 0.399 | 0.368 | 0.406 |
| ETTm2 | 96 | 0.193 | 0.286 | 0.196 | 0.286 | 0.199 | 0.289 | 0.196 | 0.286 | 0.194 | 0.284 | **0.191** | **0.283** |
| | 192 | 0.256 | 0.323 | 0.249 | 0.320 | 0.252 | 0.322 | 0.252 | 0.323 | **0.245** | **0.318** | 0.247 | 0.320 |
| | 336 | 0.314 | 0.358 | 0.298 | 0.350 | 0.292 | **0.345** | 0.306 | 0.354 | 0.292 | 0.347 | **0.286** | **0.345** |
| | 720 | 0.400 | 0.409 | 0.383 | 0.398 | 0.370 | 0.392 | 0.391 | 0.404 | 0.372 | 0.395 | **0.360** | **0.390** |
| | Avg | 0.291 | 0.344 | 0.282 | 0.339 | 0.278 | 0.337 | 0.286 | 0.341 | 0.275 | 0.336 | **0.271** | **0.334** |
| Weather | 96 | **0.169** | 0.228 | 0.172 | **0.227** | 0.179 | 0.239 | 0.171 | 0.228 | 0.174 | 0.231 | 0.177 | 0.234 |
| | 192 | **0.216** | **0.266** | 0.217 | **0.266** | 0.226 | 0.276 | 0.218 | 0.269 | 0.217 | 0.268 | 0.231 | 0.278 |
| | 336 | 0.266 | 0.303 | 0.267 | 0.302 | 0.272 | 0.307 | 0.267 | 0.304 | **0.261** | **0.299** | 0.273 | 0.307 |
| | 720 | 0.335 | 0.352 | 0.332 | **0.343** | 0.331 | 0.348 | 0.336 | 0.350 | **0.325** | **0.343** | 0.334 | 0.346 |
| | Avg | 0.247 | 0.287 | 0.247 | **0.285** | 0.252 | 0.293 | 0.248 | 0.287 | **0.244** | **0.285** | 0.253 | 0.291 |
| 1$^{st}$ Count | | | | **27** | | | | | | **17** | | | |

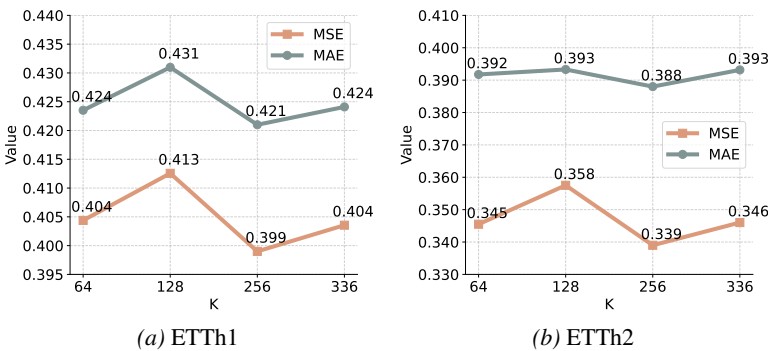

|  |  |
|---|---|
| *(a)* ETTh1 | *(b)* ETTh2 |

*Figure 15.* Parameter sensitivity analysis on the number $K$ of learnable Householder reflections $\mathbf{H}_k$ used to construct the orthogonal matrix $\mathbf{Q}$. The reported results are averaged across all prediction lengths.

*Table 15.* Sensitivity analysis of the number of resonators $M$. Results are averaged over the four prediction horizons.

| Datasets | ETTh1 | | ETTh2 | | ETTm2 | | Weather | | Electricity | | Traffic | |
|---|---|---|---|---|---|---|---|---|---|---|---|---|
| Metrics | MSE | MAE | MSE | MAE | MSE | MAE | MSE | MAE | MSE | MAE | MSE | MAE |
| $M = L/4$ | **0.399** | **0.421** | **0.339** | **0.388** | **0.291** | **0.344** | **0.247** | **0.287** | **0.188** | **0.285** | **0.458** | **0.330** |
| $M = L/2$ | 0.404 | 0.425 | 0.345 | 0.391 | 0.298 | 0.350 | 0.251 | 0.290 | 0.190 | 0.286 | 0.462 | **0.330** |
| $M = L$ | 0.404 | 0.424 | 0.350 | 0.393 | 0.305 | 0.354 | 0.255 | 0.292 | 0.191 | 0.288 | 0.459 | 0.332 |

*Table 16.* Ablations on HarmonicAttention, where we replace HarmonicAttention with other attention mechanisms.

| Datasets | | ETTh1 | | Electricity | | Weather | |
|---|---|---|---|---|---|---|---|
| Metrics | | MSE | MAE | MSE | MAE | MSE | MAE |
| HarmonicAttention | 96 | 0.370 | 0.397 | 0.158 | 0.260 | 0.169 | 0.228 |
| | 192 | 0.396 | 0.415 | 0.173 | 0.273 | 0.216 | 0.266 |
| | 336 | 0.407 | 0.424 | 0.190 | 0.288 | 0.266 | 0.303 |
| | 720 | 0.424 | 0.447 | 0.230 | 0.320 | 0.335 | 0.352 |
| Full Attention | 96 | 0.417 | 0.434 | 0.170 | 0.272 | 0.183 | 0.237 |
| | 192 | 0.448 | 0.457 | 0.187 | 0.286 | 0.233 | 0.281 |
| | 336 | 0.477 | 0.476 | 0.207 | 0.304 | 0.294 | 0.328 |
| | 720 | 0.545 | 0.513 | 0.252 | 0.339 | 0.361 | 0.374 |
| Linear Attention | 96 | 0.376 | 0.402 | 0.163 | 0.266 | 0.176 | 0.233 |
| | 192 | 0.407 | 0.422 | 0.180 | 0.280 | 0.224 | 0.276 |
| | 336 | 0.425 | 0.434 | 0.198 | 0.295 | 0.278 | 0.317 |
| | 720 | 0.438 | 0.454 | 0.236 | 0.327 | 0.351 | 0.365 |
| Nyström Attention | 96 | 0.416 | 0.436 | 0.178 | 0.277 | 0.183 | 0.237 |
| | 192 | 0.458 | 0.465 | 0.201 | 0.295 | 0.231 | 0.280 |
| | 336 | 0.499 | 0.492 | 0.221 | 0.313 | 0.289 | 0.324 |
| | 720 | 0.579 | 0.536 | 0.264 | 0.345 | 0.362 | 0.373 |

**Robustness Analysis.** To evaluate the robustness of Olivia under the zero-shot forecasting setting, we report the results over five independent training runs with different random seeds. As shown in Table 17, the standard deviations of both MSE and MAE remain consistently small across all datasets and forecasting horizons, indicating stable optimization behavior and low sensitivity to initialization. In particular, most fluctuations are within the range of 0.000–0.003, demonstrating highly consistent performance across different runs. These results verify the robustness and reliability of Olivia in zero-shot forecasting scenarios.

**Efficiency Analysis.** In addition to the efficiency analysis conducted on the ETTh1 dataset in the main text (Table 4), we further evaluate the computational efficiency of Olivia on the Weather dataset under the zero-shot setting, with detailed results reported in Table 19. Consistent with the observations on ETTh1, Olivia achieves competitive forecasting accuracy on Weather while maintaining a compact model size (5.1M parameters). Although its inference time is higher than that of SEMPO, this overhead primarily stems from the Harmonizer module, where the orthogonal temporal transformation

*Table 17.* Robustness of Olivia under the zero-shot forecasting setting. Results are reported over five independent training runs from scratch using different random seeds.

| Dataset | ETTh1 | | ETTh2 | | ETTm2 | |
|---|---|---|---|---|---|---|
| Horizon | MSE | MAE | MSE | MAE | MSE | MAE |
| 96 | $0.370 \pm 0.001$ | $0.397 \pm 0.001$ | $0.274 \pm 0.003$ | $0.340 \pm 0.002$ | $0.193 \pm 0.000$ | $0.286 \pm 0.000$ |
| 192 | $0.396 \pm 0.001$ | $0.415 \pm 0.002$ | $0.328 \pm 0.002$ | $0.378 \pm 0.002$ | $0.256 \pm 0.000$ | $0.323 \pm 0.001$ |
| 336 | $0.407 \pm 0.001$ | $0.424 \pm 0.002$ | $0.359 \pm 0.002$ | $0.402 \pm 0.002$ | $0.314 \pm 0.001$ | $0.358 \pm 0.001$ |
| 720 | $0.424 \pm 0.001$ | $0.447 \pm 0.002$ | $0.393 \pm 0.004$ | $0.432 \pm 0.002$ | $0.400 \pm 0.001$ | $0.409 \pm 0.001$ |
| Avg | $0.399 \pm 0.001$ | $0.421 \pm 0.002$ | $0.339 \pm 0.003$ | $0.388 \pm 0.002$ | $0.291 \pm 0.001$ | $0.344 \pm 0.001$ |
| Dataset | Weather | | Electricity | | Traffic | |
| Horizon | MSE | MAE | MSE | MAE | MSE | MAE |
| 96 | $0.169 \pm 0.001$ | $0.228 \pm 0.001$ | $0.158 \pm 0.001$ | $0.260 \pm 0.002$ | $0.432 \pm 0.000$ | $0.318 \pm 0.000$ |
| 192 | $0.216 \pm 0.002$ | $0.266 \pm 0.000$ | $0.173 \pm 0.002$ | $0.273 \pm 0.002$ | $0.448 \pm 0.000$ | $0.325 \pm 0.000$ |
| 336 | $0.266 \pm 0.002$ | $0.303 \pm 0.001$ | $0.190 \pm 0.002$ | $0.288 \pm 0.001$ | $0.460 \pm 0.001$ | $0.331 \pm 0.002$ |
| 720 | $0.335 \pm 0.001$ | $0.352 \pm 0.001$ | $0.230 \pm 0.003$ | $0.320 \pm 0.002$ | $0.493 \pm 0.000$ | $0.347 \pm 0.001$ |
| Avg | $0.247 \pm 0.002$ | $0.287 \pm 0.001$ | $0.188 \pm 0.002$ | $0.285 \pm 0.002$ | $0.458 \pm 0.000$ | $0.330 \pm 0.001$ |

matrix $\mathbf{Q} \in \mathbb{R}^{T \times T}$ is explicitly constructed via the sequential multiplication of $K$ learnable Householder reflections. In our implementation, we set $K = 256$, which introduces additional but moderate computational cost during inference.

Taken together, the results in Table 4 and Table 19 demonstrate that Olivia consistently strikes a favorable balance between forecasting accuracy, model capacity, and inference efficiency across datasets. Despite the added cost introduced by the Harmonizer, Olivia remains substantially more efficient than large-scale foundation models such as the Time-MoE family, while achieving superior accuracy compared to lightweight baselines. This confirms that the proposed design offers an effective accuracy–efficiency trade-off in cross-domain zero-shot forecasting scenarios.

Based on these efficiency results, a promising direction for future work is to further investigate more efficient strategies for constructing or approximating the orthogonal transformation matrix $\mathbf{Q}$, which could reduce inference overhead while preserving the benefits of PSD-consistent temporal reorganization.

*Table 18.* Efficiency comparison on ETTh1 dataset under the zero-shot scenario. The reported MSE and inference time values are averaged over all prediction horizons.

| Model | MSE | Inference Time (s) | Model Size (M) |
|---|---|---|---|
| Olivia | **0.399** | 43.051 | **5.1** |
| SEMPO | 0.410 | **8.162** | 6.5 |
| Time-MoE$_L$ | 0.435 | 10231.867 | 453 |
| Time-MoE$_B$ | 0.445 | 4050.209 | 113 |
| Timer | 0.451 | 103.822 | 67.4 |
| Moirai$_L$ | 0.466 | 67.906 | 311 |
| Moirai$_B$ | 0.433 | 60.455 | 91 |
| Moirai$_S$ | 0.448 | 54.330 | 14 |
| Sundial | 0.464 | 67.901 | 128 |

*Table 19.* Efficiency comparison on Weather dataset under the zero-shot scenario. The reported MSE and inference time values are averaged over all prediction horizons.

| Model | MSE | Inference Time (s) | Model Size (M) |
|---|---|---|---|
| Olivia | **0.247** | 541.937 | **5.1** |
| SEMPO | 0.248 | **163.310** | 6.5 |
| Time-MoE$_L$ | 0.318 | 86771.882 | 453 |
| Time-MoE$_B$ | 0.279 | 53004.187 | 113 |
| Timer | 0.292 | 1299.640 | 67.4 |
| Moirai$_L$ | 0.477 | 395.800 | 311 |
| Moirai$_B$ | 0.312 | 255.146 | 91 |
| Moirai$_S$ | 0.267 | 191.978 | 14 |
| Sundial | 0.296 | 353.997 | 128 |

