# OpenReview forum: "Olivia: Harmonizing Time Series Foundation Models with Power Spectral Density"
_ICML.cc/2026/Conference — ICML 2026 regular_

### Official Review · Reviewer_p9xh · 2026-03-02

**Soundness:** 2
**Presentation:** 2
**Significance:** 2
**Originality:** 2
**Overall Recommendation:** 2
**Confidence:** 4

**Summary:**

The authors propose Olivia, a foundational forecasting model. The authors argue that time series data varies heavily across domains with respect to temporal patterns. To address this, they propose to harmonise the power spectral density of the data. In particular, they introduce Harmonizer and HarmonicAttention.

**Compliance With Llm Reviewing Policy:**

Affirmed.

**Final Justification:**

After reading the authors' rebuttal and responses, including the ones to the other reviewers, I would like to keep my score and confidence. While an interesting stream of research, the study is unfortunately not ready for publication. Best wishes

**Key Questions For Authors:**

- In Figure 1, results of the data harmonisation are provided. In Figure 1a), spectrograms shows that data is transformed through the harmonisation. However, it seems that domain-specific patterns get lost through this process. Is this actually the case or does the harmonisation process preserve domain-specific information, such as the periodicity of the energy or transport data, that could be useful for downstream tasks? In Figure 1b), JS divergence shows that absolute differences between domains drop substantially after harmonisation. However, it seems that the relative differences increase between domains: $JS_{DE}$ / $JS_{EA}$ = 1.76 before and = 2 after harmonisation. How can we interpret these JS numbers and how does a relative increase in divergence benefit pre-training?

- In Figure 2, an overview of the architecture is provided. Is there any intuitive and simple way to explain and visualise HarmonicAttention? I am uncertain whether the figure aids to this.

- In Figure 4, visualisation of the harmonisation process is provided. What are we supposed to see from the changes in second-order temporal correlations? How does this affect pre-training?

- In Figure 14 in the Appendix, results on scaling the pre-training dataset are provided. The experiments suggest that further increasing the dataset size hurts downstream performance. This is in contrast to expectations that larger pre-training data might be beneficial. Works such as Kaplan et al. (2025) show that while scaling model size is non-trivial, downstream performance improves with increasing dataset size, **regardless** of the model size. The authors try to explain this with data heterogeneity and model complexity in Appendix G.4., however, these two points do not address the actual reason. Could the authors elaborate on this?

**Limitations:**

Unfortunately, the authors have not discussed the limitations of their work.

**Strengths And Weaknesses:**

Strengths:
- The authors identify data heterogeneity within the pre-training dataset as challenging for learning time series representations.
- The authors compare their method against diverse baselines, ranging from task-specific models to foundational forecasting models.
- The authors evaluate their method on different downstream datasets.

Weaknesses:
- While claimed to be extensive, experiments on GIFT-Eval include only (i) a subset of the dataset and (ii) two baselines (Sempo and Moirai).
- Unfortunately, visualisations are not clear and thus do not help understanding the proposed method, particularly Figures 1, 2, and 4.
- The initial definition of domains is not clear.
- Unfortunately, the discussion seems to be superficial, only providing a brief summary of the numbers in the Tables in textual form. A thorough discussion beyond the listing of numbers would be more useful to the reader.
- It is unclear why certain datasets are chosen for experiments such as the ablation in Table 3, or why datasets even vary within experiments such as the dataset and model scaling in Figure 14.
- Unfortunately, the results are not provided across multiple seeds to support robustness of the findings.
- Unfortunately, the limitations of the proposed method are not discussed.

---

> ### Author Rebuttal · Authors · 2026-03-31
>
> We sincerely thank Reviewer p9xh for the time and effort. We address the comments point by point.
> **All tables (Table 18-21) added in this rebuttal are available in this [anonymous link](https://anonymous.4open.science/r/Olivia-TableResults-18CC).**
>
> **W1.** More experiments on GIFT-Eval
>
> **A1:**
> Our evaluation is extensive in terms of breadth across benchmarks, baselines, and settings. Specifically, Olivia is evaluated on three large-scale benchmarks (TSLib, GIFT-Eval, and GluonTS) against 19 baselines, covering zero-shot, few-shot, and full-shot forecasting.
>
> For GIFT-Eval, we follow SEMPO in using a representative subset for zero-shot evaluation. In Table 18, we further extend the GIFT-Eval zero-shot evaluation with more datasets and baselines, and the results consistently support the strong performance of Olivia.
>
> **W2.** Regarding Figures
>
> **A2:**
> Figure 1(a) shows how the signal representation changes after harmonization, Figure 2 is a structural diagram of the computation flow, and Figure 4 visualizes second-order temporal correlations to illustrate cross-dataset alignment and its generalization to unseen datasets.
>
> **W3.** Initial definition of domains
>
> **A3:**
> We follow the notion of domain used in ROSE (ICML’25). In our setting, a domain refers to a source of time series data with its own statistical patterns and temporal structures. We will make this definition more explicit in the final version.
>
> **W4.** Analytical depth of discussion
>
> **A4:** Our discussion goes beyond numerical summaries and analyzes domain vs. distribution shift (Appendix D.1), the learned orthogonal matrix (Appendix F.1–F.2), non-monotonic scaling behavior (Appendix G.4), and inference-time overhead (Appendix G.6).
>
> **W5.** Dataset selection
>
> **A5:**
> The datasets were randomly selected from representative candidates to evaluate Olivia under diverse and realistic settings, which we believe better reflects its true performance. To further confirm that the conclusions are not dataset-specific, we additionally report ablation results on all datasets related to Table 3 (Table 19) and the two scaling analyses on the same datasets (Tables 20–21), all of which consistently support our conclusions.
>
> **W6.** Robustness
>
> **A6:**
> We trained Olivia from scratch across five runs with different random seeds, and the standard deviation of zero-shot performance on TSLib (see A2 to Reviewer 9t21) demonstrates its robustness.
>
> **W7.** Limitation
>
> **A7:** One limitation is the additional computational overhead introduced by the orthogonal transformation, which leads to an efficiency trade-off compared with lighter models. We discuss this point in Appendix G.6, and we will make it more explicit in the final version.
>
> **Q1.** Clarification regarding Figure 1
>
> **A8:**
> - For 1a), a detailed explanation of the visualization is provided in A2; concerns regarding information loss are addressed in A1 to Reviewer HL5N.
> - For 1b), we first clarify that the key observation is the substantial reduction in JS divergence across all domain pairs after applying the Harmonizer, which directly demonstrates its effectiveness in aligning dataset-level PSD distributions.
> Second, regarding the ratios $JS_{DE}/JS_{EA}$, we would like to note that **such ratios are not a meaningful measure of domain discrepancy**. JS divergence is a bounded divergence measure and belongs to the class of f-divergences. It behaves as a distance-like quantity rather than a ratio-scale variable; therefore, **ratios between JS values do not admit a well-defined statistical interpretation**.
>
> **Q2.** Explanation of HarmonicAttention
>
> **A9:** Please refer to A2 to Reviewer DYg8.
>
> **Q3.** Explanation of Figure 4
>
> **A10:**
> - More explanation of Figure 4 is provided in A2.
> - Empirical evidence of how the Harmonizer affects pre-training is provided in A1 to Reviewer 9t21.
>
> **Q4.** Scaling analysis
>
> **A11:**
> We respectfully disagree with the premise of this comment. For time series foundation models, downstream performance does **not** always improve monotonically with increasing dataset size,
> nor is it independent of model size.
> As discussed in recent TSFM scaling-law studies [1],
> the benefit of additional data depends on model capacity, architecture, data quality and diversity, domain alignment, and downstream adaptation strategy.
> Therefore, the expectation that larger pre-training datasets should always improve downstream performance is not generally valid in this setting.
>
> **Importantly, phenomena similar to our observation in Figure 14 have also been observed in existing time series foundation models**. For example:
> (i) as shown in Figure 10 of SEMPO, when the pre-training data scale increases from 100% to 170%, the MSE rises from 0.376 to 0.387;
> (ii) in Table 18 of Timer, on the ETTm2 dataset, the forecasting errors of TIMER-1B, TIMER-16B, and TIMER-28B are 0.213, 0.234, and 0.247, respectively.
>
> References:
>
> [1] Towards Neural Scaling Laws for Time Series Foundation Models, ICLR 2025

---

> > ### Author Rebuttal · Reviewer_p9xh · 2026-04-01
> >
> > I would like to thank the authors for their responses. While an interesting stream of research, I remain unconvinced and believe the proposed study is not yet ready for publication. Unfortunately, (i) visualisations remain unclear, (ii) baseline results are taken from the original work without further clarification, (iii) std results are provided (a) only for Olivia and (b) only on 6 datasets not allowing to compare robustness against baselines, (iv) discussion of findings and limitations in the main manuscript are very superficial and not convincing after the rebuttal. In light of this, I cannot recommend accepting the study at this point and leave my scores unchanged. However, I strongly encourage the authors to further improve the quality of the study as the research stream is very interesting. Best regards, Reviewer p9xh

---

> > > ### Author Response · Authors · 2026-04-05
> > >
> > > Thanks for your feedback, **but unfortunately we believe your final assessment is highly biased and cannot be sufficiently supported. We sincerely request you to reconsider your acknowledgement by reading our paper and rebuttal more carefully again**.
> > >
> > > First, we have provided 11 point-by-point responses, together with additional experiments and clarifications, addressing nearly all of the concerns raised. However, the final comment does not reflect these responses and instead repeats several points as if they had not been addressed. This is particularly problematic for issues where the original criticism was itself based on incorrect interpretation, including the use of ratios between JS-divergence values and the expectation of monotonic scaling behavior in time series foundation models. In this sense, selecting (c) as the acknowledgement option does not appear fully consistent with the rebuttal record. **We highly doubt that you misunderstood it and kindly remind you to provide fair and meaningful judgement**.
> > >
> > > Second, the four newly emphasized concerns do not logically justify the conclusion that the study is “not yet ready for publication.” We are very curious whether you are an expert in deep learning for time series, since your argument seems highly biased and unreasonable, and focuses only on superficial and minor issues rather than demonstrating a proper understanding of our paper. Still, for better clarification, we respond to them as follows:
> > >
> > > (i) This statement is too vague to be actionable. In rebuttal, we already explained the distinct purposes of Figures 1, 2, and 4: Figure 1 illustrates representation changes after harmonization, Figure 2 is a structural diagram of the computation flow, and Figure 4 visualizes second-order temporal correlations and generalization to unseen datasets. Merely stating that the figures are “unclear,” without identifying a specific issue, does not provide a concrete basis for rejecting the paper.
> > > Besides, other reviewers do not mention any of this. At minimum, if you have such a claim, you would need to specify what is incorrect or misleading in the figures, rather than simply restating a subjective attitude. Based on this, we kindly remind you to have an unbiased and fair evaluation.
> > >
> > > (ii) The baseline results in our paper are reported in a strictly traceable manner, consistent with standard practice in the community. To make this explicit:
> > > - Zero-shot (Tables 1, 9, 10, 11, and 14): The baseline results (e.g., SEMPO, Time-MoE family, Timer, Moirai family, Chronos family, TimesFM, Moment, TTM, ROSE) are consistent with those reported in the original SEMPO paper (Tables 1, 8, 10, and 12). In particular, the SEMPO results on the ETTm2 dataset are consistent with those provided in the “logs/Zero-shot” of the official SEMPO GitHub repository. And the zero-shot results for ROSE are taken from Table 15 of the original ROSE paper.
> > > - Few-shot (Table 12): The baseline results (e.g., SEMPO, TTM, Time-LLM, GPT4TS, S²IP-LLM, iTransformer, DLinear, PatchTST, TimesNet, Stationary, FEDformer) are consistent with those reported in Table 15 of the original SEMPO paper.
> > > - Full-shot (Figure 3, Table 13): The baseline results (e.g., SEMPO, GPT4TS, S2IP-LLM, iTransformer, DLinear, PatchTST) are consistent with those reported in Table 13 of the original SEMPO paper.
> > >
> > > (iii) We respectfully note that this reporting choice is consistent with common practice in time series forecasting literature. It is widely accepted practice to report standard deviations on a representative subset of datasets, often only for the proposed model. For example, iTransformer (ICLR 2024) reports standard deviations for 6 out of 19 datasets and only for the proposed model in Table 5. Similarly, SOFTS (NeurIPS 2024) reports standard deviations for 6 out of 12 datasets (Table 9). **Reframing this as a decisive weakness specific to our paper is therefore not well justified. We would appreciate if you could read more papers in this domain and reconsider your ratings.**
> > >
> > > (iv) We disagree with this characterization. Our discussion goes beyond numerical summaries and includes analysis of domain shift vs. distribution shift (Appendix D.1), the properties and generalization behavior of the learned orthogonal matrix (Appendix F.1–F.2), the observed non-monotonic scaling behavior (Appendix G.4), and the inference-time overhead of Olivia (Appendix G.6). We agree that some of these points could be made more explicit in the main text; however, that is very different from claiming that the discussion is “very superficial” or that the rebuttal did not address the concern.
> > >
> > > Overall, we believe that your comments do not adequately support the conclusion you reached. **Since we have made our great efforts to push the border of the community of deep learning based time series foundation models, we hope you read more related papers in this domain, and then have a more unbiased, fair, and careful judgement towards our submission**.

---

### Official Review · Reviewer_DYg8 · 2026-03-09

**Soundness:** 4
**Presentation:** 3
**Significance:** 4
**Originality:** 4
**Overall Recommendation:** 5
**Confidence:** 4

**Summary:**

This paper proposes Olivia, a time series foundation model designed to address cross-domain heterogeneity in large-scale pretraining. The core idea is to implicitly harmonize dataset-level power spectral densities (PSDs) via a learnable orthogonal reparameterization of temporal dynamics. The authors introduce a Harmonizer module based on Householder-parameterized orthogonal transformations, which aligns second-order temporal statistics across datasets, and a HarmonicAttention mechanism that exploits the resulting low-rank spectral structure to perform efficient global interactions through a small set of resonators. Extensive experiments on TSLib, GIFT-Eval, and GluonTS benchmarks demonstrate strong zero-shot, few-shot, and full-shot forecasting performance, often outperforming recent time series foundation models.

**Compliance With Llm Reviewing Policy:**

Affirmed.

**Final Justification:**

My concerns are all addressed, so i maintain my original positive scores.

**Key Questions For Authors:**

1. Is the Harmonizer intended to enforce spectral invariance across datasets, or should it be viewed as a reparameterization of temporal correlations? Relatedly, can the Harmonizer be interpreted as learning a shared canonical coordinate system for heterogeneous time series datasets?
2. Why is an orthogonal transformation a particularly appropriate choice for PSD harmonization, as opposed to more general linear transformations?
3. HarmonicAttention is significantly more efficient than full attention. Does this efficiency come at the cost of expressive power? More specifically, how does HarmonicAttention fundamentally differ from generic low-rank or Nyström-based approximations of attention?

**Limitations:**

Yes

**Strengths And Weaknesses:**

Strengths

1. The paper identifies a concrete and well-motivated challenge in time series foundation models, cross-domain spectral heterogeneity, and formulates it using normalized PSDs, which is a natural and theoretically grounded descriptor for temporal structure.
2. The use of orthogonal reparameterization to decouple shared and domain-specific second-order statistics is elegant. Proposition 1 and Proposition 2 provide non-trivial theoretical justification for both the Harmonizer and the low-rank interaction assumption behind HarmonicAttention.
3. Unlike many prior works that rely on MoE, prompts, or domain-specific heads, Olivia enforces a shared latent structure via PSD-consistent alignment. This represents a more structural and arguably more fundamental approach to domain generalization.
4. The model is evaluated across multiple benchmarks, settings (zero-shot, full-shot), and baselines, including recent foundation models. The gains are consistent and often substantial, especially under zero-shot transfer.

Weaknesses

1. Although PSD harmonization is well motivated, the conceptual and practical differences from stationarity normalization techniques (e.g., RevIN or instance normalization) could be articulated more clearly, particularly with respect to their impact on temporal structure.
2. While the theoretical analysis motivates HarmonicAttention, its practical implications could be more explicitly discussed.

---

> ### Author Rebuttal · Authors · 2026-03-31
>
> We sincerely thank Reviewer DYg8 for the insightful comments. We will respond to the comments point by point below.
>
> **W1.** Difference from normalization methods in temporal structure modeling
>
> **A1:**
> We have partially investigated this aspect in Appendix D.1 (Distribution Shift vs Domain Shift). Here, we further clarify the conceptual differences between normalization-based methods and our approach.
> Normalization techniques such as RevIN operate on marginal statistics (e.g., mean and variance) in a point-wise manner, and mainly aim to mitigate instance-level distribution shift. In contrast, our method performs an orthogonal transformation across the temporal dimension, directly acting on second-order temporal correlations, which are closely related to PSD via the Wiener–Khinchin theorem. As a result, normalization may alter amplitude-related properties but does not explicitly address temporal dependency structures, whereas our method reorganizes temporal correlations while preserving all information due to orthogonality.
>
> This distinction is also supported by the empirical results in Table 5. We observe that applying RevIN alone does not significantly reduce cross-domain JS divergence, while the Harmonizer consistently achieves substantial alignment both with and without RevIN. This indicates that normalization and our method operate at different levels and have complementary roles.
>
> **W2&Q3.** Explanation of HarmonicAttention
>
> **A2:** We clarify the design and implications of HarmonicAttention as follows:
> - **(i) What is HarmonicAttention.** HarmonicAttention replaces dense token–token interactions with interactions mediated by a small set of shared resonators, enabling global information exchange in a low-dimensional space.
> This design is motivated by the low-rank structure of PSD-aligned token interactions (Proposition 2), which shows that token correlations can be decomposed into a dominant low-rank principal component with a bounded residual.
> In practice, it is implemented via a token–resonator–token interaction scheme, as illustrated in Figure 2(c), enabling efficient global information exchange with reduced complexity.
> - **(ii) Difference from existing efficient attention mechanisms.**
> Existing efficient attention mechanisms (e.g., linear-based or Nyström-based approaches) typically approximate the full attention matrix via decomposition or sampling.
> In contrast, HarmonicAttention does not rely on approximating dense attention, but is designed in accordance with the structured low-rank property of PSD-aligned token interactions (Proposition 2), defining a structured interaction mechanism in a low-dimensional space.
> As a result, it should be viewed as redefining the interaction mechanism rather than approximating standard attention.
> - **(iii) Effectiveness and sensitivity.** HarmonicAttention does not sacrifice expressive power. As shown in Table 19 in this [anonymous link](https://anonymous.4open.science/r/Olivia-TableResults-18CC), it consistently outperforms full attention, as well as Nyström and linear attention, across multiple datasets.
> In addition, performance remains stable across different choices of the resonator number $M$ (see A2 to Reviewer HL5N), indicating low sensitivity to this hyperparameter and suggesting that the low-dimensional interaction space does not impose a restrictive bottleneck.
>
> **Q1.** Interpretation of Harmonizer’s role
>
> **A3:**
> The Harmonizer is not intended to enforce spectral invariance across datasets. It does not explicitly match or constrain the power spectral density.
> Instead, it should be viewed as an orthogonal reparameterization of temporal correlations, which reorganizes second-order structure through an invertible and energy-preserving transformation. This allows different datasets to be expressed in a more consistent form without removing domain-specific information.
> From this perspective, the Harmonizer can be interpreted as learning a shared coordinate system for heterogeneous time series, where signals are mapped into a common representation space while preserving their intrinsic characteristics. We will clarify this interpretation in the revision.
>
> **Q2.** Justification for orthogonal transformation choice
>
> **A4:**
> Orthogonal transformations are particularly suitable as they are energy-preserving and invertible, corresponding to a change of basis rather than a distortion of the signal. This ensures that the overall spectral energy is maintained, which is important for PSD-related representations.
> In contrast, general linear transformations may introduce scaling or distortion, potentially altering the energy distribution or causing information loss.
> Therefore, orthogonality allows the Harmonizer to reparameterize temporal correlations without modifying the underlying signal content, aligning with our goal of harmonizing structure rather than changing the signal.

---

> > ### Author Rebuttal · Reviewer_DYg8 · 2026-04-02
> >
> > My concerns have been adequately addressed.

---

### Official Review · Reviewer_9t21 · 2026-03-13

**Soundness:** 4
**Presentation:** 3
**Significance:** 4
**Originality:** 3
**Overall Recommendation:** 5
**Confidence:** 4

**Summary:**

The authors investigate cross-domain heterogeneity in large-scale time series pretraining from a spectral perspective. Rather than directly minimising discrepancies between dataset-level PSDs, the authors propose to encourage spectral consistency through a learnable orthogonal reparameterization based on Householder reflections. Building on this idea, the paper further introduces a compact attention mechanism that leverages the resulting low-rank structure for more efficient temporal modelling. The overall framework is instantiated in a time series foundation model, Olivia, and evaluated on several large-scale benchmarks under zero-shot, few-shot, and full-shot settings.

**Compliance With Llm Reviewing Policy:**

Affirmed.

**Final Justification:**

After reading the rebuttal, I would like to keep my original score with high confidence. I could also defend for my original evaluation and am open to discussion if needed.

**Key Questions For Authors:**

1) Could the authors provide a more detailed complexity analysis of the Householder reflections used in the Harmonizer? In particular, it would be helpful to understand how the computational cost scales with the number of learnable reflections and how sensitive the model performance is to this hyperparameter.
2) During the tuning stage, the Harmonic Encoder is frozen while the Aligner remains learnable. Did the authors experiment with freezing the Aligner as well? Such an ablation could help clarify whether the pre-harmonised representation space learned during pretraining is already sufficiently universal.
3) Are there other potential approaches for addressing cross-domain heterogeneity beyond the one explored in this paper? It would be interesting to hear the authors’ perspective on alternative directions that might also be viable.

**Limitations:**

See Strengths And Weaknesses.

**Strengths And Weaknesses:**

Strengths:
1) The use of normalised PSD as a dataset-level descriptor to characterise cross-domain heterogeneity is well motivated from a signal processing perspective. This formulation also distinguishes the work from many existing approaches that primarily rely on architectural modifications or increased model capacity for domain generalization.
2) The design of Harmonizer and Harmonic attention appears conceptually coherent. In particular, the attention mechanism is naturally aligned with the low-rank structure induced by spectral harmonization, rather than functioning merely as an ad-hoc efficiency modification.
3) The empirical evaluation is relatively extensive. Experiments on several large-scale benchmarks show consistent improvements under zero-shot, few-shot, and full-shot settings, which provides reasonable empirical support for the proposed framework.

Weaknesses:
1) The paper largely focuses on the representational benefits of spectral harmonization, while the potential implications of the Harmonizer on optimization dynamics during pretraining are less clearly discussed. It would be helpful to understand how the orthogonal transformation interacts with gradient flow during training.
2) Given the non-convex nature of the training objective and the flexibility of the orthogonal parameterization, it would be useful to examine the stability of the learned harmonised representations across different random seeds.

---

> ### Author Rebuttal · Authors · 2026-03-31
>
> We appreciate the reviewer’s careful assessment and thoughtful suggestions. We will respond to the comments point by point below.
>
> **W1.** Impact of the orthogonal transformation on gradient flow and optimization
>
> **A1:**
> - From a theoretical perspective, orthogonal transformations preserve inner products and therefore maintain vector norms[1].
> As a result, gradient magnitudes are not distorted during backpropagation. This property is well established in deep learning literature, where orthogonal matrices are known to preserve gradient norms and mitigate vanishing/exploding gradients, thereby stabilizing training [2,3].
> - Empirically, we further observe that the Harmonizer improves optimization dynamics during pretraining. As shown in the following table (values are scaled by 1000 for clarity), models with the Harmonizer achieve lower losses in early epochs, indicating faster convergence, and exhibit more stable training with reduced fluctuations.
> We attribute this to the fact that the Harmonizer projects heterogeneous time series into a shared coordinate system, reducing inconsistencies in second-order temporal structures across domains. This leads to more coherent gradient directions during training, thereby facilitating more stable and efficient optimization.
>
> |Pretrain Epoch||1||2||3||4||5||6||7||8||9||10|
> |-|-|-|-|-|-|-|-|-|-|-|-|-|-|-|-|-|-|-|-|-|
> |Loss|Vali|Test|Vali|Test|Vali|Test|Vali|Test|Vali|Test|Vali|Test|Vali|Test|Vali|Test|Vali|Test|Vali|Test|
> |w Harmonizer|2.0014|1.9124|0.6708|0.6334|0.6619|0.6224|0.6161|0.5821|0.8118|0.7677|0.7356|0.6957|0.7571|0.7134|1.0803|1.0195|0.8142|0.7633|0.7394|0.6959|
> |w/o Harmonizer|2.3020|2.1503|1.2120|1.1233|0.7314|0.6718|0.8977|0.8244|1.1086|1.0292|0.7994|0.7443|1.4066|1.3038|1.2308|1.1449|1.0373|0.9575|-|-|
>
> **W2.** Robustness of Olivia performance
>
> **A2:**
> We trained Olivia from scratch across five random seeds and report the standard deviation of zero-shot performance on TSLib. The low variance indicates strong robustness.
>
> |Datasets|ETTh1|ETTh2|ETTm2|Weather|Electricity|Traffic|
> |-|-|-|-|-|-|-|
> |MSE|0.399$\pm$0.001|0.339$\pm$0.003|0.291$\pm$0.001|0.247$\pm$0.002|0.188$\pm$0.002|0.458$\pm$0.000|
> |MAE|0.421$\pm$0.002|0.388$\pm$0.002|0.344$\pm$0.001|0.287$\pm$0.001|0.285$\pm$0.002|0.330$\pm$0.001|
>
> **Q1.** Complexity and sensitivity of the Householder reflection parameterization
>
> **A3:**
> - Regarding complexity, each Householder reflection can be applied in $O(T)$, leading to an overall cost of $O(KT)$ for $K$ reflections. In practice, this remains modest compared to quadratic-cost attention when $K≪T$.
> - Regarding sensitivity to $K$, we have already provided a detailed analysis in Figure 15. The results show that performance remains stable across a wide range of $K$ values (e.g., 64–336), with only minor variations in both MSE and MAE. Notably, no monotonic improvement is observed with increasing $K$, indicating that the model does not rely on careful tuning of this hyperparameter. This suggests that the gains mainly stem from structural alignment rather than increased parameterization.
>
> **Q2.** Freezing Aligner ablation
>
> **A4:**
> We conducted an ablation by freezing the Aligner during tuning. As shown in the following table, the learnable Aligner achieves better or comparable performance than the frozen variant.
> This suggests that while the harmonised representation learned during pretraining is already strong and largely transferable, retaining a learnable Aligner provides additional flexibility for adapting to task-specific variations. This empirical evidence supports our design choice in Figure 7 to keep the Aligner learnable during the tuning stage.
>
> |Datasets||ETTh1||ETTh2||ETTm2||Weather||Electricity||Traffic|
> |-|-|-|-|-|-|-|-|-|-|-|-|-|
> |Metrics|MSE|MAE|MSE|MAE|MSE|MAE|MSE|MAE| MSE |MAE|MSE|MAE|
> |learnable|0.399|0.421|**0.339**|**0.388**|**0.291**|**0.344**|**0.247**|**0.287**| **0.188** |**0.285**|0.458|0.330|
> |frozen|**0.394**|**0.419**|0.344|0.392|0.301|0.354|0.252|0.292|0.189|0.286|**0.455**|**0.326**|
>
> **Q3.** Other potential approaches for addressing cross-domain heterogeneity
>
> **A5:**
> Cross-domain heterogeneity can be addressed via distribution alignment (e.g., adversarial or divergence-based methods), normalization, or domain-invariant representation learning. These approaches primarily operate on marginal statistics or instance-level features, reflecting different strategies for reducing distribution discrepancies.
> Our method aligns second-order temporal correlations via orthogonal reparameterization, providing a complementary, structure-level perspective. Exploring combinations of these directions is a promising avenue for future work.
>
>
> References:
>
> [1] Linear algebra and its applications, 2012
>
> [2] On orthogonality and learning recurrent networks with long term dependencies (ICML 2017)
>
> [3] Existence, Stability and Scalability of Orthogonal Convolutional Neural Networks (JMLR 2022)

---

> > ### Author Rebuttal · Reviewer_9t21 · 2026-04-04
> >
> > Thanks for the detailed response. All of my concerns have been addressed. I also noticed there are some different opinions from other reviewers that cannot convince me. Thus, I would like to keep my original score and defend for my original evaluation if needed. I am also open to further discussion.

---

### Official Review · Reviewer_HL5N · 2026-03-17

**Soundness:** 3
**Presentation:** 4
**Significance:** 3
**Originality:** 3
**Overall Recommendation:** 4
**Confidence:** 2

**Summary:**

This paper introduces Olivia,a time series foundation model, utilizing Power Spectral Density to harmonize cross-domain dataset, thereby improving the effectiveness of large-scale pre-training.
Recognizing that direct joint training often yields suboptimal results due to varying temporal patterns across domains, the authors propose the Harmonizer module.
This module implicitly aligns spectral structures across datasets via orthogonal transformations, achieving a reparameterization of second-order temporal correlations.
To complement the aligned feature space, a HarmonicAttention mechanism is designed to interact through a small set of resonators in a low-dimensional space, reducing computational complexity.
Extensive experiments on benchmarks such as TSLib, and GluonTS display that Olivia achieves state-of-the-art performance across zero-shot, dew-shot(appendix)and full-shot forecasting tasks.
·Oblivia tackles the domain shift problem in foundation model pre-training from a signal processing perspective is both unique and inspiring.

**Compliance With Llm Reviewing Policy:**

Affirmed.

**Final Justification:**

I am not an expert of this field. But the author address my concern so I would like to change from marginal reject to marginal accept.

**Key Questions For Authors:**

1.How to ensure that projecting time series, or interacting through a number of resonators, doesnt lose specific signals and high-frequency burst features for certain prediction tasks
, such as anomaly detection or extreme weather forecasting?

2.As the Weakness.2: Given that the computational complexity of HarmonicAttention is reduced, how sensitive is the model's prediction Acc to the choice of the resonator number M?
Could a mechanism be introduced in the future to adaptively determine the value of M when dealing with information-density-different datasets?

3.This paper performs well in handling domain shift between different datasets, but if it encounters severe distribution shift within a single dataset, since PSD calculation relies on global statistics
, how would Olivia's alignment mechanism cope with such dynamic internal shifts?

**Limitations:**

Yes

**Strengths And Weaknesses:**

## Strength

1.Olivia introduces power spectral density (PSD) and Jensen-Shannon divergence into time series pre-training.
 This mathematical alignment is more  interpretable than simple feature concatenation.

2.The architecture uses interaction via M compact resonators, successfully reducing attention complexity from O(L^2P)to O(LMP+M^2P),
 maintaining a global receptive field while significantly boosting inference speed for long sequences,which is effective.

3.On large-scale independent benchmarks including TSLib, Olivia achieves significant and consistent error (MSE and MAE) reductions across different forecasting settings
 with far fewer parameters compared to peer foundational models like Moirai etc..

## Weaknes

1.Although the Harmonizer preserves second-order correlations through orthogonal transformations, the process of compressing and reorganizing time series
to fit a globally shared PSD distribution may filter out high-frequency abrupt changes or local non-stationary features that are critical in certain domains.
In real scenarios with extreme volatility, this might limit the model's peak prediction capability.

2.HarmonicAttention relies on a fixed number M of resonators as a bottleneck for information interaction. The paper lacks extensive ablation studies on M.
A fixed M may fail to achieve optimal capacity allocation for downstream datasets with different complexities.

3.The paper primarily demonstrates the method's strength in handling cross-dataset domain shift. However, we can see in real time series, severe internal concept drift is common.
Whether the globally static PSD-alignment method remains robust under the dynamic scenarios lacks sufficient empirical evidence.

---

> ### Author Rebuttal · Authors · 2026-03-30
>
> We sincerely thank Reviewer HL5N for the careful reading and constructive feedback. We will respond to the comments point by point below.
>
> **W1&Q1.** About the information processing
>
> **A1:** We would like to clarify that Harmonizer is **not a compressive or filtering operation**.
> Instead, it performs an **invertible, orthogonal, and energy-preserving reparameterization** of the temporal coordinate system.
> As the transformation is invertible and energy-preserving, it improves cross-domain consistency while preserving the underlying signal content [1-2]. Therefore, it does **not explicitly suppress high-frequency bursts or local non-stationary patterns**. Empirically, Olivia achieves strong performance on volatile datasets (e.g., Electricity and Traffic), suggesting that the proposed harmonization does not undermine predictive fidelity in volatile settings.
>
> Similarly, the resonators in HarmonicAttention do not act as a hard truncation of information. Instead, they provide a **structured low-rank communication space** for global interaction, motivated by the observation that the dominant cross-token dependencies in the PSD-aligned space can be captured compactly. Empirically, our sensitivity analysis (as shown in A2) on the resonator number shows that a relatively small number of resonators already achieves the best forecasting performance, suggesting that the mechanism captures the dominant interaction structure without sacrificing predictive fidelity.
>
> **W2&Q2.** About the resonator number M
>
> **A2:** The design of HarmonicAttention is motivated by the low-rank structure of token interactions in the PSD-aligned space (Proposition 2), where most temporal energy is concentrated in a small number of dominant modes. Therefore, **M serves as an efficiency–capacity trade-off parameter, rather than a fragile bottleneck**.
>
> To verify this empirically, we conducted a sensitivity analysis with M $\in$ \{ $L/4, L/2, L$}, with results averaged over forecasting horizons {96,192,336,720}. As shown in the following table, performance shows only moderate variations across different choices of M. Notably, **smaller values (e.g., M $= L/4$) already achieve the best performance across all datasets**, while increasing M does not yield improvements and may even introduce slight degradation.
> **This suggests that the proposed mechanism effectively captures the dominant structure without requiring a large resonator set.** We will add this analysis in our final version.
>
> We nevertheless agree that adaptive resonator allocation (e.g., data-dependent or learnable M) is a promising direction for handling datasets with different information densities, and we will explore this as an important future extension.
>
> | Datasets ||ETTh1||ETTh2||ETTm2||Weather||Electricity||Traffic|
> |-|-|-|-|-|-|-|-|-|-|-|-|-|
> | Metrics |MSE|MAE|MSE|MAE|MSE|MAE|MSE|MAE|MSE|MAE|MSE|MAE|
> | $M=L/4$  |**0.399**|**0.421**|**0.339**|**0.388**|**0.291**|**0.344**|**0.247**|**0.287**|**0.188**|**0.285**|**0.458**|**0.330**|
> | $M=L/2$  |0.404|0.425|0.345|0.391|0.298|0.350|0.251|0.290|0.190|0.286|0.462|**0.330**|
> | $M=L$    |0.404|0.424|0.350|0.393|0.305|0.354|0.255|0.292|0.191|0.288|0.459|0.332|
>
> **W3&Q3.** About the distribution shift within a single dataset
>
> **A3:**
> The scenario described here corresponds to intra-dataset distribution shift, which is distinct from the main problem studied in our paper, namely cross-dataset domain shift.
>
> In Olivia, these two challenges are handled by different components, namely RevIN and the Harmonizer. RevIN  mitigates instance-level non-stationarity through input normalization, while the Harmonizer is designed to reduce structural mismatch across datasets.
> **We have clarified this in Appendix D.1**.
> Therefore, Olivia includes mechanisms that improve robustness to both cross-dataset shift and intra-dataset non-stationarity.
>
> Empirically, we provide multiple pieces of evidence.
> (i) As detailed in the following table, ADF statistics[3] consistently decrease after applying the Harmonizer-Aligner, indicating a more stable temporal representation.
> (ii) Olivia achieves strong zero-shot performance (Table 1) on widely recognized non-stationary TSLib datasets[4],
> demonstrating its ability to model time-varying distributions.
> (iii) Figure 4 shows that the learned alignment generalizes well to datasets not used during pretraining,
> suggesting that the method is not tied to a fixed global distribution but remains flexible under varying data conditions.
>
> | Datasets       | ETTh1  |ETTh2| ETTm2  | Weather | Electricity | Traffic |
> |-|-|-|-|-|-|-|
> | Before Aligner | -3.411 |-2.895| -3.843 | -3.222 | -4.483| -5.385 |
> | After Aligner  | -8.839 |-9.570| -8.982 | -9.132 | -10.488 | -8.634 |
>
> References:
>
> [1] Linear algebra and its applications. 2012.
>
> [2] Matrix analysis. 2012.
>
> [3] Efficient tests for an autoregressive unit root. 1992.
>
> [4] Frequency Adaptive Normalization For Non-stationary Time Series Forecasting (NeurIPS 2024)

---

> > ### Author Rebuttal · Reviewer_HL5N · 2026-04-08
> >
> > I am not an expert of this field. But the author address my concern so I would like to change from marginal reject to marginal accept.

---

### Decision · Program_Chairs · 2026-04-30

**Decision:**

Accept (regular)

**Comment:**

I recommend acceptance because the paper presents a novel and technically coherent approach to cross-domain heterogeneity in time series foundation models. Its central idea (implicitly harmonizing power spectral densities through an orthogonal reparameterization) is well motivated, and the proposed HarmonicAttention mechanism makes effective use of the resulting low-rank structure to model global dependencies efficiently. The paper is also supported by broad empirical evidence across multiple large-scale benchmarks, with ablations indicating that both the harmonization component and the attention design contribute meaningfully to the gains. While the presentation can be improved in places, the core contribution is technically sound, distinctive, and likely to be useful to the time series forecasting community.